# Single-molecule dynamics of the TRiC chaperonin system in vivo

Rongqin Li[1,3], Niko Dalheimer[1,3], Martin B. D. Müller[2] & F. Ulrich Hartl[1✉]

The essential chaperonin T-complex protein ring complex (TRiC) (also known as chaperonin containing TCP-1 (CCT)) mediates protein folding in cooperation with the co-chaperone prefoldin (PFD)[1–5]. In vitro experiments have shown that the cylindrical TRiC complex facilitates folding through ATP-regulated client protein encapsulation[6–9]. However, the functional dynamics of the chaperonin system in vivo remain unexplored. Here we developed single-particle tracking in human cells to monitor the interactions of TRiC–PFD with newly synthesized proteins. Both chaperones engaged nascent polypeptides repeatedly in brief probing events typically lasting around one second, with PFD recruiting TRiC. As shown with the chaperonin client actin[8], the co-translational interactions of PFD and TRiC increased in frequency and lifetime during chain elongation. Close to translation termination, PFD bound for several seconds, facilitating TRiC recruitment for post-translational folding involving multiple reaction cycles of around 2.5 s. Notably, the lifetimes of TRiC interactions with a folding-defective actin mutant were markedly prolonged, indicating that client conformational properties modulate TRiC function. Mutant actin continued cycling on TRiC until it was targeted for degradation. TRiC often remained confined near its client protein between successive binding cycles, suggesting that the chaperonin machinery operates within a localized 'protective zone' in which free diffusion is restricted. Together, these findings offer detailed insight into the single-molecule dynamics and supramolecular organization of the chaperonin system in the cellular environment.

Protein folding is assisted by molecular chaperones, both during and after translation[1,2,10–13]. A key unresolved aspect of this fundamental process is the dynamic behaviour of chaperone systems in vivo: how long, how often and when during protein biogenesis chaperones interact with their client proteins. In eukaryotic cells, the chaperones nascent polypeptide-associated complex (NAC) and ribosome-associated complex (RAC) interact early with nascent chains on the ribosome[10,13]. As translation proceeds, nascent chains may be engaged by Hsp70–Hsp40 as well as the chaperonin TRiC and its co-chaperone prefoldin[2,9,14]. TRiC is an abundant approximately 900 kDa double-ring protein complex with ATPase activity that transiently encapsulates non-native substrate protein for folding in its central chamber[6–9]. Each ring consists of eight essential, paralogous subunits (CCT1–8) that feature a common three-domain architecture[9] (Extended Data Fig. 1a). The apical domains bind unfolded protein substrate at the ring opening and mediate its enclosure through α-helical extensions that form an iris-like lid. This is triggered by ATP hydrolysis in the equatorial domains and conducted by the intermediate domains. TRiC interacts with 5–10% of newly synthesized proteins, including many essential proteins with complex fold topologies, such as actin, tubulins and WD repeat-containing protein 40 (WDR40) domain proteins[15–17]. Mutations in several TRiC subunits are the cause of developmental brain malformations[18]. Although TRiC alone promotes folding of chemically denatured proteins such as actin in vitro[8], it functionally cooperates in vivo with its major co-chaperone PFD and with phosducin-like protein 2A (PhL-P2A)[2,5,19]. PFD, a hetero-hexamer complex of coiled-coil subunits[20] (Extended Data Fig. 1b), is thought to deliver client proteins into the TRiC chamber[21] and is required for efficient actin folding, enhancing both folding rate and yield[22].

Despite extensive mechanistic and structural studies, the functional dynamics of the TRiC chaperonin system remain largely unexplored. Given that the highly crowded cellular environment can profoundly affect protein–protein interactions[23–25], we developed single-particle tracking (SPT) strategies[26–29] to directly observe the co- and post-translational interactions of TRiC and PFD with client proteins in live human cells. This methodological platform is broadly applicable to analysing the function of different chaperone systems, co-factors and client proteins in vivo.

## Co-translational chaperonin function

To analyse the interactions of TRiC and PFD with nascent chains on translating ribosomes, we expressed the chaperones in human U-2 OS cells as fusions with the self-labelling HaloTag[30] and the large ribosomal

[1]Department of Cellular Biochemistry, Max Planck Institute of Biochemistry, Martinsried, Germany. [2]Gene Center, Department of Biochemistry, Ludwig Maximilian University of Munich, München, Germany. [3]These authors contributed equally: Rongqin Li, Niko Dalheimer. ✉e-mail: uhartl@biochem.mpg.de

protein L10A (RPL10A) with SNAP tag (SNAP–L10A)[31] (Fig. 1a). TRiC–Halo was generated by CRISPR knock-in of the HaloTag into a loop region of TRiC subunit CCT4[32] (Extended Data Fig. 1a,c). Tagged CCT4 co-assembled into the TRiC complex and interacted with client proteins and PFD (Extended Data Fig. 1d). To label PFD, we attached the HaloTag to the C terminus of endogenous PFD4 using CRISPR-Cas9 editing[21], thus enabling functional assembly of the PFD complex (Extended Data Fig. 1b,e,f). SNAP–L10A was stably expressed (in addition to endogenous L10A) and assembled quantitatively into translating ribosomes and polysomes (Extended Data Fig. 1g–i).

To enable single-molecule detection, we labelled about 3% of TRiC or 9% of PFD molecules (sparse labelling) with membrane-permeant Janelia Fluor (JF) dyes (covalently binding to HaloTag)[33] and equivalent numbers of ribosomes with trifluoromethyl fluorobenzyl pyrimidine (TF)–tetramethylrhodamine (TMR) dye (covalently binding to SNAP-tag)[34] for dual-colour analysis (Fig. 1a and Extended Data Fig. 1j–l). This approach ensured comparable numbers of labelled PFD and TRiC complexes, as PFD is about three times less abundant than TRiC[35]. The puromycin-sensitive spatial and temporal correlation of TRiC–PFD and ribosome movement trajectories served as criteria to define chaperone–nascent chain interactions. Puromycin-insensitive contacts lasting less than 0.5 s were defined as non-specific (Methods). Both TRiC and PFD engaged nascent chains in brief interactions typically lasting for around 1 s (Fig. 1b–e, Extended Data Fig. 1m,n and Supplementary Videos 1 and 2), with PFD apparently binding more frequently to nascent chains than TRiC (Fig. 1e). The interaction data were fitted with a single-exponential decay model suggesting an average lifetime of chaperone binding to translating substrates of around 0.8 s (Extended Data Fig. 1o,p).

Knockdown of PFD subunit 3 (PFD3) leads to downregulation of the entire prefoldin complex (referred to as PFD knockdown), whereas TRiC levels remain unchanged (Extended Data Fig. 2a–c and Methods). PFD knockdown caused reduction of around 65% in the frequency of TRiC–nascent chain interactions (Fig. 1d,e), suggesting a critical role of PFD in TRiC recruitment.

To further characterize the range of nascent chain clients of both chaperones, we used selective ribosome profiling[17,36]. We identified 511 and 1,174 interactors of TRiC–Halo and PFD–Halo, respectively (Extended Data Fig. 3a–c and Methods), in line with a higher frequency of PFD–nascent chain contacts (Fig. 1e). Most TRiC interactors (around 80%), including actin and tubulins, also interacted with PFD (Extended Data Fig. 3d–f), consistent with PFD recruiting TRiC. Analysis of the relative positions of ribosome-protected fragment (RPFs) from shared substrates suggested that both PFD and TRiC recognize multiple sites along nascent chains with only minimal (approximately 1%) overlap (Extended Data Fig. 3g), with the first PFD binding preceding TRiC binding in 64% of open reading frames (Extended Data Fig. 3h). Notably, the shared substrates are enriched in TRiC subunits as well as Hsp90 and Hsp70 chaperone proteins (Extended Data Fig. 3i), highlighting the central role of the PFD–TRiC axis in maintenance of proteostasis.

Together, these findings indicate that both TRiC and PFD monitor a broad range of nascent chains primarily through brief, dynamic interactions, with PFD serving as recruitment factor for TRiC for most substrates.

## Interactions with nascent actin

To investigate the co-translational interactions of TRiC–PFD with an obligate client protein of the chaperonin system, we selected β-actin[8,21]. To avoid possible artefacts from tagging the actin nascent chain itself, we instead labelled the actin mRNA and monitored the colocalization of chaperone and mRNA signals. To this end, we inserted 24 copies of the MS2 phage stem-loop sequence in the 3′ untranslated region of the β-actin mRNA, which enabled binding by constitutively expressed GFP-tagged MS2 coat protein (MCP–GFP)[37,38] for visualization (Fig. 1f).

Unbound MCP–GFP was targeted to the nucleus[39] to reduce cytosolic background fluorescence. This system enabled the observation of individual, translationally active molecules of actin mRNA in the cytosol (Extended Data Fig. 4a,b).

Dual-colour imaging revealed colocalization events of actin mRNA–MS2 with TRiC–Halo (Extended Data Fig. 4c,d and Supplementary Video 3) and PFD–Halo (Extended Data Fig. 4e,f and Supplementary Video 4). These interactions represented chaperone binding to actin nascent chains: they were markedly reduced in duration and frequency by puromycin or upon translation of an mRNA–MS2 construct encoding only the first 20 amino acids of actin (Fig. 1g and Extended Data Fig. 4g) that is expected to reside within the ribosomal exit tunnel, which is inaccessible for TRiC. The interactions of TRiC with actin nascent chains upon translation of full-length actin displayed an average lifetime of approximately 1.2 s (Fig. 1g), similar to the interactions with nascent chains in general (Fig. 1d), but with a greater proportion (around 37%) of binding events with durations of more than 1 s (Fig. 1g and Extended Data Fig. 4h). PFD knockdown reduced TRiC binding to actin nascent chains by around 70% (Fig. 1g and Extended Data Fig. 4g), confirming the role of PFD in recruiting TRiC. Notably, the TRiC interactions remaining after PFD knockdown were shifted to shorter lifetimes (Fig. 1g), with only 11% of binding events lasting for more than 1 s (Extended Data Fig. 4h). This suggests that PFD, beyond recruiting TRiC, prolongs TRiC–nascent chain associations, either through direct interaction or by acting as a holding chaperone in stabilizing actin in a conformation that is competent for TRiC binding.

Both TRiC and PFD have multiple substrate binding sites, which suggested that their co-translational interactions might be dependent on nascent chain length. β-Actin (375 amino acids) has a discontinuous fold consisting of two domain lobes, each divided into structurally interdependent subdomains (SD1–SD2 and SD3–SD4)[40] (Fig. 1h). SD1 contains both N-terminal and C-terminal residues of actin and thus must fold post-translationally. To explore a possible length dependence of nascent chain binding, we generated MS2-labelled mRNAs expressing truncated actin chains of 101, 168, 305 and 368 amino acids with termination codons, avoiding translation arrest (Fig. 1i). Similar numbers of ribosome-engaging mRNA molecules were detected in the cytosol for these constructs (Extended Data Fig. 5a,b). Both PFD and TRiC interacted with the respective nascent chains in a length-dependent manner with respect to dwell time and binding frequency (Fig. 1j,k). PFD binding events with actin(1–305), exposing up to the complete SD2 and SD4, were around 4 times more frequent than binding to actin(1–101), exposing only the N-terminal segment of SD1 (Fig. 1i, j and Extended Data Fig. 5c), consistent with the longer availability of actin(1–305) nascent chains on the ribosome during translation. In addition, an increase in interaction lifetimes during translation was observed from actin(1–305) to actin(1–368) to full-length actin (Fig. 1j), suggesting that PFD binding strength was also enhanced by emergent conformational properties of the growing nascent chains. Remarkably, the average duration of PFD binding events increased from about 1.6 s for actin(1–368) to 2.1 s for full-length actin (Fig. 1j), although actin(1–368) and full-length actin differ by only 7 amino acids. Indeed, whereas the binding data for PFD–actin(1–101) fitted to a single-exponential decay with a lifetime of approximately 1.0 s, the lifetime data for actin(1–368) and full-length actin were bimodal, with 24% of interactions having a lifetime of around 6 s when full-length actin was translated (Extended Data Fig. 5d). Thus, at a translation speed of about 5 amino acids per second[41], PFD would bind actin nascent chains increasingly during translation of the approximately 30 C-terminal amino acids, consistent with data from PFD-selective ribosome profiling for actin (Extended Data Fig. 5f). The combined increase in dwell time and frequency of binding suggest a substantial increase in affinity of PFD for long actin chains, consistent with the presence of multiple chaperone sites in the PFD hexamer.

TRiC binding mirrored the chain length dependence of the PFD interactions (Fig. 1k and Extended Data Fig. 5f). Both the frequency

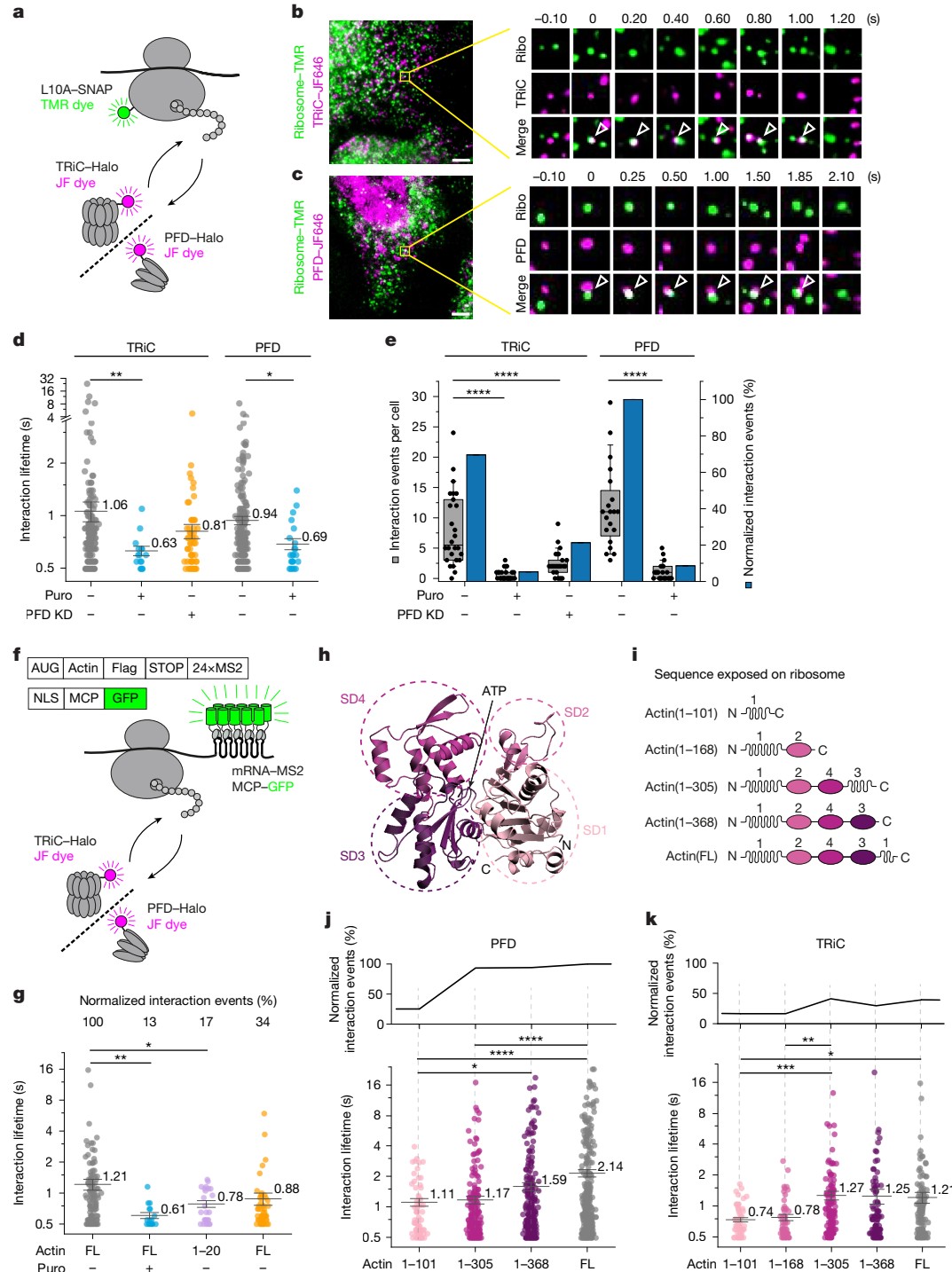

**Fig. 1 | Interactions of TRiC–PFD with nascent polypeptides. a**, Schematic of imaging of TRiC–PFD and ribosomes. **b**,**c**, TRiC–ribosome (**b**) and PFD–ribosome (**c**) associations. Left, dual-colour imaging. Right, selected time points. Scale bars, 5 μm. **d**, Duration of chaperone–ribosome interactions (number of cells (*n*): TRiC, no treatment, *n* = 26; Puro, *n* = 26; PFD3 knockdown (KD), *n* = 26; PFD, no treatment, *n* = 20; Puro, *n* = 23). The centre line is the mean and error bars show s.e.m. **P* = 0.0086 (one-way Welch's ANOVA); *P* = 0.0314 (Mann–Whitney test). Puro, puromycin. **e**, Co-movement events per cell (grey) and normalized co-movement events (total interactions divided by number of cells; blue) of interactions as in **d**. The horizontal line indicates the median, boxes delineate top and bottom quartiles and whiskers extend between 10th and 90th percentiles. ****P* < 0.0001 (one-way ANOVA); *****P* < 0.0001 (Mann–Whitney test). **f**, Schematic of dual-colour imaging of TRiC–PFD and translating actin. Actin mRNA is imaged with GFP-tagged MS2 coat protein (GFP–MCP).

**g**, Number of normalized co-movement events (top) and distribution of duration (bottom) for actin–TRiC interactions (number of cells (*n*): no treatment, *n* = 38; Puro, *n* = 40; actin(1–20), *n* = 42; PFD knockdown, *n* = 45). The centre line is the mean and error bars show s.e.m. **P* = 0.042, ***P* = 0.006 (one-way Welch's ANOVA). **h**, Actin structure (Protein Data Bank (PDB) visualization of AlphaFold predicted structure: AF_AFP60709F1). **i**, Actin truncations used in experiments. FL, full length. **j**,**k**, Normalized frequency of co-movement events (top) and distribution of duration (bottom) of actin truncations with PFD (**j**) or TRiC (**k**) (number of cells (*n*): PFD–actin: actin(1–101), *n* = 29; actin(1–305), *n* = 30; actin(1–368), *n* = 33; actin(FL), *n* = 38; TRiC–actin: actin(1–101), *n* = 42; actin(1–168), *n* = 43; actin(1–305), *n* = 36; actin(1–368), *n* = 39; actin(FL), *n* = 38). The centre line is the mean and error bars show s.e.m. PFD–actin, **P* = 0.036, *****P* < 0.0001; TRiC–actin, **P* = 0.0183, ***P* = 0.003, ****P* = 0.0005 (one-way Welch's ANOVA).

and duration of binding increased from actin(1–168) to actin(1–305), in agreement with ribosome profiling data (Extended Data Fig. 5f). However, the increase in binding frequency was less pronounced than for PFD, suggesting that actin nascent chains interact preferentially with PFD. Moreover, the TRiC interactions displayed only a single decay component (time constant ($\tau$) ≈ 1.0 s) (Extended Data Fig. 5e).

In summary, PFD recruits TRiC to actin nascent chains. Both chaperones bind dependent on actin chain length, an effect that is most pronounced for PFD. Whereas short probing associations predominate early in translation, binding events with extended dwell times become more frequent with longer nascent chains. The marked increase in PFD engagement near translation termination is likely to stabilize full-length actin and facilitate TRiC recruitment for post-translational completion of folding.

## Post-translational actin folding

Post-translational folding of actin is thought to involve multiple cycles of protein encapsulation in the TRiC cavity[8] and occurs with an overall half-time ($t_{1/2}$) of around 50 s (ref. 42). Actin folding is monitored by DNase I binding, which interacts specifically and with high affinity (dissociation constant ($K_d$) ≈ 2 nM) with a loop region in SD2 of folded actin[43]. The near-full-length actin(1–368) was not recognized by DNase I, demonstrating that the complete actin sequence must be available for folding (Extended Data Fig. 6a).

Analysing the post-translational chaperonin cycles was challenging, as diffusion of monomeric actin is too fast for reliable SPT detection. To slow its diffusion, we tethered actin to the ribosome via a C-terminal SunTag peptide array[44] and a XBP1u+ translation arrest sequence[45,46] (Fig. 2a and Extended Data Fig. 6b). TRiC efficiently folds actin fused to GFP by actin-selective encapsulation, with the linker between actin and GFP protruding through the central oculus of the closed TRiC chamber[47]. The SunTag peptide epitope was visualized using a constitutively expressed GFP-tagged single-chain antibody[44]. As the XBP1u+ sequence retards, but does not prevent, ribosome release, only newly synthesized actin–SunTag is ribosome-associated (Extended Data Fig. 6c). Ribosome-tethered actin–SunTag was folding-competent (Extended Data Fig. 6d,e), and assembled into actin filaments upon ribosome release (Extended Data Fig. 6f). Actin–SunTag pulldown with TRiC–Halo during a chase with puromycin indicated a single exponential rate for passage through TRiC with a $t_{1/2}$ of about 48 s (Extended Data Fig. 6g,h), equivalent to the folding rate of wild-type actin[42]. Note that newly synthesized actin–SunTag–XBP1u+ migrates as a range of bands between around 50 and 80 kDa, containing an incomplete SunTag array (Extended Data Fig. 6g). This ribosome-associated form of actin–SunTag, rather than full-length actin–SunTag (approximately 79 kDa), interacted predominantly with TRiC (Extended Data Fig. 6g). Thus, TRiC-mediated folding initiates immediately when the complete actin sequence is available and proceeds during further SunTag elongation. We therefore define these TRiC interactions as post-translational (post), as opposed to the co-translational (co) interactions with incomplete actin nascent chains (Fig. 1f–k).

Dual-colour imaging revealed dynamic interactions of actin–SunTag with TRiC–Halo and PFD–Halo (Fig. 2b,c, Extended Data Fig. 6i,j and Supplementary Videos 5 and 6). The binding events of TRiC fitted to a two-component decay model with lifetimes of Approximately 0.9 s (44%) and 2.4 s (56%) (Fig. 2d,f), the latter being specific for the post-translational interaction mode (Fig. 2f and Extended Data Fig. 5e). By contrast, the post-translational interactions of PFD were similar to those during translation, with lifetimes of around 0.8 s (70%) and 4.9 s (30%) (Fig. 2d,g). PFD knockdown reduced the frequency of the post-translational TRiC–actin contacts by about 40% (Fig. 2d,e)—that is, to a lesser extent than during translation (Fig. 1g). Notably, PFD knockdown eliminated the longer TRiC binding component (Fig. 2d,h).

Thus, PFD mediates prolonged engagement of TRiC with actin during post-translational folding.

To obtain information on the kinetics of client protein transfer between PFD and TRiC, we used cells that stably expressed TRiC–Halo and PFD–SNAP at endogenous levels (Fig. 2i). Under basal conditions, we observed PFD–TRiC interactions with an average lifetime of about 0.9 s (Fig. 2j). Coincidental collisions with TRiC, analysed using VCP–Halo (VCP is also known as p97 or Cdc48), were also detected, but were less frequent and of shorter duration (Extended Data Fig. 6l–n). Puromycin treatment reduced the frequency of PFD–TRiC contacts by around 25% (Fig. 2k), without change in lifetime (Fig. 2j). This suggests that the majority of PFD–TRiC interactions occur post-translationally.

To selectively enrich for PFD–TRiC interactions during the post-translational folding of actin, we over-expressed actin–SunTag, which accumulates in a folding-competent but ribosome-tethered form. This resulted in a threefold increase in observable PFD–TRiC association events with a lifetime of approximately 0.7 s (Fig. 2j,k and Extended Data Fig. 6k), indicating an extensive functional interplay between PFD and TRiC during post-translational folding, with TRiC–PFD apparently prioritizing the obligate client actin.

These results support a model in which post-translational folding in vivo involves successive interaction cycles of newly synthesized actin with PFD and TRiC. PFD mediates prolonged TRiC interactions with actin lasting around 2.5 s, presumably facilitating productive folding.

## Interactions with folding-defective actin

We next explored whether the chaperonin adapts its functional dynamics to the folding properties of specific client proteins. To address this question, we analysed the interactions of TRiC–PFD with a folding-defective actin mutant. Mutation of the conserved glycine 150 of β-actin, located in the hinge between the actin lobes, to proline (G150P) (Fig. 3a), is thought to interfere with formation of the actin nucleotide binding site[8,48,49], resulting in a 'chaperonin-arrested' state upon in vitro translation[48].

The co-translational interactions of TRiC with nascent chains of actin(G150P) were similar to those with wild-type actin (Fig. 3b,c), consistent with the mutation affecting at late folding step. However, PFD binding was shifted to shorter lifetimes, specifically lacking the prolonged dwell times observed with wild-type actin close to translation termination (Figs. 1j and 3b,c). Thus, the G150P mutation interferes with co-translational conformational changes that enable extended PFD binding. This supports the notion that PFD normally modulates conformational properties of actin nascent chains.

We next analysed how TRiC and PFD interact post-translationally with actin(G150P) expressed as a ribosome-tethered SunTag fusion protein (see Fig. 2a). Notably, the TRiC interaction cycles were around 3 to 4 times longer than for wild-type actin (Fig. 3b), with lifetime components of 3.8 s (78%) and 12.6 s (22%) (Fig. 3c). Thus, release of wild-type actin from TRiC appears to be modulated by a rotation at the G150 hinge, with the G150P mutation resulting in a misfolded state. Misfolded actin(G150P)–SunTag remained soluble (Extended Data Fig. 7a) and was rapidly degraded via the proteasome ($t_{1/2}$ of around 36 min), whereas wild-type actin–SunTag was stable (Extended Data Fig. 7b,c). Remarkably, during a chase with puromycin (in the presence of proteasome inhibitor), TRiC-bound actin(G150P)–SunTag did not appreciably decay from TRiC for at least 10 min (Extended Data Fig. 7d,e). Thus, the mutant protein cycles continuously on TRiC before being transferred to the proteasome for degradation. Note that mainly full-length actin(G150P)–SunTag was associated with TRiC (Extended Data Fig. 7d), as the SunTag was completely translated during prolonged cycling.

The G150P mutation also altered the post-translational interactions with PFD, more than doubling the binding frequency compared with wild-type actin–SunTag (Fig. 3d) and shortening interaction lifetimes

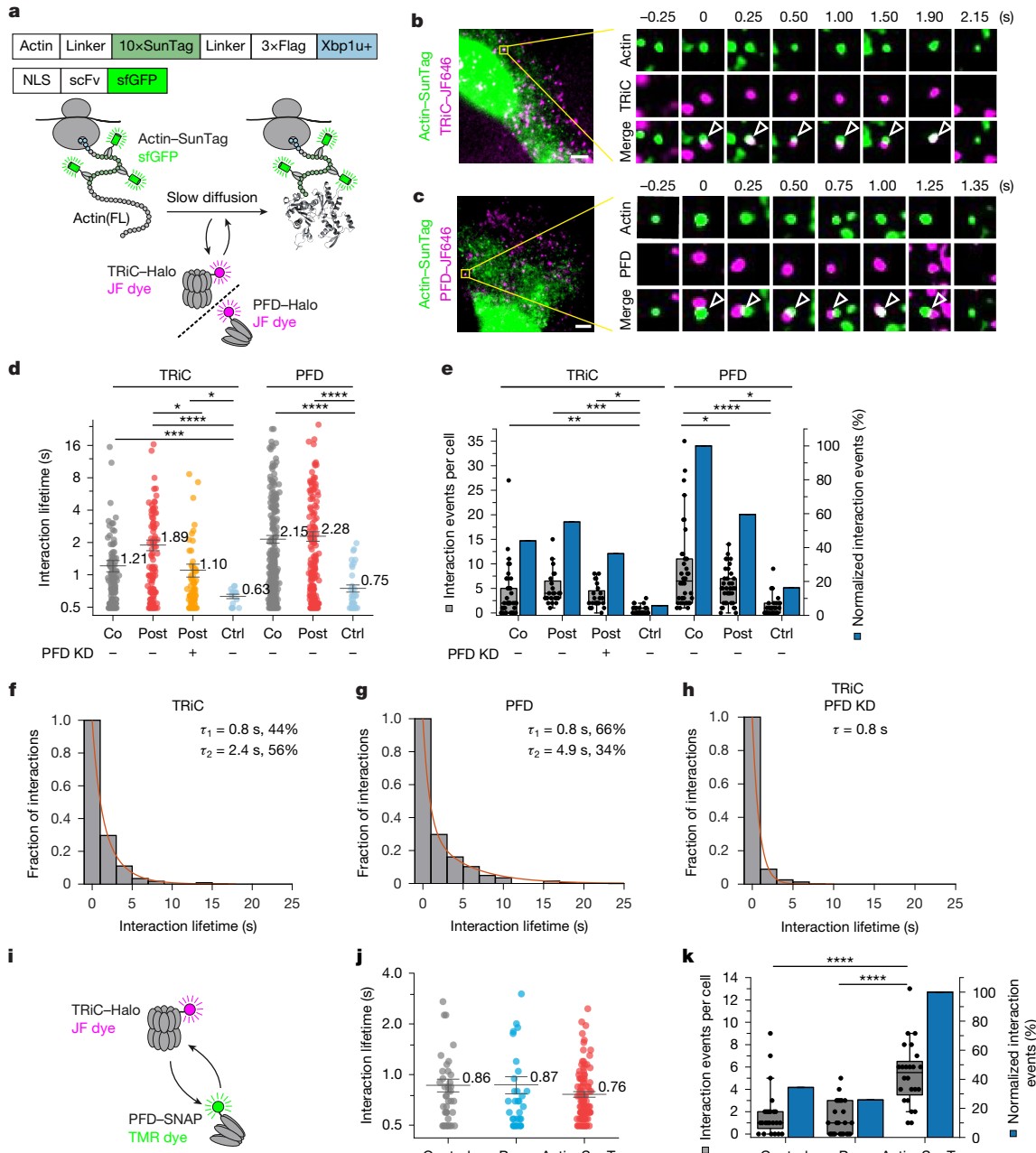

**Fig. 2 | Chaperonin dynamics in post-translational actin folding. a**, Schematic of imaging of TRiC–PFD and translated actin. The 10×SunTag array enables actin imaging via sfGFP-tagged single chain variable fragment (scFv). The Xbp1u+ sequence retains actin in the focal plane. **b,c**, Association of TRiC–actin (**b**) and PFD–actin (**c**). Left, representative dual-colour images. Right, selected time points. Scale bars, 5 µm. **d**, Distribution of co-movement duration of co-translational (Co; from Fig. 1j,k) and post-translational (Post) chaperone–actin interactions. Post-translational interactions of PFD or TRiC with SunTag–Xbp1u+ alone were analysed as control (Ctrl). Number of cells (*n*): TRiC–actin Co, *n* = 38; TRiC–actin Post, *n* = 24; TRiC–actin Post with PFD KD, *n* = 24; TRiC–SunTag, *n* = 29; PFD–actin Co, *n* = 38; PFD–actin Post, *n* = 38; PFD–SunTag, *n* = 31. The centre line is the mean and error bars show s.e.m. TRiC–actin, *\*P* = 0.0169 for PFD KD versus SunTag, *\*P* = 0.0192 for actin Post versus PFD KD, *\*\*\*P* = 0.0008 for actin Co versus SunTag, *\*\*\*\*P* < 0.0001 for actin Post versus SunTag; PFD–actin,

*\*\*\*\*P* < 0.0001 (one-way Welch's ANOVA). **e**, Co-movement events per cell (grey) and normalized co-movement events (blue) of TRiC–PFD–actin as in **d**. TRiC–actin, *\*P* = 0.0328, *\*\*P* = 0.0011, *\*\*\*P* = 0.0001; PFD–actin, *\*P* = 0.0170 for actin Co versus actin Post, *\*P* = 0.0150 for actin Post versus SunTag, *\*\*\*\*P* < 0.0001 (one-way ANOVA). Number of cells as in **d**. **f**–**h**, Survival function (1 − cumulative distribution function (CDF)) of post-translational interaction lifetime for TRiC–actin (**f,h**) and PFD–actin (**g**). **i**, Schematic of imaging of TRiC–Halo and PFD–SNAP. **j**, Distribution of the interaction lifetimes of TRiC–Halo and PFD–SNAP (number of cells (*n*): control, *n* = 24; Puro, *n* = 24, actin–SunTag expression, *n* = 24). The centre line is the mean and error bars show s.e.m. **k**, Co-movement events per cell (grey) and normalized co-movement events (blue) of TRiC–Halo and PFD–SNAP as in **j**. *\*\*\*\*P* < 0.0001 (Welch's ANOVA). Number of cells as in **j**. In **e,k**, the horizontal line indicates the median, boxes delineate top and bottom quartiles and whiskers extend between 10th and 90th percentiles.

(from 2.3 s to 1.5 s on average) (Fig. 3b,c), specifically eliminating the long-lifetime component seen with wild-type actin (Figs. 2g and 3c). Thus, PFD engages the mutant protein more often but more transiently. Conversely, the interactions between PFD and TRiC in the presence of

actin(G150P)–SunTag were about 60% less frequent than for wild-type actin (Fig. 3e,f). Given that PFD exclusively associates with the open conformation of the TRiC ring[21], this implies that TRiC remains predominantly in the closed state when interacting with mutant actin.

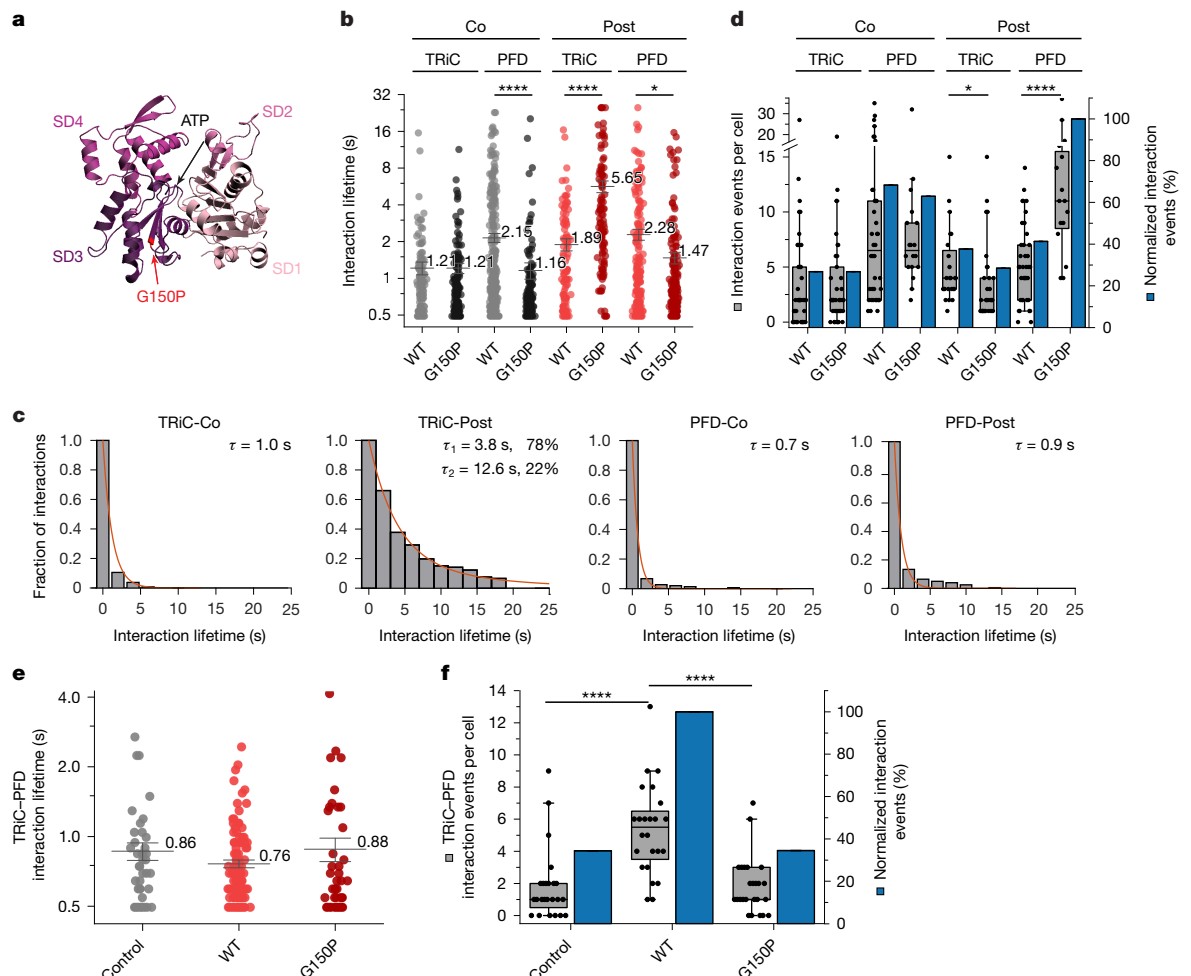

**Fig. 3 | Interactions of TRiC–PFD with folding-defective actin. a**, Structure of actin (PDB visualization of AlphaFold predicted structure: AF_AFP60709F1) with the G150P mutation. **b**, Distribution of co-movement duration for co-translational and post-translational TRiC and PFD interactions with actin(G150P) (number of cells (*n*): Co TRiC–actin(WT), *n* = 38; Co TRiC–actin(G150P), *n* = 38; Co PFD–actin(WT), *n* = 38; Co PFD–actin(G150P), *n* = 18; Post TRiC–actin(WT), *n* = 24; Post TRiC–actin(G150P), *n* = 29; Post PFD–actin(WT), *n* = 38; Post PFD–actin(G150P), *n* = 16). The centre line is the mean and error bars show s.e.m. *P* = 0.0198, ****P* < 0.0001 (one-way Welch's ANOVA). WT, wild type. **c**, Survival function of co-translational and post-translational interaction lifetimes for TRiC and PFD with actin(G150P). **d**, Co-movement events per cell (grey) and normalized co-movement events (blue) of TRiC–actin(G150P) and PFD–actin(G150P) interactions during co-translational and post-translational conditions as in **b**. *P* = 0.0176, ****P* < 0.0001 (Mann–Whitney test). Number of cells as in **b**. **e**, Lifetime distribution for interactions of TRiC–Halo and PFD–SNAP (number of cells (*n*): control, *n* = 24; actin(WT), *n* = 24; actin(G150P), *n* = 25). The centre line is the mean and error bars show s.e.m. **f**, Co-movement events per cell (grey) and normalized co-movement events (blue) of TRiC–Halo and PFD–SNAP interactions under conditions in **e**. ****P* < 0.0001 (one-way ANOVA). Number of cells as indicated in **e**. In **d**,**f**, The horizontal line indicates the median, boxes delineate top and bottom quartiles and whiskers extend between 10th and 90th percentiles.

Accordingly, the frequent binding of PFD to actin(G150P) is less coupled with protein transfer to TRiC and may instead have a role in targeting the mutant protein for proteasomal degradation.

These findings demonstrate that the G150P mutation profoundly alters the interactions of TRiC and PFD with actin. Individual TRiC interaction cycles are markedly prolonged, indicating that the chaperonin 'senses' client protein folding. Mutant actin undergoes futile cycles of binding and release until it is transferred to proteasomes.

## Observing successive chaperonin cycles

We next explored mechanisms that underlie the efficient re-binding of client proteins across successive chaperonin cycles within the crowded cellular environment. TRiC and PFD molecules often remain localized near the client protein after release (off state), thereby avoiding diffusion into the bulk cytosol and allowing re-binding for a subsequent interaction cycle (on state) (Fig. 4a–c, Extended Data Fig. 8a–c and Supplementary Videos 7–10). In the off state TRiC remained confined

for 0.5 to 2 s, exploring regions up to around 1.5 µm (0.7 µm on average) away from the labelled polysome before re-binding (Fig. 4a,d and Extended Data Fig. 8f), which exceeds the average distance of approximately 70 nm between nascent chains and encoding mRNA (Extended Data Fig. 8d,e). On–off–on binding behaviour was observed in 12–18% of all recorded co-translational and post-translational interaction trajectories. However, owing to the limited depth of total internal reflection fluorescence (TIRF) microscopy (less than 500 nm), TRiC may leave the focal plane during the off state and thus re-binding cannot be tracked, thus probably underestimating the re-binding frequency. Indeed, we observed on–off–on cycles for 30–40% of TRiC complexes that could be tracked long enough in the off state to detect re-binding (Extended Data Fig. 8g). A specific PFD molecule may even undergo three consecutive binding cycles with actin nascent chains translated from the same mRNA (possibly engaging adjacent nascent chains in polyribosomes) before diffusing away (Extended Data Fig. 8b). Notably, in the off state, the majority (70–80%) of observable TRiC complexes were stationary, with the remainder showing sub-diffusive behaviour

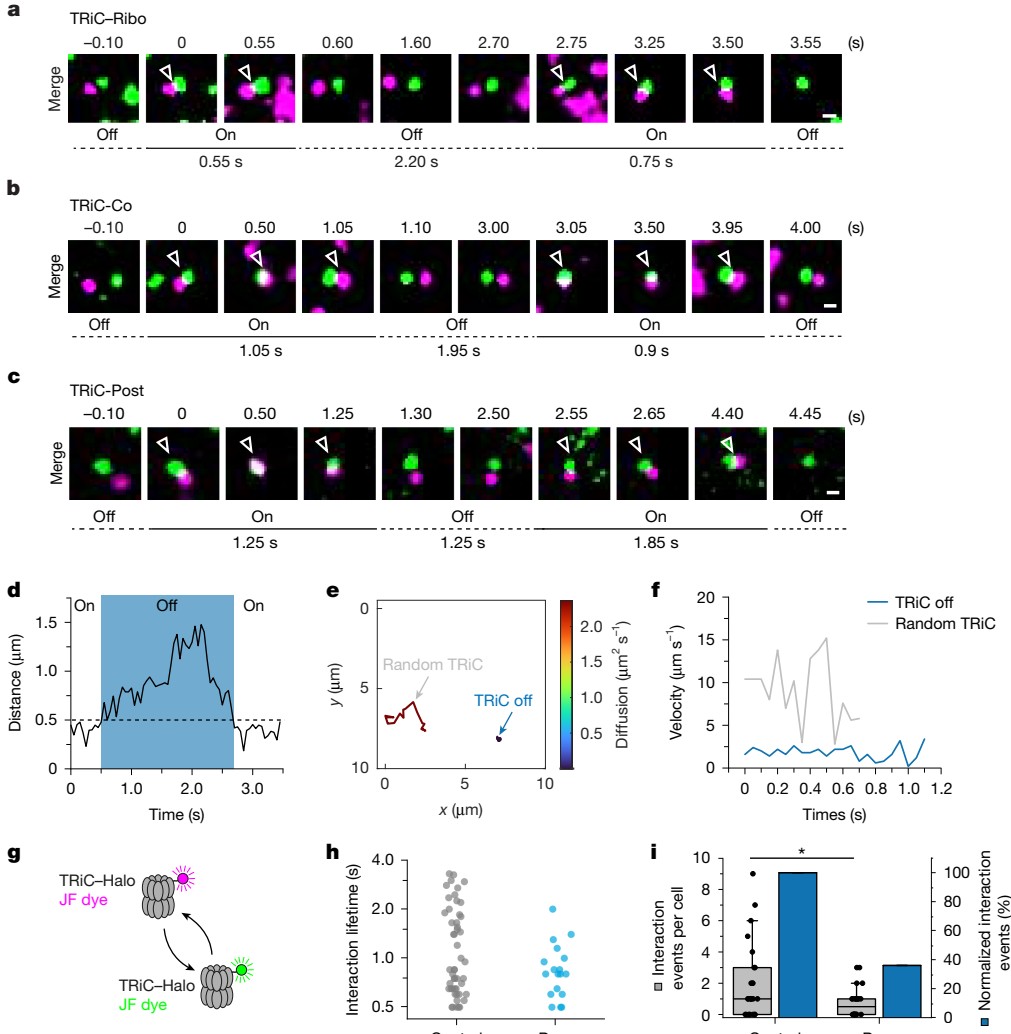

**Fig. 4 | Imaging of successive chaperonin cycles. a–c**, Dual-colour imaging of successive binding of TRiC with translating ribosomes (Ribo) (**a**), translating actin nascent peptide chain (Co) (**b**) and translated actin (Post) (**c**). On–off-on events are summarized in Extended Data Fig. 8g. Scale bars, 500 nm. **d**, Distance between TRiC and translating ribosome (polysome) as in **a** during on and off states. The dotted line indicates the threshold used to define interactions, which requires tracking for at least 10 frames (500 ms in total), with a distance below 0.5 µm to be considered as 'on'. **e**, Diffusion map of TRiC molecule in the off state (TRiC off, blue) as in **c**, compared with randomly sampled TRiC molecules in proximity (random TRiC, grey). Trajectories are colour-coded by diffusion coefficient. **f**, Velocity traces over time for the tracks in **e**. **g**, Schematic of imaging of TRiC–TRiC interactions. **h**, Distribution of the interaction lifetime of different TRiC–Halo molecules (number of cells (*n*): control, *n* = 24; Puro, *n* = 24). **i**, Co-movement events per cell (grey) and normalized co-movement events (blue) of different TRiC–TRiC interactions under the conditions in **h**. The horizontal line indicates the median, boxes delineate top and bottom quartiles and whiskers extend between 10th and 90th percentiles. *\*P* = 0.0402 (Mann–Whitney test). Number of cells as in **h**.

(Fig. 4c,e,f and Extended Data Fig. 8h), suggesting that the chaperonin is retained through interactions such as low-affinity associations with the translation machinery and/or other chaperones.

A recent study using cryo-electron tomography reported that a fraction of TRiC complexes in human cells form linear and circular clusters of between two and seven molecules[50]. Such clustering may have an auxiliary role in concentrating TRiC molecules near a client protein. To begin exploring this possibility, we analysed TRiC–TRiC interactions using sparse labelling of TRiC–Halo with JF549 and JF646 dyes for dual-colour SPT (Fig. 4g and Extended Data Fig. 8i). We observed two classes of TRiC–TRiC contacts with average lifetimes of around 0.7 s and 2 s (Fig. 4h and Extended Data Fig. 8j), similar to the interactions of TRiC with actin (Fig. 2f). Inhibition of translation with puromycin diminished the long-lived associations (Fig. 4h,i and Extended Data Fig. 8j), suggesting that they are functionally related to protein biogenesis. Client proteins may thus be channelled between TRiC molecules.

## Discussion

Here we established a set of versatile methods to explore co- and post-translational chaperone functions by real-time SPT in intact cells. Our analysis provides insight into the functional dynamics of the PFD/TRiC chaperonin system in vivo. We show that both chaperones interact co-translationally with a range of nascent chains in multiple binding events typically lasting for around 1 s, with PFD recruiting TRiC (Fig. 1a–e). Using the obligate chaperonin client actin[8,21], these interactions displayed pronounced nascent chain length dependence (Fig. 1f–k). Whereas brief probing contacts predominate early in translation, both frequency and duration of PFD and TRiC binding increase significantly once 80% of the actin sequence is exposed on the ribosome (Fig. 5a), with PFD–actin complexes persisting for several seconds near translation termination. Thus, full-length actin is poised to be released from the ribosome in complex with PFD, ensuring stabilization in a conformational state competent for transfer to TRiC and post-translational folding.

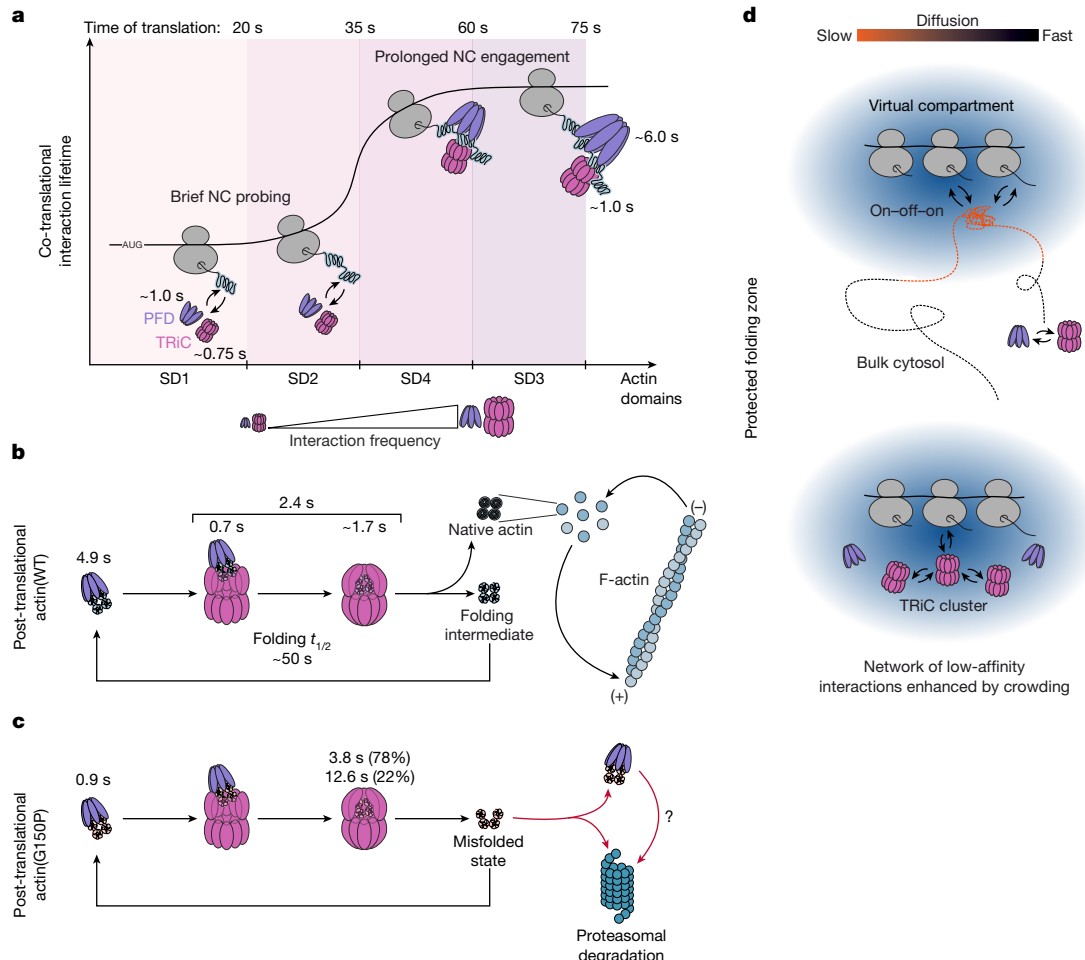

**Fig. 5 | Dynamics of TRiC–PFD-mediated protein folding in vivo. a**, Model of co-translational folding of the nascent actin chain assisted by TRiC–PFD. As SD1 and SD2 emerge from the ribosome, TRiC and PFD transiently engage the nascent chain (NC) with brief probing interactions, with PFD recruiting TRiC. When SD4 and SD3 become exposed, chaperone interactions increase in frequency and duration, with PFD showing markedly prolonged binding close to translation termination. **b**, Interaction dynamics between TRiC–PFD and actin during post-translational folding. PFD binding to near-full-length actin chains ensures efficient TRiC recruitment for post-translational folding involving reaction cycles with a lifetime of about 2.4 s, consisting of a ternary PFD–TRiC–actin complex ($\tau \approx 0.7$ s) and a binary TRiC–actin complex ($\tau \approx 1.7$ s).

Additionally, PFD functions in actin retrieval to TRiC ($\tau \approx 4.9$ s) for successive TRiC interaction cycles. **c**, Post-translational interaction between TRiC–PFD and folding incompetent actin(G150P). Compared to wild-type actin (**b**), the interaction of PFD with actin(G150P) is substantially shorter in duration (0.9 s), and the interaction of TRiC shows two prolonged interaction lifetimes: 3.8 s (78%) and 12.6 s (22%). Actin(G150P) cycles on TRiC until it is degraded by the proteasome. **d**, Proposed model for a 'protected folding zone' surrounding translation-active ribosomes as a virtual compartment formed by a network of low-affinity interactions and enhanced by local macromolecular crowding. TRiC clusters[50] may support client channelling.

To acquire its native tertiary fold, the complete actin protein must undergo encapsulation in the TRiC cavity (Fig. 5b). We found the post-translational TRiC–actin interactions to be kinetically bimodal, with approximately 60% of actin molecules displaying a residence time of around 2.4 s and the remainder having a shorter lifetime of around 0.9 s (Fig. 2a–h). The long-lifetime component was eliminated upon PFD dysfunction, suggesting that the functional interplay with PFD facilitates productive folding by TRiC (Fig. 5b). This interpretation is supported by structural data showing that PFD associates with a pre-formed TRiC–actin complex (with TRiC in the open state), seemingly 'pushing' the substrate protein into the TRiC cavity[5,21]. The lifetime of the ternary TRiC–actin–PFD complex (with TRiC in the open state) was approximately 0.7 s (Fig. 2j), thus leaving 1.7 s for actin encapsulation (Fig. 5b). The shorter actin–TRiC contacts, observed approximately 40% of the time, may represent proofreading of already folded actin or binding events with failed encapsulation. The post-translational PFD–actin interactions were also bimodal, with average lifetimes of about 0.8 s (66%) and 4.9 s (34%), with the former probably corresponding

to the role of PFD in modulating actin folding on TRiC and the latter to the function of PFD in retrieving actin for TRiC re-binding (Fig. 5b). Given that overall actin folding takes around 50 s (ref. 42), individual actin molecules generally require multiple folding attempts. In each round the actin chain folds to the native (or a near-native) state with a biologically relevant probability.

This proposed mechanism for TRiC-assisted folding is supported by our analysis of the folding-defective actin(G150P) mutant (Fig. 3a–f). The dwell time of actin(G150P) on TRiC is three to four times longer than for wild-type actin (Fig. 5c). Since the frequency of PFD–TRiC interactions was reduced correspondingly, the mutant protein appears to spend more time in the TRiC-encapsulated state, which is inaccessible to PFD. Thus, there is an unexpected interdependence between client conformational properties and encapsulation time by TRiC. Actin(G150P) is thought to be impaired in a late folding step involving a relative rotation of the major domain lobes[48]. Contacts of the enclosed protein with the cavity wall[6,51] may regulate the TRiC ATPase to trigger opening of the folding chamber, allowing TRiC to sense the client protein conformation.

Misfolded actin(G150P) retains high affinity for TRiC until transferred to the proteasome. Mutations in abundant client proteins like actin might therefore impair proteostasis by occupying available TRiC capacity.

TRiC-mediated folding hinges on the ability of not-yet folded client proteins to return to chaperonin for repeated folding attempts—a remarkably efficient process, as indicated by the continuous cycling of mutant actin. We directly observed successive chaperonin cycles, revealing the surprising phenomenon that between binding events, TRiC and PFD may be retained in close proximity to client protein for up to around 2 s, avoiding diffusion into the bulk cytosol (Figs. 4a–f and 5d). This finding suggests a supramolecular organization of the chaperonin system, perhaps most prevalent at local translation hotspots[27], that generates a 'virtual folding compartment' through low-affinity interactions with translation machinery and/or other chaperone factors, effective under conditions of local macromolecular crowding[25,52,53]. Within this environment, PFD–TRiC and TRiC–TRiC interactions may support client protein channelling for efficient folding (Fig. 5d). Identifying the mechanisms that underlie functional compartmentalization of the chaperone machinery will be important for understanding how protein folding fidelity is maintained in the crowded cellular environment.

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

## Methods

### Plasmids

To visualize co-translational interactions, plasmids encoding human β-actin, a 3×Flag tag, and 24 MS2 stem-loop repeats were constructed. The β-actin-24×MS2 (MBSV5) sequence[54] (Addgene #102718) was amplified and subcloned into pRetroQ-AcGFP-C1 (TaKaRa) via PCR amplification, replacing AcGFP1. Additionally, truncated versions, containing the first 20, 101, 168, 305 and 368 amino acids of β-actin, were generated by amplifying the respective fragment lengths and introducing a stop codon. The plasmid pUbC-NLS-ha-stdMCP-stdGFP[55], which expresses the MS2 coat protein fused to GFP (MCP–GFP), was used without modification (Addgene #98916). For imaging post-translational interactions, plasmids encoding β-actin, a 10×SunTag, a 3×Flag tag, and Xbp1u+ were designed. A synthetic β-actin gene (Uniprot P60709) was ordered from Integrated DNA Technologies (IDT) and cloned into pHRdSV40-K560-24×GCN4_v4[44] (Addgene #72229) by replacing the kinesin-1 motor domain via PCR amplification. The 24×SunTag repeats were then reduced to 10, and the Xbp1u+ sequence[45] was introduced using PCR. To generate a plasmid encoding β-actin with the G150P point mutation fused to 10×SunTag, 3×Flag tag and Xbp1u+ the Q5 Site-Directed Mutagenesis Kit (NEB) was used according to the manufacturer's instructions. To estimate non-specific interaction of chaperone and SunTag alone, a plasmid encoding the 10×SunTag, 3×Flag tag and XBP1u+ sequence without β-actin was constructed by PCR amplification to excise the β-actin gene from the previously generated plasmid. The plasmid pHR-scFv-GCN4-sfGFP-GB1-NLS-dWPRE, which expresses an scFv against the SunTag epitope fused to sfGFP (scFv-sfGFP), was used without modification[44] (Addgene #60906). To estimate the distance between mRNA and translating polypeptides, 24×MoonTag[56] epitope was placed at the N-terminal end of β-actin-24×MS2 plasmid. Plasmids encoding MoonTag specific nanobody 2H10 were generously provided by M. Tanenbaum. To estimate coincidental collisions with TRiC, we analysed interactions between TRiC and the 500-kDa VCP (P97) complex, which has diffusion properties similar to PFD (Extended Data Fig. 6l). A plasmid encoding VCP with C-terminal HaloTag was provided by Z. Liu.

A plasmid expressing CCT4 with SNAP-tag was generated by inserting SNAP-tag in a loop region between S202 and V203, obtained from Integrated DNA Technologies and cloned into pCW57.1-MAT2A (Addgene #100521).

### Cell lines

Human U-2 OS cells (ATCC, HTB-96) were grown in complete media with DMEM containing 4.5 g l$^{-1}$ D-glucose (Gibco), 10% fetal bovine serum (Gibco), 1% penicillin/streptomycin (Gibco) and 1% non-essential amino acids (Gibco) at 37 °C and 5% $CO_2$. Cells were either transfected with Lipofectamine 3000 (Thermo Fisher) or electroporated (Neon transfection system, Invitrogen) following the manufacturer's instructions. Cells were regularly tested for mycoplasma contamination and no contamination was detected.

The monoclonal U-2 OS cell line expressing CCT4–HaloTag was generated using CRISPR–Cas9-mediated gene integration. A CCT4 specific single guide RNA (sgRNA) (ATCTCTAAGATCTACACTGG) was cloned into the pSpCas9(BB)-2A-Puro (PX459) vector, which encodes SpCas9 and puromycin resistance[57] (Addgene #48139). U-2 OS cells were transfected with the Cas9–sgRNA vector and a donor encoding HaloTag using Lipofectamine 3000 following the manufacturer's instructions. Forty-eight hours after transfection, cells were selected with 1.5 μg ml$^{-1}$ puromycin (Gibco). Surviving cells were further labelled with a fluorescent HaloTag ligand (Promega) and sorted via fluorescence-activated cell sorting to obtain single clones. The same strategy was applied to generate PFD4–Halo and PFD4–SNAP stable cell lines using PFD4 specific sgRNA (TTCAGTTGTATGCAAAATTC).

The polyclonal U-2 OS cell line expressing RPL10A–SNAP-tag was generated using lentivirus transduction. Lentiviruses were produced

and transfected as described[58]. Forty-eight hours after transfection, cells were selected with puromycin (Gibco) and sorted with a fluorescent SNAP-tag ligand (NEB).

The PFD3 knockdown stable cell line was generated using a PFD3 specific sgRNA (TAAAGTGTGTCTGTGGTTGG) using CRISPR–Cas9-mediated gene deletion. U-2 OS cells were transfected with the Cas9–sgRNA vector using Lipofectamine 3000 and selected with puromycin after 48 h transfection. Cells were further sorted to isolate single clones.

### Biochemical assays

**SDS–PAGE and immunoblotting.** Protein samples were boiled in 2× SDS Sample Buffer (Sigma-Aldrich) at 95 °C for 5 min. Equal amounts of total protein were loaded on SDS polyacrylamide gels. Proteins were separated by electrophoresis on NuPAGE 4–12% Bis-Tris Protein Gels (Thermo Fisher Scientific) using NuPAGE MOPS SDS Running Buffer (Thermo Fisher Scientific) at 120 V. Proteins were transferred from polyacrylamide gels to polyvinylidene difluoride (PVDF) membranes (Roche) in transfer buffer (25 mM Tris-HCl pH 7.5, 190 mM glycine, 0.1% SDS, 20% methanol) at a constant voltage of 100 V for 1 h. Membranes were blocked in blocking buffer (10 mM Tris-HCl pH 7.5, 150 mM NaCl, 0.05% Tween-20, 5% low-fat milk) for 1 h at room temperature. Membranes were incubated with primary antibody in blocking buffer for 1 h at room temperature or overnight at 4 °C. Immunodetection was performed using: anti-β-actin (Abcam, ab8226, 1/1,000 dilution), anti-CCT4 (Merck, HPA029349, 1/1,000 dilution), anti-DYKDDDDK Tag (Cell Signal, 2368S, 1/1,000 dilution), anti-GAPDH (Merck, MAB374, 1/1,000 dilution), anti-HaloTag (Promega, G9211, 1/1,000 dilution), anti-prefoldin 3 (Santa Cruz, sc-390524, 1/200 dilution), anti-prefoldin 4 (Thermo Fisher, 16045-1-AP, 1/300 dilution), anti-RPL10A (Abcam, ab174318, 1/1,000 dilution) anti-RPL29 (Thermo Fisher, PA5-27545, 1/1,000 dilution), anti-GCN4 (SunTag) (Addgene, 218104-rAb, 1/1,000 dilution) and anti-SNAP-tag (NEB, P9310S, 1/1,000 dilution). Blots were then washed 3 times for 10 min with blocking buffer without low-fat milk at room temperature and incubated with secondary antibody, conjugated anti-mouse immunoglobulin G (IgG)–horseradish peroxidase (HRP) (Merck, A4416, 1/10,000 dilution) or anti rabbit IgG–HRP (Merck, A9169, 1/10,000 dilution), in blocking buffer for 1 h at room temperature. Blots were washed 3 times for 10 min and developed on Amersham Image Quant 800 control software 2.1.0.3. Images were analysed and quantified in Fiji.

**HaloTag pulldown.** U-2 OS cells were lysed in standard lysis buffer (50 mM Tris-HCl pH 7.5, 150 mM NaCl, 5 mM $MgCl_2$, 0.5% IGEPAL CA-630, 20 U ml$^{-1}$ apyrase, 1 mM PMSF, 2 mM DTT, cOmplete Protease Inhibitor, Benzonase) on an end-over-end rotor for 30 min at 4 °C. The lysate was cleared by centrifugation at 16,000$g$ for 20 min at 4 °C. Protein concentration was determined by Bradford assay. Halo-Trap Magnetic Agarose beads (Chromotek) were equilibrated in ice cold lysis buffer and subsequently washed 2 times with ice cold washing buffer (50 mM Tris-HCl pH 7.5, 100 mM NaCl, 2 mM $MgCl_2$, cOmplete Protease Inhibitor) containing 5% glycerol and 2 times in washing buffer without glycerol. Cleared lysate was added to equilibrated Halo-Trap beads and incubated for 3 h at 4 °C with vertical rotation. The beads were washed three times in washing buffer containing 5% glycerol and three times in washing buffer without glycerol. For immunoblotting the bound protein was eluted in 2× SDS Sample Buffer by boiling at 95 °C for 5 min. For mass spectrometry the bound protein was eluted and digested with Elution Buffer I (50 mM Tris-HCl pH 7.5, 2 M urea, 5 μg ml$^{-1}$ trypsin, 1 mM TCEP-HCl) for 30 min at 400 rpm at 30 °C. The sample was then combined with Elution Buffer II (50 mM Tris-HCl pH 7.5, 2 M Urea, 5 mM iodoacetamide) and incubated overnight at 400 rpm at 32 °C. The reaction was stopped by adding 25% trifluoroacetic acid (TFA, Merk).

**Kinetics of actin transit to TRiC reflecting TRiC-mediated folding.** U-2 OS cells were electroporated with plasmid encoding either

actin–SunTag–Xbp1u+ or actin(G150P)–SunTag–Xbp1u+. For cells expressing actin(G150P)–SunTag–Xbp1u+, 10 μM MG-132 was added 4 h prior to the puromycin chase. Twenty-four hours after transfection, cells were pre-treated with 300 μg ml$^{-1}$ puromycin (Gibco) for 2.7 min. The time 0 ($t = 0$) sample was collected immediately after pre-treatment. The chase was performed at time points of 0.3 min, 1.3 min, 2.3 min, 3.3 min, 5.3 min and 9.3 min. Cells were washed two times with ice cold PBS supplemented with 100 μg ml$^{-1}$ cycloheximide (CHX) and frozen in a dry ice 2-propanol bath. Cells were lysed with 500 μl standard lysis buffer. The lysate was cleared by centrifugation at 14,000$g$ for 20 min at 4 °C. Protein concentration was determined by Bradford assay. Anti-HaloTag pull down and sample preparation for immunoblot were performed as mentioned above. Immunoblots were analysed and quantified using Fiji. The average intensity from three replicates was plotted. Folding half-time was determined by fitting the data in SigmaPlot 14.0 using an exponential decay function: $f = y_0 + a \times e^{-b \times x}$.

**Cycloheximide chase.** U-2 OS cells were electroporated with a plasmid encoding either actin–SunTag–Xbp1u+ or actin(G150P)–SunTag–Xbp1u+. Twenty-four hours after electroporation, cells were treated with 150 μg ml$^{-1}$ CHX (Sigma-Aldrich) for 0.5 h, 0.75 h, 1 h and 2 h. The time 0 (T0) sample remained untreated. Cells were collected and subsequently lysed in RIPA buffer (Thermo Fisher Scientific) supplemented with cOmplete Protease Inhibitor and Benzonase. The lysate was cleared by centrifugation at 14,000$g$ for 20 min at 4 °C. Protein concentration was determined by Bradford assay. For further immunoblot analysis, proteins were denatured in 2× SDS Sample Buffer by boiling at 95 °C for 5 min. Immunoblots were analysed and quantified using Fiji. Degradation half-time was determined by fitting the data in SigmaPlot 14.0 using an exponential decay function mentioned above.

**Preparation of DNase I Sepharose beads.** Cyanogen bromide-activated Sepharose 4B beads (1 g; Sigma-Aldrich) were washed up to 10 times with 1 mM HCl. 100 mg DNase I (Roche) was dissolved in 0.1 M NaHCO$_3$ and 0.5 mM CaCl$_2$. DNase I solution was added to washed Cyanogen bromide-activated Sepharose 4B beads and incubated at 4 °C with vertical rotation overnight. Unreacted groups in the resin were blocked upon incubation with 0.2 M ethanolamine pH 8.0 and 0.5 mM CaCl$_2$ for 2 h at 4 °C. DNase I conjugated Sepharose 4B beads were washed with two alternating cycles of 0.1 M sodium acetate pH 4.5 and 0.1 M ethanolamine pH 8.0. Beads were stored in 10 mM Tris-HCl pH 7.4, 1 mM CaCl$_2$, 10% glycerol, 1 mM DTT, 0.02% NaN$_3$, cOmplete Protease Inhibitor at 4 °C.

**DNase I pulldown.** U-2 OS cells were lysed in lysis buffer (10 mM Tris-HCl pH 7.4, 1 mM CaCl$_2$, 1 mM DTT, 0.2 mM ATP, 0.5% Triton X-100, 1 mM PMSF, cOmplete Protease Inhibitor, Benzonase) on an end-over-end rotor for 30 min at 4 °C. Lysate was cleared by centrifugation at 16,000$g$ for 20 min at 4 °C. Protein concentration was determined by Bradford assay. DNase I conjugated Sepharose 4B beads were equilibrated by washing 3 times with DNase I binding buffer (10 mM Tris-HCl pH 7.4, 1 mM CaCl$_2$, 10% glycerol, 1 mM DTT, 0.2 mM ATP, cOmplete Protease Inhibitor). Beads were centrifuged at 500$g$ for 2 min at 4 °C and supernatant was discarded. Equilibrated DNase I conjugated Sepharose 4B beads were added to cleared lysate and incubated for 3 h at 4 °C with vertical rotation. Beads were washed once with DNase I washing buffer (10 mM Tris-HCl pH 7.4, 1 mM CaCl$_2$, 10% glycerol, 1 mM DTT, cOmplete Protease Inhibitor), once with DNase I washing buffer supplemented with 0.3 M NaCl, and twice with DNase I washing buffer. For immunoblotting the bound protein was eluted in 2× SDS Sample Buffer by boiling at 95 °C for 5 min.

**Ribosome isolation.** U-2 OS cells were lysed in standard lysis buffer on an end-over-end rotor for 30 min at 4 °C. Lysate was cleared by centrifugation at 15,000$g$ for 15 min at 4 °C to remove large debris. Pre-cleared lysate was further centrifuged at 330,000$g$ for 1 h at 4 °C in a SW 55 Ti rotor (Beckman Coulter) to pellet ribosomes. The supernatant and pellet fractions were used for immunoblot analysis.

**Polysome gradient analysis.** Cells were treated with 100 μg ml$^{-1}$ CHX and subsequently lysed in lysis buffer (5 mM Tris-HCl pH 7.4, 1.5 mM NaCl, 2.5 mM MgCl$_2$, 1 mM DTT, 0.5% Triton X-100, 0.5% sodium deoxylate, 100 μg ml$^{-1}$ CHX, 100 U ml$^{-1}$ SUPERase*In, cOmplete Protease Inhibitor, Benzonase). The RNA concentration of the pre-cleared lysate was determined by measuring absorbance at 260 nm, and a total RNA amount of 300 μg was loaded onto the gradients. Sucrose density gradients (10%–45%) were prepared in SW41 ultracentrifuge tubes (Steton) using a BioComp Gradient Master (BioComp Instruments) according to the manufacturer's instructions. The individual 10% and 45% sucrose solutions were prepared in polysome buffer (25 mM Tris-HCl pH 7.4, 150 mM NaCl, 15 mM MgCl$_2$, 1 mM DTT, 100 μg ml$^{-1}$ CHX, cOmplete Protease Inhibitor). The gradients were centrifuged for 2.5 h at 40,000 rpm in a SW 41 Ti rotor (Beckman Coulter) at 4 °C. Polysome profiles were obtained using the Triax Flow Cell Firmware v.2.30.4.211. The gradients were fractionated using a piston gradient fractionator coupled to an A254 nm spectrophotometer (Biocomp). Polysome fractions were precipitated with 10% trichloroacetate and protein precipitates were washed with ice cold acetone. For immunoblotting the proteins were denatured in 2× SDS Sample Buffer by boiling at 95 °C for 5 min.

**Solubility assay.** U-2 OS cells were electroporated with a plasmid encoding either actin–SunTag–Xbp1u+ or actin(G150P)–SunTag–Xbp1u+. Twenty-four hours after electroporation cells were washed twice with ice cold PBS and lysed in RIPA buffer supplemented with cOmplete Protease Inhibitor and Benzonase. Protein concentration was determined by Bradford assay. The sample was divided into a total fraction and a fraction subjected to centrifugation at 20,000$g$ for 20 min at 4 °C (pellet). The supernatant was transferred to a fresh tube, while the pellet was washed with RIPA buffer supplemented with cOmplete Protease Inhibitor and Benzonase, then centrifuged again under the same conditions. The total and supernatant fractions were denatured in 2× SDS Sample Buffer, while the pellet fraction was denatured in 1× SDS Sample Buffer by boiling at 95 °C for 5 min. Fractions were analysed by immunoblotting.

### Single-molecule tracking in live cells

**Fluorescence microscopy.** Two-colour simultaneous imaging was performed on a Zeiss Elyra PS.1 inverted super-resolution microscope equipped with two Andor iXonEM+ DU-897D EM-CCD cameras and a custom-built image splitter with motorized $xyz$–rotation adjustment. Camera alignment was automatically performed with a built-in alignment pattern. All images were acquired using ZEN black 2.1 SP3 software using Zeiss alpha Plan-Apochromat 100×/1,46 Oil DIC oil immersion objective in TIRF (total internal reflection fluorescence) mode. The TIRF angle was set between 63° and 65°. The two fluorescence channels were acquired in parallel with the two cameras. Each time series consisted of 500 frames with 50 ms exposure time. During imaging, cells were incubated in a Tokai Hit stage-top incubator at 37 °C and 5% CO$_2$.

**Sparse labelling.** Cells used for imaging were cultured in a glass-bottom dish (Ibidi, μ-Dish 35 mm, high glass bottom). Before labelling, cells were washed with Hanks' Balanced Salt Solution (HBSS, Gibco, 14025092) and incubated with fresh culture media. Janelia Fluor (JF) HaloTag ligands (Promega) and SNAP-tag ligand TF-TMR dye (provided by K. Johnsson)[34] were dissolved in DMSO to prepare stock solutions. Fluorescent ligands were added to the cells and incubated for 20–30 min. Excess dye was washed out by rinsing the cells three times with HBSS. Cells were incubated with fresh media for at least 5 min and washed again before imaging.

**SPT and colocalization analysis.** Tracking of individual particles was performed in Fiji[59] with the plugin TrackMate[60]. Particles in each frame were detected using the Laplacian of Gaussian detector. Multiple trajectories were then determined from the detected particles using the simple linear assignment problem (LAP) tracker. The linking max distance, gap-closing max distance, and gap-closing max frame gap parameters were set to 0.5 µm, 0.5 µm and 0 frames, respectively. Trajectories detected in the two channels were analysed to identify colocalization events using KNIME as described[61]. Pairwise colocalization analysis was performed by measuring the distances from each molecule in the first channel to all the molecules in the second channel. Molecule pairs were classified as co-localization if their distance was less than 500 nm. Tracks with co-localized molecules that persisted for more than 500 ms were used to analyse the interaction lifetime between chaperones and substrates. A 500 ms threshold was applied to exclude short-lived contacts, which are dominated by non-specific encounters and obscure the detection of condition-specific differences, thereby enriching for biologically meaningful interactions. Mean square displacement was used to determine the diffusion coefficient of each track.

**Statistical analysis of lifetime and interaction events.** Welch's ANOVA test was used for statistical comparison of measured lifetimes across more than two groups. This test is suitable for datasets that are not normally distributed and have unequal sample sizes, provided that each group contains more than 15 observations[62,63]. For comparison of two groups, the non-parametric Mann–Whitney $U$-test was applied. When comparing interaction events across more than two groups, a one-way ANOVA followed by Tukey's post hoc test was used if a significant overall difference was detected (hereafter referred to as ANOVA).

**Kinetic modelling of chaperone–substrate interactions.** The interaction lifetime between chaperone and substrates was calculated from the duration of their co-movement events. To resolve distinct kinetic components, we fitted the 1 − CDF of the interaction lifetimes using a constrained nonlinear least-squares fit. Two kinetic models were evaluated: a single-component model ($F(t) = e^{-t/\tau}$) and a two-component model ($F(t) = \alpha e^{-t/\tau_1} + (1-\alpha)e^{-t/\tau_2}$). In the two-component model, $\tau_1$ and $\tau_2$ represent the lifetime of each component, and $\alpha$ and $1 - \alpha$ are their respective fractions. The two-component model was only accepted if it showed both higher coefficient of determination ($R^2$) than the single-component model and a statistically significant second population ($\alpha > 5\%$).

**Fraction of fluorescently labelled molecules.** To estimate the percentage of labelled Halo-tagged protein at certain concentrations of fluorescent ligand, cells were incubated with Halo ligand JF646 dye (Promega) at the following concentrations: 100 nM, 150 nM, 250 nM and 500 nM for cells expressing TRiC–Halo and 2 nM, 10 nM, 100 nM and 250 nM for cells expressing PFD–Halo. Images were acquired using a fluorescence microscope. Average fluorescence intensity was quantified at different dye concentrations. The highest dye concentrations corresponding to the plateau of normalized fluorescence intensity (250 nM for PFD–Halo and 500 nM for TRiC–Halo), as well as concentrations 2-fold higher (500 nM for PFD–Halo and 1 µM for TRiC–Halo), were used for in-gel fluorescence analysis. These samples were compared to those labelled with the dye concentrations used for SPT (800 pM for PFD–Halo and 50 pM for TRiC–Halo). Cells treated in same way were lysed in RIPA buffer supplemented with cOmplete Protease Inhibitor and Benzonase, and proteins were separated by SDS–PAGE as previously described. To minimize quantification bias, fivefold less total protein was loaded for lysates from samples labelled at saturated dye concentrations compared to those used for SPT. Fluorescence signals were detected using an Amersham Typhoon Biomolecular Imager and quantified with Fiji. Coomassie staining was performed as a loading control (Extended Data Fig. 1k).

## Mass spectrometry

**Sample preparation for total proteome analysis.** Cell pellets were resuspended in 400 µl of SDC buffer containing 1% sodium deoxycholate (SDC; Sigma-Aldrich), 40 mM 2-chloroacetamide (Sigma-Aldrich), 10 mM tris(2-carboxyethyl)phosphine (TCEP, Thermo Fisher Scientific), and 100 mM Tris, pH 8.0. After incubation for 5 min at 95 °C, the samples were ultrasonicated for 10 min using 10 cycles of 30 s at high intensity with a 30 s pause between cycles (Bioruptor, Diagenode). The incubation and ultrasonication steps were repeated once more. The samples were then diluted 1:1 with MS-grade water (VWR), and proteins were digested with 1 µg Lys-C (Wako) for 4 h at 37 °C, followed by an overnight digestion at 37 °C with 2 µg trypsin (Promega). The resulting peptide solution was acidified with TFA to a final concentration of 1%, and then desalted using SCX-stage tips.

**Liquid chromatography–mass spectrometry data acquisition.** Liquid chromatography–mass spectrometry analysis was performed using an Easy-nLC 1200 (Thermo Fisher Scientific) nanoflow system coupled with a QExactive HF mass spectrometer (Thermo Fisher Scientific). Chromatographic separation was achieved on a 30-cm column (inner diameter: 75 µm; packed in-house with ReproSil-Pur C18-AQ 1.9 µm beads, Dr. Maisch GmbH). Peptides were injected onto the column in buffer A (0.1% (v/v) formic acid), with the column heated to 60 °C. Peptides were eluted at a flow rate of 250 nl min⁻¹ using a gradient from 2% to 30% buffer B (80% acetonitrile, 0.1% formic acid) over 120 min (QExactive HF), followed by a ramp to 60% over 10 min, then to 95% over the next 5 min, which was maintained for another 5 min to measure the total proteome. For immunoprecipitation measurements, peptides were eluted using a gradient from 7% to 30% buffer B over 60 min, followed by an increase to 60% over 15 min, then to 95% over the next 5 min, and maintained at 95% for an additional 5 min. The QExactive HF mass spectrometer was operated in data-dependent mode, with survey scans acquired over the $m/z$ range of 300–1,650 at a resolution of 60,000 at $m/z = 200$. Up to 10 of the top precursors were selected and fragmented using higher-energy collisional dissociation (HCD) with a normalized collision energy of 30. The AGC target for MS and MS2 scans were set to 3E6 and 1E5, respectively, with maximum injection times of 100 ms for MS and 60 ms for MS2. Dynamic exclusion was set to 30 s.

**Mass spectrometry data analysis.** Raw data were processed using the MaxQuant computational platform (v.2.2.0.0) with default settings. The peak list was searched against the human proteome database (UniProt: SwissProt and TrEMBL) as well as protein sequences of interest, with an allowed precursor mass deviation of 4.5 ppm and an allowed fragment mass deviation of 20 ppm. The default setting for individual peptide mass tolerances in MaxQuant was used in the search. Cysteine carbamidomethylation was set as a static modification, while methionine oxidation, N-terminal acetylation, deamidation on asparagine and glutamine, and phosphorylation on serine, threonine, and tyrosine were set as variable modifications. Protein quantification across samples was performed using the label-free quantification (LFQ) algorithm in MaxQuant, and iBAQ (intensity-based absolute quantification) values were calculated for each protein.

## Selective ribosome profiling

**Preparation of ribosome fractions for selective ribosome profiling.** Cells were treated with 100 µg ml⁻¹ CHX for 1 min at 37 °C before collection. Cells were washed two times with PBS containing $Ca^{2+}/Mg^{2+}$ and incubated with 2 mM DSP (dithiobis(succinimidyl propionate)) for 30 min at room temperature for crosslinking. DSP solution was removed and incubated with quenching buffer (20 mM Tris-HCl pH 7.5, 300 mM glycine) for 15 min at room temperature. Cells were washed 2 times with ice cold PBS containing $Ca^{2+}/Mg^{2+}$ and lysed in lysis

buffer (20 mM HEPES, 150 mM NaCl, 5 mM MgCl$_2$, 1% IGEPAL CA-630, 20 U ml$^{-1}$ apyrase, 1 mM PMSF, 2 mM DTT, 100 µg ml$^{-1}$ CHX, cOmplete Protease Inhibitor, benzonase) on an end-over-end rotor for 30 min at 4 °C. Lysate was digested with 500 U per 1 mg RNase I (Lucigen) at 4 °C for 1 h. Ribosome pellets were isolated using a 25% sucrose cushion prepared in lysis buffer without IGEPAL CA-630 and supplemented with 20 U ml$^{-1}$ SUPERase*In (Invitrogen). Samples were centrifuged at 55,000 rpm in a SW 55 Ti rotor (Beckman Coulter) for 1 h at 4 °C. Pellets were resuspended in lysis buffer and used for anti-HaloTag pull down. Samples of the total RNA from the ribosome pellets were resuspended in TRIzol reagent (Invitrogen). HaloTag pull down was performed as described above. RPFs were recovered and libraries were prepared as described[36]. Libraries were excised from an 8% TBE polyacrylamide gel and sequenced on an Illumina NextSeq 500 or NovaSeq 6000 system.

**Ribosome profiling data analysis.** Data were analysed as previously described[64]. In brief, reads were trimmed and demultiplexed using an awk script. The UMI were extracted using umi-tools with the flag '--extract-method=regex --bc-pattern = " ˆ (?P$'", which serves to remove duplicated reads arising from PCR amplification. The remaining reads were mapped against a non-coding RNA library, including rRNA using Bowtie2 (v.2.4.2) with the parameters '-N 1 -L 15'. The remaining unaligned reads were mapped against the human genome (hg38) using STAR (v.2.7.10a) with parameters '--outFilterMismatchNmax 2 --quantMode TranscriptomeSAM GeneCounts --outSAMattributes MD NH --outFilterMultimapNmax 1'. P-site offsets were calculated using the R-package riboWaltz. Co-translational interactions of TRiC and PFD were identified as described previously[17]. In summary, genes were filtered with a coverage below an average of 0.5 reads per codon. Then, positional enrichments were calculated using a two-tailed Fisher's exact test to compare TRiC–PFD enriched ribosome–nascent chain complexes to input fractions at each position along the coding sequence. This process yielded an enrichment score, defined as an odds ratio, which compared the expected ratio at a given position to the actual observed ratio. A Benjamini–Hochberg correction was applied to test for significance at each position. Interactions were considered valid when an enrichment was observed for at least five codons and the adjusted $P$ value was below 0.05. For plotting binding profiles of chaperones with individual transcripts, one read was added at each position prior to the calculation of the odds ratio in order to avoid division by zero.

### Reporting summary

Further information on research design is available in the Nature Portfolio Reporting Summary linked to this article.

## Data availability

The mass spectrometry proteomics data generated in this study have been deposited in the ProteomeXchange Consortium via the PRIDE partner repository under the accession number PXD066622. Selective ribosome profiling data for TRiC and PFD have been deposited in the Gene Expression Omnibus (GEO) under accession number GSE304022. Source data are provided with this paper.

## Code availability

MATLAB and KNIME scripts used for SPT analysis are available from Zenodo at https://doi.org/10.5281/zenodo.16569197 (ref. 65).

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

**Acknowledgements** We thank R. Körner for assistance with mass spectrometry experiments; N. Wischnewski, R. Lange and S. Gärtner for technical support; P. Xu for assistance with analysing imaging data; M. Tanenbaum for providing the MoonTag plasmid system; Z. Liu for providing the VCP-Halo plasmid; K. Johnsson and S. Kühn for providing the TF-TMR dye and for valuable advice; and M. Hayer-Hartl for fruitful discussion. We acknowledge the MPIB core facilities, in particular R. Kim for next-generation sequencing (RRID:SCR_025746), B. Steigenberger for mass spectrometry (RRID:SCR_025745), M. Spitaler and M. Oster for live-cell imaging data acquisition, and G. Cardone for support with image data analysis (RRID:SCR_025739). This work was supported by an EMBO Long-Term Fellowship (ALTF 309-2019) to R.L. and the European Research Council (ERC Advanced Grant no. 101052783 - INSITUFOLD).

**Author contributions** R.L. and N.D. planned and performed all experiments and analysed the data. M.B.D.M., N.D. and R.L. performed the ribosome profiling experiments and M.B.D.M. analysed the related data. F.U.H. conceived the project and participated in data interpretation. R.L., N.D. and F.U.H. wrote the manuscript.

**Funding** Open access funding provided by Max Planck Society.

**Competing interests** The authors declare no competing interests.

**Additional information**
**Correspondence and requests for materials** should be addressed to F. Ulrich Hartl.

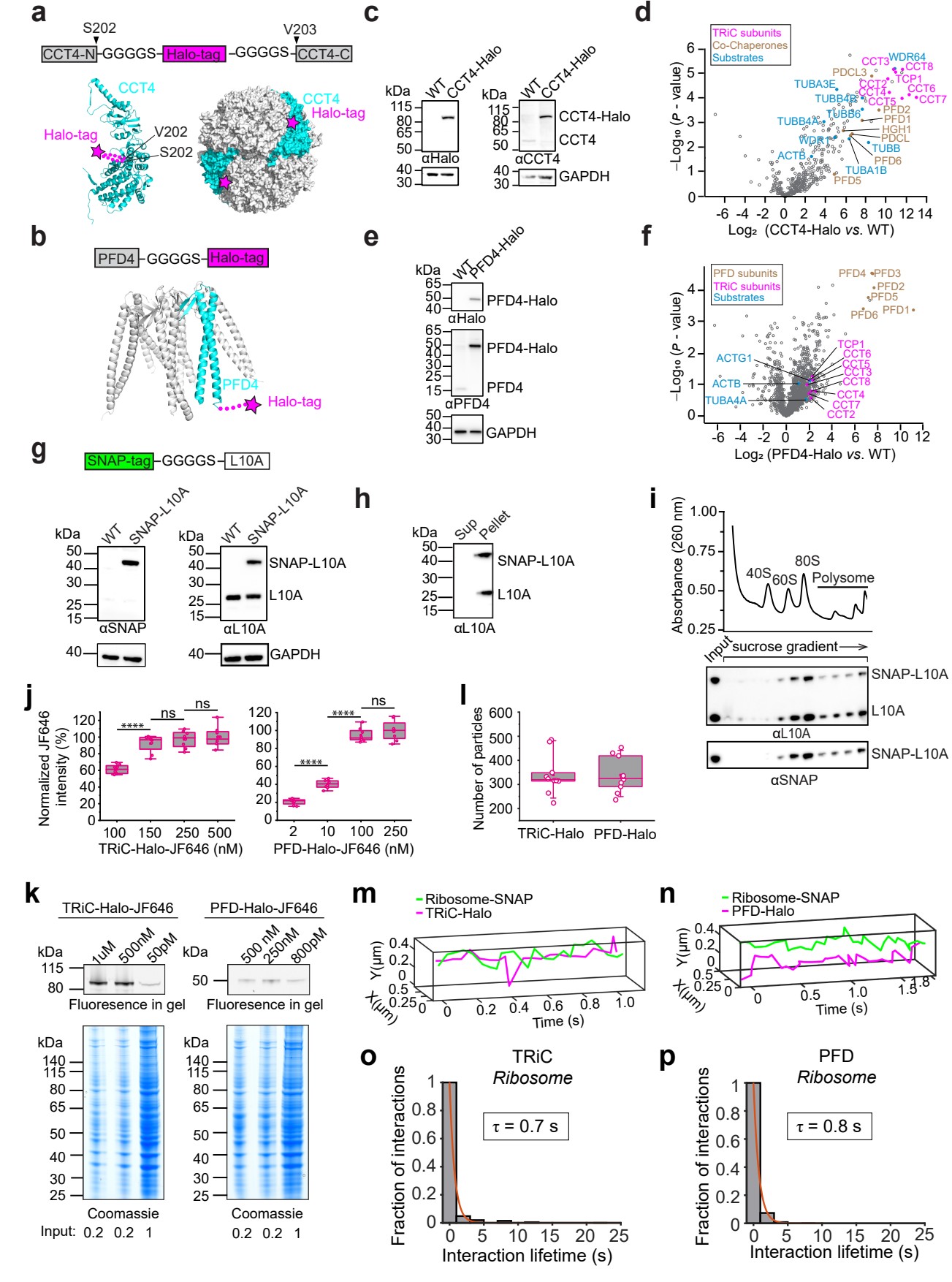

**Extended Data Fig. 1** | See next page for caption.

**Extended Data Fig. 1 | Labeling strategy for single molecule imaging of TRiC, PFD and ribosome. a**, CRISPR-Cas9 mediated knock-in of a Halo-tag at a loop region between S202 and V203 of CCT4. Structures of CCT4 in cartoon model and of TRiC in surface map (PDB:7X6Q) are shown with the site of insertion indicated. **b**, CRISPR-Cas9 mediated knock-in of Halo-tag at C-terminus of PFD4. The structure of PFD (PDB:1FXK) is shown in cartoon model with PFD4 highlighted. **c**, Immunoblot analysis of CCT4-Halo expressed in U2OS cells using anti-Halo antibody (left) and anti-CCT4 antibody (right). GAPDH served as loading control. **d**, Volcano plot representation of label-free proteome analysis of TRiC-Halo-tag pulldown fractions from cell lysate as in (c) (also see Supplementary Table 1a). **e**, Immunoblot analysis of PFD4-Halo using anti-Halo antibody (top) and anti-PFD4 antibody (middle). GAPDH served as loading control (bottom). **f**, Volcano plot representation of label-free proteome analysis of PFD-Halo-tag pulldown fractions from cell lysate as in (e) (also see Supplementary Table 1b). **g**, Construct of the large ribosomal protein L10A labeled with SNAP-tag (SNAP-tag-L10A). Immunoblot analysis of SNAP-L10A using anti-SNAP antibody (left) and anti-L10A antibody (right). **h**, Immunoblot analysis of soluble and pellet fractions from stable cell line expressing SNAP-L10A. **i**, Lysate from cells stably expressing SNAP-L10A was analyzed by sucrose gradient fractionation. UV absorbance at 260 nm was used to assess 40S, 60S, 80S and polysome traces (top). Immunoblot analysis of sucrose gradient fractions using anti-L10A and anti-SNAP-tag antibody (bottom) showing SNAP-L10A incorporation into polysomes similar to endogenous L10A. **j**, Normalized fluorescence intensity of cells endogenously expressing TRiC-Halo (left) or PFD-Halo (right) when incubated with fluorescent ligand JF646. "no significance (ns)" indicates the dye concentration at which the fluorescence signal reaches a plateau ($n$ = 8 cells for each condition). **k**, In-gel fluorescence assay used to quantify the fraction of labeled TRiC-Halo (left) and PFD-Halo (right). To avoid saturation, 5-fold less protein was loaded for samples at plateau dye concentration than for those at the imaging concentration. Fluorescence from the gel (top) and total protein staining with Coomassie (bottom). **l**, Numbers of TRiC-Halo and PFD-Halo single molecules detected in the first frame of each video. Cell numbers [n]: TRiC-Halo, $n$ = 10, PFD-Halo, $n$ = 10. **m,n**, Co-movement trajectory of TRiC-ribosome (**m**) and PFD-ribosome (**n**) as shown in Fig. 1b and c, respectively. **o,p**, 1-cumulative distribution function (1-CDF) of interaction lifetimes of ribosome with TRiC (**o**) and PFD (**p**) were fitted with one-component decay models.

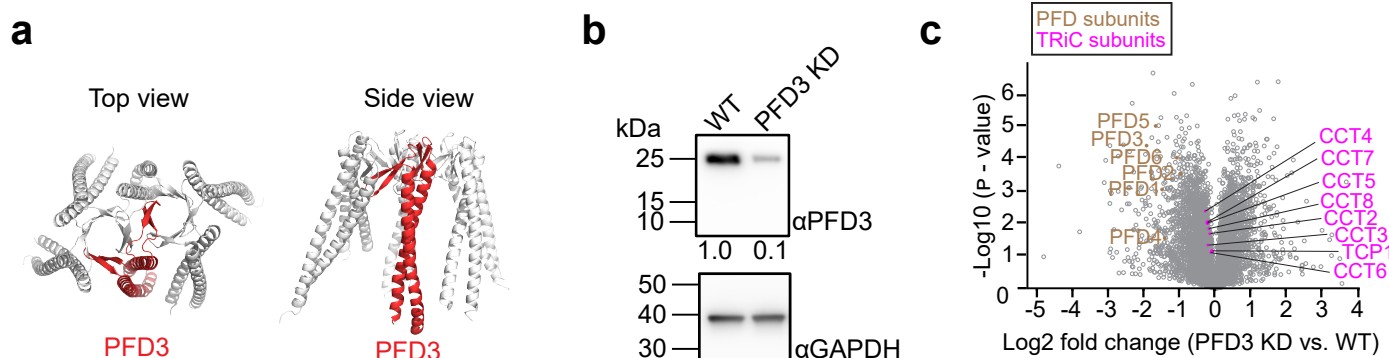

**a**

Top view    Side view

PFD3    PFD3

**b**

**c**

**Extended Data Fig. 2 | Generation of stable cell line with PFD knock-down.**
**a**, Structure of PFD (PDB: 1FXK) in cartoon model. The position of subunit PFD3, essential for complex assembly, is highlighted in red. **b**, Immunoblot analysis of cell lysate from wild-type (WT) and PFD3 knock-down (KD) cells using anti-PFD3 antibody. Protein amount of PFD3 in PFD3 KD cells was reduced by ~90% relative to WT cells. GAPDH served as a control. **c**, Volcano plot representation of label-free proteome analysis of total cell lysate of PFD3 KD cells (also see Supplementary Table 1c). PFD and TRiC subunits are marked.

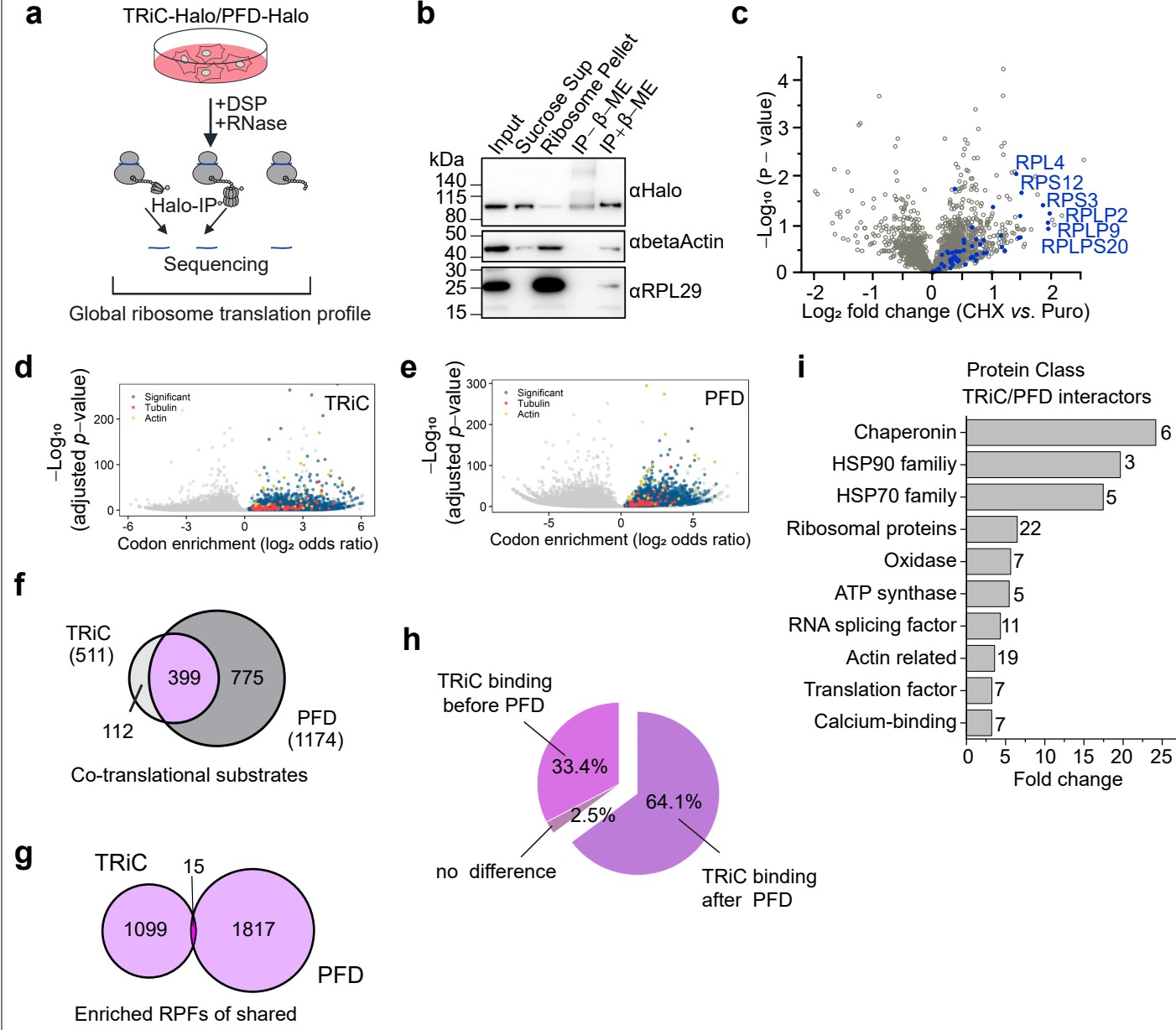

**Extended Data Fig. 3 | Selective TRiC/PFD ribosome profiling. a**, Schematic representation of selective ribosome profiling of TRiC-Halo and PFD-Halo in U2OS cells. Cells expressing TRiC-Halo or PFD-Halo are chemically crosslinked with DSP to enrich chaperone associated ribosome-nascent chain complexes (RNCs). Polysomes are digested with RNase to yield monosomes with ribosome protected fragments (RPFs). Digested polysomes are centrifuged through a sucrose gradient to generate a ribosome pellet. RPFs are isolated from the total population of RNCs, as well as TRiC- or PFD-bound RNCs following Halo-tag pulldown, and then sequenced. **b**, Immunoblot of samples harvested at different steps in (**a**) showing that TRiC-Halo associated RNCs are enriched. β-Mercaptoethanol (β-ME) served as a reducing reagent to cleave the disulfide bond of DSP. **c**, Volcano plot representation of label-free proteome analysis of ribosome pellet fraction followed by TRiC-Halo-tag pulldown as illustrated in (**a**)

with cells treated with 100 μg/mL cycloheximide (CHX) relative to cells treated with 300 μg/mL puromycin (Puro) (also see Supplementary Table 1d). **d,e**, Volcano plot of chaperone enrichment, with each point representing a codon in the translatome. Colored points show the positions with significant TRiC (**d**) or PFD (**e**) enrichment classified as binding sites and used to identify substrates, such as actin (yellow) and tubulin (red). **f**, Overlap of co-translational TRiC (511) and PFD (1174) substrates. The number of shared substrates (399) is indicated (also see Supplementary Table 2a,b). **g**, Overlap of chaperone-enriched positions in substrates shared by TRiC and PFD. **h**, Fraction of ORFs of shared substrates for which the first TRiC binding site is either after or before the first binding site of PFD. **i**, GO term analysis (PANTHER protein class) shows categories of shared substrates of TRiC and PFD. Numbers indicate proteins per category.

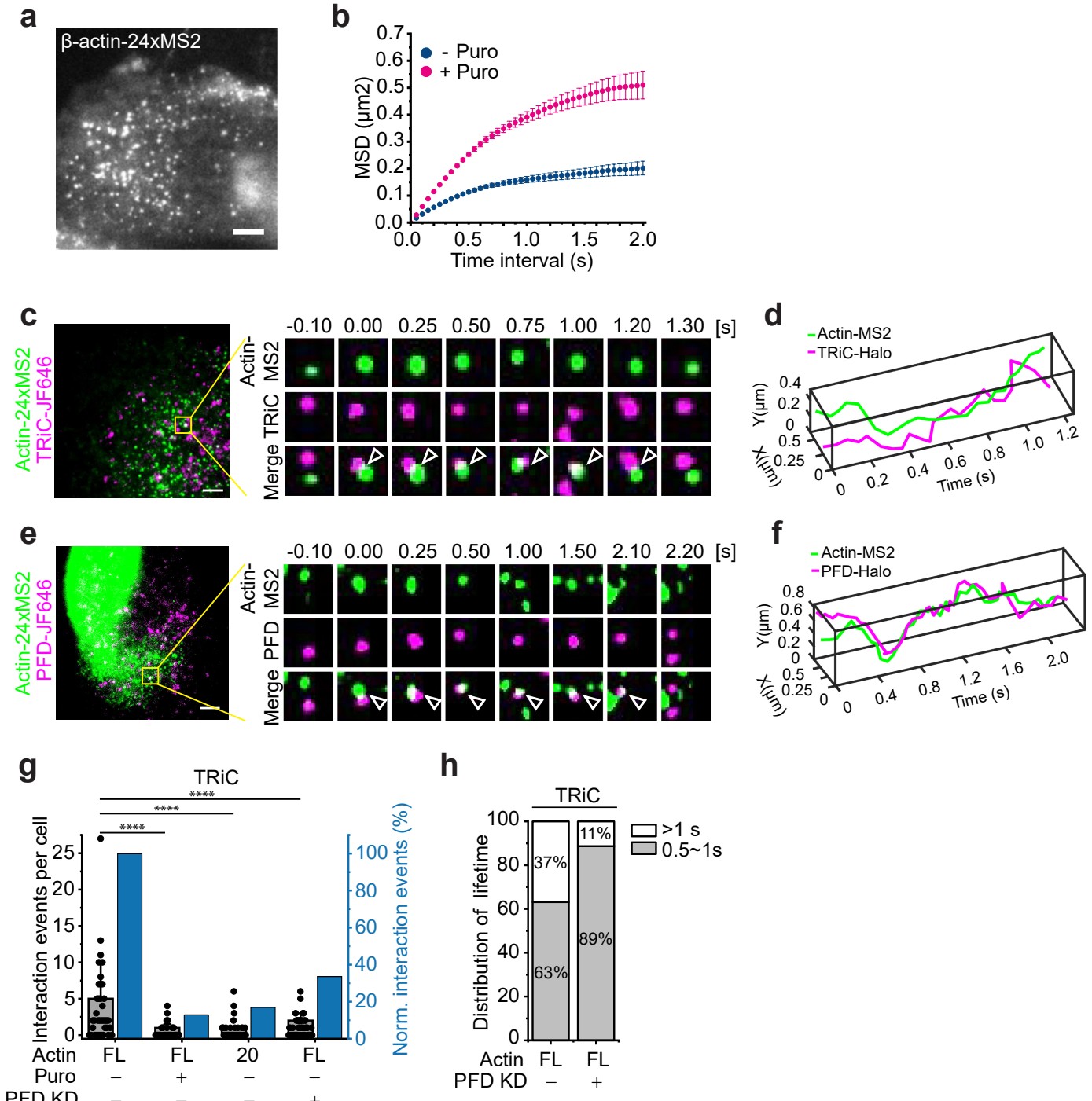

**Extended Data Fig. 4 | Co-translational interaction of TRiC/PFD and actin.**
**a**, Representative image of actin-24xMS2-GFP mRNA in live cells. **b**, Mean square displacement (MSD) of actin mRNAs under normal conditions (blue) and after puromycin (Puro) treatment (magenta). mRNA diffuses faster after Puro treatment to release NCs from ribosomes, suggesting these mRNAs are translationally active (number of cells [n]: Control, $n = 10$; Puro, $n = 8$). **c,e**, Live-cell imaging showing the co-translational association of TRiC-actin (**c**) and PFD-actin (**e**). Left, dual-color imaging. Right, selected fluorescence time points of the co-movement events corresponding to the yellow square at left.

**d,f**, Co-movement trajectory of TRiC-actin (**d**) and PFD-actin (**f**) as shown in (**c**) and (**e**), respectively. **g**, Co-movement events per cell (gray) and normalized co-movement events (total interaction events divided by total number of cells; blue) of TRiC-ribosome and PFD-ribosome interactions under the conditions of Fig. 1g. The horizontal line within the box indicates the median, boxes indicate upper and lower quartile and whisker caps 10th – 90th percentile, respectively. **** $P < 0.0001$ by one way ANOVA. **h**, Distribution of co-translational interaction lifetimes for interactions of TRiC with full-length (FL) actin in control cells (left) and after PFD knock down (KD) (right) under the conditions of Fig. 1g.

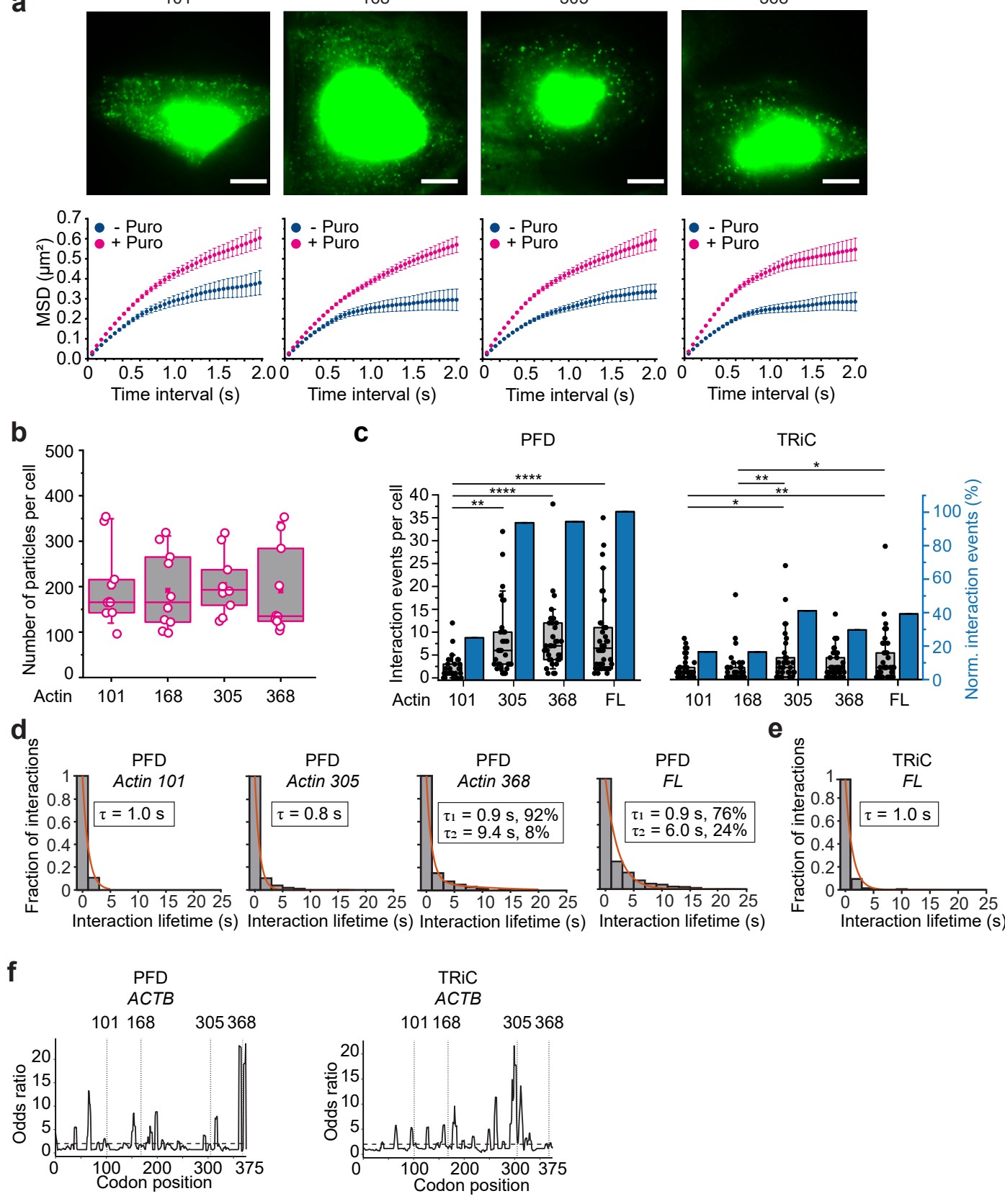

**Extended Data Fig. 5** | See next page for caption.

**Extended Data Fig. 5 | Co-translational interaction of TRiC/PFD with actin chains of increasing length. a**, Representative images of cells expressing MS2-MCP-GFP mRNAs coding for actin constructs of 101, 168, 306 and 368 amino acids. Scale bar, 5 μm (top). Mean square displacement (MSD) of different truncated actin mRNAs under normal conditions (blue) and after puromycin (Puro) treatment for 30 min (magenta). mRNA diffuses faster after Puro treatment to release NCs from ribosomes, suggesting the mRNA is translationally active (number of cells [n]: Control: 101, $n = 8$; 168, $n = 10$; 305, $n = 10$; 368, $n = 10$; Puro: 101, $n = 16$; 168, $n = 15$; 305, $n = 10$; 368. $n = 9$). **b**, Number of truncated actin mRNA molecules detected in the first frame of each video (number of cells [n]: 101, $n = 10$; 168, $n = 10$; 305, $n = 10$; 368, $n = 10$; FL, $n = 10$). The horizontal line within the box indicates the median, the star within the box indicates the mean, boxes indicate upper and lower quartile and whisker caps $10^{th} – 90^{th}$ percentile, respectively. **c**, Frequency of co-movement events per cell (gray) and normalized frequency of co-movement events (total interaction events divided by total number of cells; blue) of PFD (left) and TRiC (right) with actin mRNAs translating actin truncations under the condition in Fig. 1j,k. The horizontal line within the box indicates the median, boxes indicate upper and lower quartile and whisker caps $10^{th} – 90^{th}$ percentile, respectively. *$P < 0.05$ and **$P < 0.01$ and ****$P < 0.0001$ by one-way ANVOA. **d**, 1-cumulative distribution function (1-CDF) of co-translational interaction lifetimes of PFD with actin 101, actin 305, actin 368 and full-length (FL) actin were fitted with different exponential decay models. **e**, 1-CDF of co-translational interaction lifetimes of TRiC with FL actin fitted with a one-component exponential decay model. **f**, PFD and TRiC enrichment profiles of β-actin (ACTB) from selective ribosome profiling in Extended Data Fig. 3.

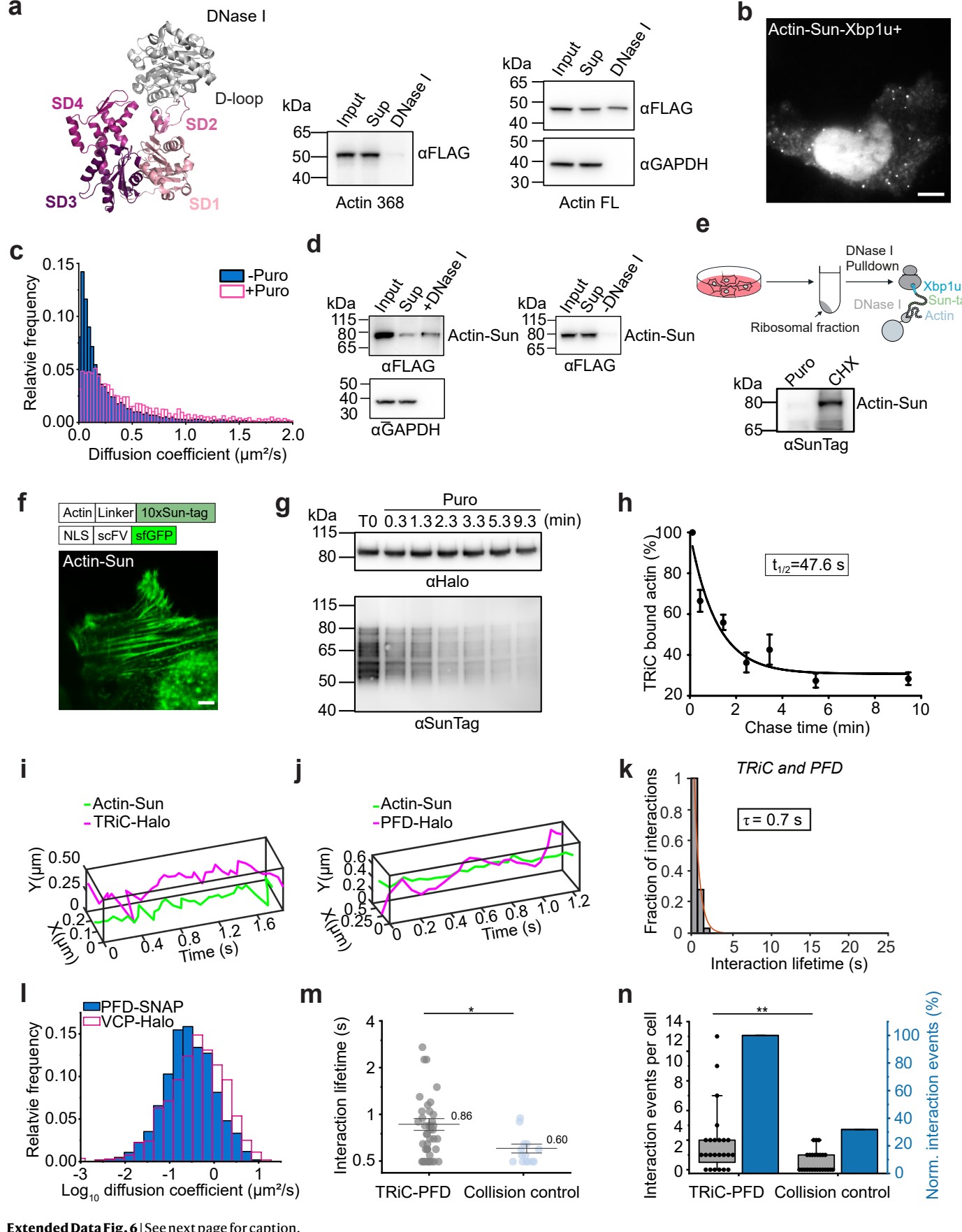

**Extended Data Fig. 6** | See next page for caption.

**Extended Data Fig. 6 | Post-translational interaction of TRiC/PFD and actin.**
**a**, DNase I pulldown of actin 368 (in the presence of MG-132 to prevent rapid degradation of actin 368) and full-length (FL) actin. Structure of the actin-DNase I complex (PDB: 3W3D) in cartoon model (left). Immunoblot analysis of DNase I pulldown fractions using anti-FLAG antibody to detect FLAG tag that is placed before the stop codon. GAPDH served as control. **b**, Representative fluorescence image of actin-Sun-Xbp1u+ in live cells (see Fig. 2a). Scale bar, 5 μm. **c**, Distribution of diffusion coefficients of individual actin-Sun molecules under normal condition (blue) and after puromycin (Puro) treatment (magenta). Actin-Sun diffuses faster after Puro treatment to release NCs from ribosomes (number of cells [n]: Control, $n = 13$; Puro, $n = 13$). **d**, Immunoblot analysis of DNase I pulldown fractions from cells expressing actin-Sun in the presence of MG-132 using anti-FLAG antibody. Actin-Sun binds DNase I (left) and there is no non-specific binding to beads without DNase I (right). GAPDH served as control. **e**, Top, schematic of DNase I pulldown assay with ribosome pellets from cells expressing actin-Sun. Bottom, immunoblot analysis of DNase I pulldown fractions prepared in the presence of 300 μg/mL puromycin (Puro) or 100 μg/mL cycloheximide (CHX). This result demonstrates that actin-Sun folds to the native state while still associated with the ribosome. **f**, Representative fluorescence image of actin-Sun without Xbp1u+ arrest sequence, showing incorporation into actin filaments in live cells. Scale bar, 5 μm. **g**, Immunoblot analysis of fractions of anti-Halo immunoprecipitation (IP) from cells expressing TRiC-Halo and actin-Sun after different times of chase after addition of 300 μg/mL puromycin (Puro) using anti-Halo and anti-SunTag antibody ($n = 3$ independent experiments). Note that the C-terminal SunTag peptide array is still incomplete when actin folds post-translationally on TRiC, indicating that folding initiates while the actin-Sun-Xbp1u+ construct is still ribosome associated. **h**, Actin-Sun shows a dissociation halftime of ~48 s from TRiC based on data in **g**. Data represent the mean ± s.e.m. ($n = 3$ independent experiments). **i,j**, Co-movement trajectories of TRiC (**i**) and PFD (**j**) with translated actin as shown in Fig. 2b,c, respectively. **k**, 1-cumulative distribution function (1-CDF) of lifetimes for interactions of TRiC and PFD fitted with a one-component exponential decay model. **l**, Distribution of diffusion coefficients of PFD-SNAP (blue) and VCP-Halo (magenta) are similar. **m**, Distribution of the interaction lifetimes of TRiC-PFD interactions (data from Fig. 2j) and TRiC-VCP (collision control) (number of cells [n]: control, $n = 24$; collision control, $n = 25$). Error bars represent mean ± s.e.m. * $P < 0.05$ by Mann Whitney test. **n**, Co-movement events per cell (gray) and normalized co-movement events (blue) of TRiC-PFD interactions (data from Fig. 2k) and TRiC-VCP (collision control) under the conditions in (**m**). The horizontal line within the box indicates the median, boxes indicate upper and lower quartile and whisker caps 10th – 90th percentile, respectively. ** $P < 0.01$ by Mann Whitney test.

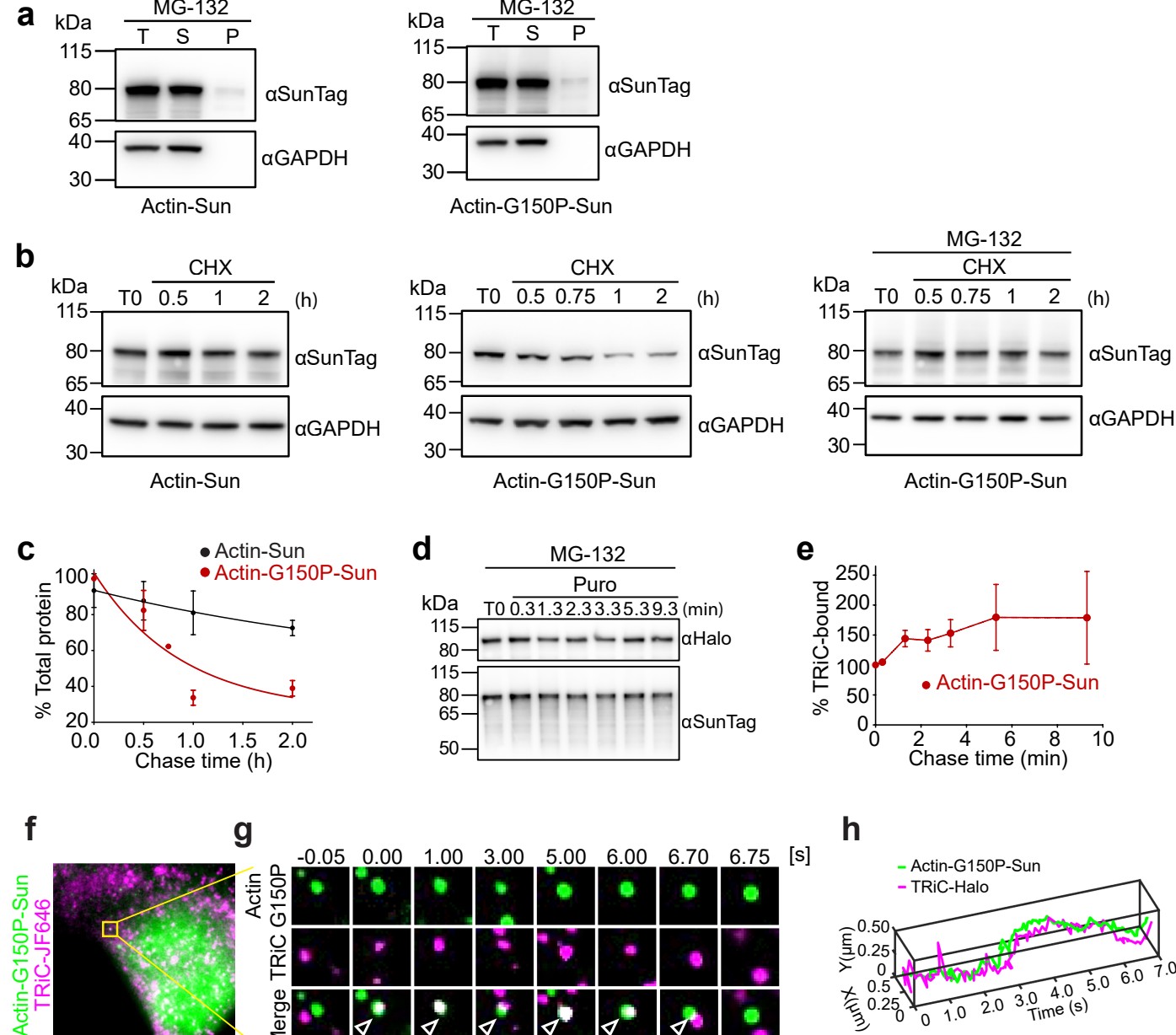

**Extended Data Fig. 7 | Chaperonin interactions with folding-defective actin.**
**a**, Solubility of actin-G150P-Sun. Lysates from cells expressing either actin-Sun (left) or actin-G150P-Sun (right) in the presence of 10 µM MG-132 were analyzed by centrifugation at 20,000 x g, followed by immunoblotting of total (T), soluble (S) and pellet (P) fractions with anti-SunTag antibody. GAPDH served as control ($n = 3$ independent experiments). **b**, Turnover of actin-Sun and actin-G150P-Sun measured by cycloheximide (CHX) chase. Actin-Sun (left) and actin-G150P-Sun (middle and right) without Xbp1u+ arrest sequence were expressed for 24 h, followed by treatment with 100 µg/mL CHX for the times indicated. Actin-G150P-Sun was analyzed without (middle) and with (right) proteasome inhibitor MG-132 (10 µM). Cell lysates were analyzed by immunoblotting with anti-SunTag antibody. GAPDH served as control ($n = 3$ independent experiments). **c**, Degradation kinetics of actin-Sun and actin-G150P-Sun based on chase experiments as in **b**.

Data represent the mean ± s.e.m. ($n = 3$ independent experiments). **d**, Actin-G150P-Sun shows prolonged interaction with TRiC. Cells expressing TRiC-Halo and actin-G150P-Sun-Xbp1u+ were subjected to a chase with puromycin (300 µg/mL) in the presence of MG-132 (10 µM). At the times indicated, TRiC-Halo was isolated by anti-Halo immunoprecipitation, followed by anti-Halo-tag and anti-SunTag immunoblotting. **e**, Dissociation kinetics of actin-G150P-Sun-Xbp1u+ from data in **d** ($n = 3$ independent experiments). Data represent the mean ± s.e.m. Transit through TRiC for actin-Sun-Xbp1u+ is shown in Extended Data Fig. 6g,h. **f**, Live cell imaging showing the association of TRiC and actin-G150P-Sun. **g**, Prolonged co-movement of TRiC (magenta) and actin-G150P-Sun (green) highlighted by the yellow square in **f**. **h**, Co-movement trajectory of TRiC-actin-G150P as shown in **g**.

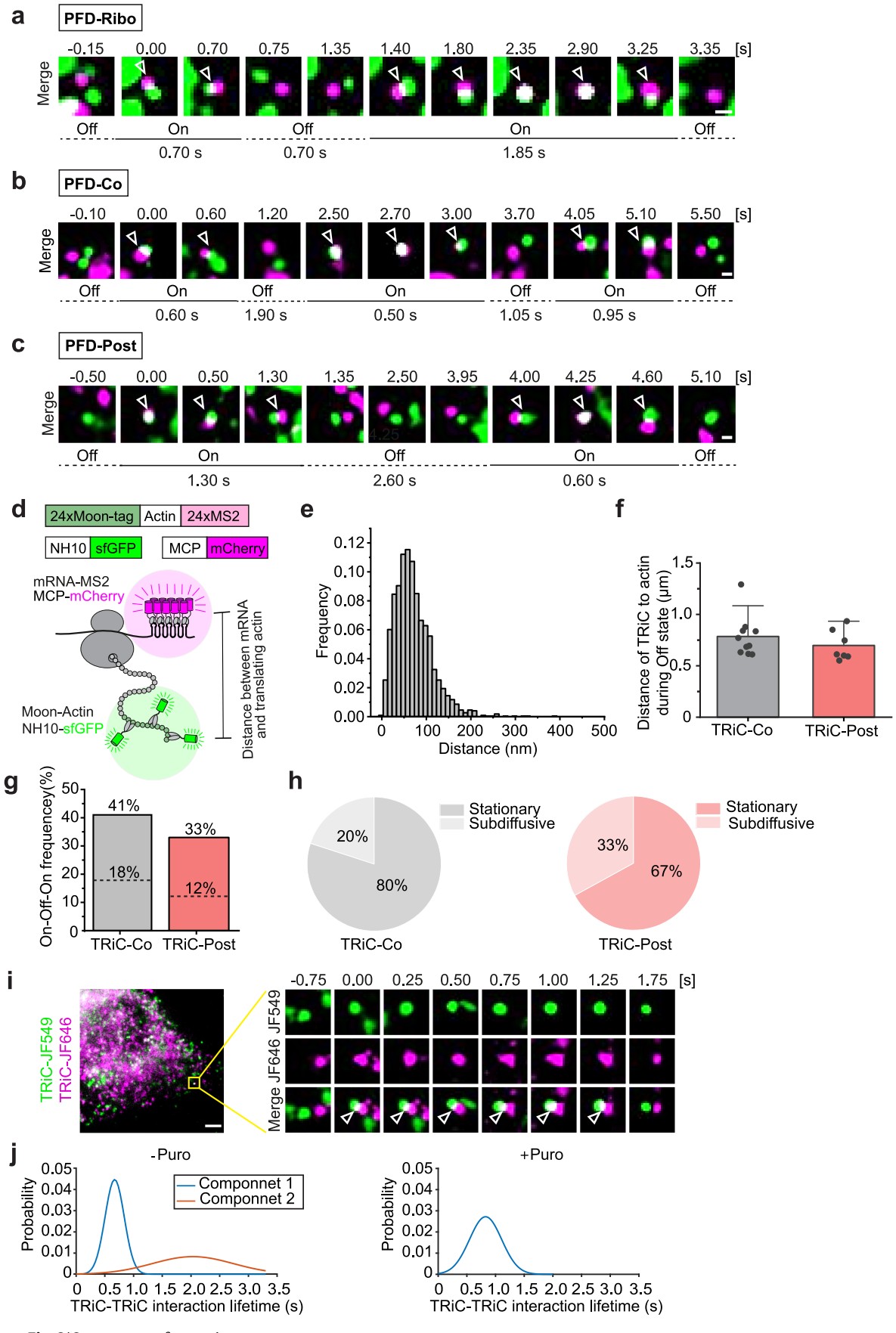

**Extended Data Fig. 8** | See next page for caption.

**Extended Data Fig. 8 | Live cell imaging of successive chaperonin cycles.**
**a-c**, Dual-color live cell imaging showing successive binding of PFD with translating ribosome (Ribo) (**a**), translating actin nascent chain (Co) (**b**) and translated actin polypeptide (Post) (**c**). Scale bars, 500 nm. **d**, Schematic of distance ruler to measure the distance between mRNA and translating polypeptides. A 24xMoon-tag peptide array (15 residues per epitope)[56] is placed at the N-terminus of actin to allow real-time imaging of translating actin via sfGFP tagged Moon-tag specific nanobody NH10 (NH10-sfGFP). Actin mRNA is imaged by mCherry tagged MS2 coat protein (MCP-mCherry) that binds to 24 copies of MS2 stem loops in the 3'UTR. **e**, Distance distribution between mRNA and translating polypeptides as described in **d**. **f**, Average maximal distance between TRiC-Halo and actin in the co- and post-translational off state. **g**, Frequency of on-off-on events of TRiC-Halo and actin during co- and -post-translational interactions. Number of trajectories analyzed: Co, 59 trajectories with off-state-TRiC remaining in focus for at least 500 ms out of 133 total trajectories, among which 24 trajectories with on-off-on behaviour (41%); Post, 42 trajectories with off-state-TRiC remaining in focus for at least 500 ms out of 118 total trajectories, among which 14 trajectories with on-off-on behaviour (33%). Dashed lines indicate the overall frequency of on-off-on events compared to total trajectories: 18% and 12% for co-and post-translational interactions, respectively. Note that restricting the analysis to TRiC complexes remaining in focus for 500 ms may over-estimate the frequency of on-off-on events, while including all trajectories likely under-estimates these events, as particles leaving the focal plane during the off-state cannot be tracked for potential rebinding. Number of cells: TRiC-Co, $n = 38$; TRiC-Post, $n = 24$. **h**, Diffusion behaviour of TRiC-Halo during the off state before rebinding to actin during co- and post-translational function (number of cells: TRiC-Co, $n = 38$; TRiC-Post, $n = 24$). **i**, Live cell imaging showing the association of TRiC molecules labeled with different JF dyes. Left, representative dual-color images. Right, selected fluorescence time points of co-movement events highlighted by the yellow square on the left. Scale bar, 5 μm. **j**, Distributions of interaction lifetimes for interactions between individual TRiC molecules (assayed as in Fig. 4g,i) under control conditions (left) and after puromycin (Puro) treatment (right) are fitted using a two-component Gaussian function.

## Reporting Summary

## Statistics

For all statistical analyses, confirm that the following items are present in the figure legend, table legend, main text, or Methods section.

| n/a | Confirmed | |
|---|---|---|
| ☐ | ☒ | The exact sample size (*n*) for each experimental group/condition, given as a discrete number and unit of measurement |
| ☐ | ☒ | A statement on whether measurements were taken from distinct samples or whether the same sample was measured repeatedly |
| ☐ | ☒ | The statistical test(s) used AND whether they are one- or two-sided<br>*Only common tests should be described solely by name; describe more complex techniques in the Methods section.* |
| ☒ | ☐ | A description of all covariates tested |
| ☒ | ☐ | A description of any assumptions or corrections, such as tests of normality and adjustment for multiple comparisons |
| ☐ | ☒ | A full description of the statistical parameters including central tendency (e.g. means) or other basic estimates (e.g. regression coefficient) AND variation (e.g. standard deviation) or associated estimates of uncertainty (e.g. confidence intervals) |
| ☐ | ☒ | For null hypothesis testing, the test statistic (e.g. *F*, *t*, *r*) with confidence intervals, effect sizes, degrees of freedom and *P* value noted<br>*Give P values as exact values whenever suitable.* |
| ☒ | ☐ | For Bayesian analysis, information on the choice of priors and Markov chain Monte Carlo settings |
| ☒ | ☐ | For hierarchical and complex designs, identification of the appropriate level for tests and full reporting of outcomes |
| ☒ | ☐ | Estimates of effect sizes (e.g. Cohen's *d*, Pearson's *r*), indicating how they were calculated |

*Our web collection on statistics for biologists contains articles on many of the points above.*

## Software and code

Policy information about availability of computer code

Data collection
Microscopy images were collected using the ZEN black 2.1 SP3 software.
TRiC/PFD selective ribosome profiling data were obtained by using Trim Galore v0.6.10, Bowtie2 v2.4.2, STAR v2.7.10a, riboWaltz v2.0 and R software v4.3.1.
Western blot images were acquired using the Amersham ImageQuant 800 Control Software, version 2.1.0.3.
Polysome profiles were obtained using the Triax Flow Cell Firmware v2.30.4.211.
Proteomics data was recorded on an Easy-nLC 1200 system coupled to a QExactive HF mass spectrometer (QExactive HF -Orbitrap 2.13 build 3162) or an Orbitrap Exploris 480 mass spectrometer (Orbitrap Exploris 480 4.1 335.19) using XCalibur 4.3.
In-gel fluorescence images were acquired on an Amersham Typhoon Biomolecular Imager.

| Data analysis | TRiC and PFD selective ribosome profiling data were processed analyzed following the analysis pipeline described by Stein et al. 2019.<br>Statistical analysis and graphs were generated with GraphPad Prism 10 and Origin version 2023b.<br>Data fitting was performed with SigmaPlot v14.0.<br>Western blot bands and microscopy pictures were quantified using ImageJ v2.1.0/1.54f.<br>Trajectories of molecules from live-cell single-particle tracking were acquired using the ImageJ plugin TrackMate v6.0.1.<br>Co-movement dynamics between chaperones and substrates were analyzed using KNIME v4.5.2.<br>Diffusion coefficients of single molecules and decay fitting of interaction lifetimes were calculated using MATLAB (R2021b).<br>Proteomics data were processed using the MaxQuant computational platform (version 2.2.0.0) with default settings. The peak list was searched against the human proteome database (UniProt: SwissProt and TrEMBL) as well as protein sequences of interest.<br>Volcano plots and statistical analysis of proteomic data were performed using Perseus v1.6.2.3.<br>Code availability: The MATLAB and KNIME scripts used to analyze single particle tracking data in this study are available on Zenodo under the DOI 10.5281/zenodo.16569197. |
|---|---|

For manuscripts utilizing custom algorithms or software that are central to the research but not yet described in published literature, software must be made available to editors and reviewers. We strongly encourage code deposition in a community repository (e.g. GitHub). See the Nature Portfolio guidelines for submitting code & software for further information.

## Data

Policy information about availability of data

All manuscripts must include a data availability statement. This statement should provide the following information, where applicable:
- Accession codes, unique identifiers, or web links for publicly available datasets
- A description of any restrictions on data availability
- For clinical datasets or third party data, please ensure that the statement adheres to our policy

The mass spectrometry proteomics data have been deposited to the ProteomeXchange Consortium via the PRIDE partner repository with the dataset identifier PXD066622.
The TRiC/PFD selective ribosome profiling data generated in this study have been deposited in the Gene Expression Omnibus (GEO) under accession number GSE304022.

## Research involving human participants, their data, or biological material

Policy information about studies with human participants or human data. See also policy information about sex, gender (identity/presentation), and sexual orientation and race, ethnicity and racism.

| Reporting on sex and gender | Not applicable |
|---|---|
| Reporting on race, ethnicity, or other socially relevant groupings | Not applicable |
| Population characteristics | Not applicable |
| Recruitment | Not applicable |
| Ethics oversight | Not applicable |

Note that full information on the approval of the study protocol must also be provided in the manuscript.

# Field-specific reporting

Please select the one below that is the best fit for your research. If you are not sure, read the appropriate sections before making your selection.

☒ Life sciences    ☐ Behavioural & social sciences    ☐ Ecological, evolutionary & environmental sciences

For a reference copy of the document with all sections, see nature.com/documents/nr-reporting-summary-flat.pdf

# Life sciences study design

All studies must disclose on these points even when the disclosure is negative.

| Sample size | At least three independent experiments were performed in all cases except for the selective ribosome profiling experiments, which were performed with two biological replicates. These replicate numbers were chosen in accordance with commonly accepted field standards and to enable statistical analysis. |
|---|---|
| Data exclusions | No data were excluded in this study. |
| Replication | A minimum of three independent replicates were performed for all experiments, except for the selective ribosome profiling experiments, which were performed with two biological replicates. The exact number of replicates is specified in the figure legends. |

| Randomization | Randomization was not performed because there was no assignment of data points to distinct groups. |
| Blinding | No blinding was performed, as the risk of bias from the experimentalist was considered negligible for this study. |

# Behavioural & social sciences study design

All studies must disclose on these points even when the disclosure is negative.

| Study description | *Briefly describe the study type including whether data are quantitative, qualitative, or mixed-methods (e.g. qualitative cross-sectional, quantitative experimental, mixed-methods case study)* |
| Research sample | *State the research sample (e.g. Harvard university undergraduates, villagers in rural India) and provide relevant demographic information (e.g. age, sex) and indicate whether the sample is representative. Provide a rationale for the study sample chosen. For studies involving existing datasets, please describe the dataset and source.* |
| Sampling strategy | *Describe the sampling procedure (e.g. random, snowball, stratified, convenience). Describe the statistical methods that were used to predetermine sample size OR if no sample-size calculation was performed, describe how sample sizes were chosen and provide a rationale for why these sample sizes are sufficient. For qualitative data, please indicate whether data saturation was considered, and what criteria were used to decide that no further sampling was needed.* |
| Data collection | *Provide details about the data collection procedure, including the instruments or devices used to record the data (e.g. pen and paper, computer, eye tracker, video or audio equipment) whether anyone was present besides the participant(s) and the researcher, and whether the researcher was blind to experimental condition and/or the study hypothesis during data collection.* |
| Timing | *Indicate the start and stop dates of data collection. If there is a gap between collection periods, state the dates for each sample cohort.* |
| Data exclusions | *If no data were excluded from the analyses, state so OR if data were excluded, provide the exact number of exclusions and the rationale behind them, indicating whether exclusion criteria were pre-established.* |
| Non-participation | *State how many participants dropped out/declined participation and the reason(s) given OR provide response rate OR state that no participants dropped out/declined participation.* |
| Randomization | *If participants were not allocated into experimental groups, state so OR describe how participants were allocated to groups, and if allocation was not random, describe how covariates were controlled* |

# Ecological, evolutionary & environmental sciences study design

All studies must disclose on these points even when the disclosure is negative.

| Study description | *Briefly describe the study. For quantitative data include treatment factors and interactions, design structure (e.g. factorial, nested, hierarchical), nature and number of experimental units and replicates.* |
| Research sample | *Describe the research sample (e.g. a group of tagged Passer domesticus, all Stenocereus thurberi within Organ Pipe Cactus National Monument), and provide a rationale for the sample choice. When relevant, describe the organism taxa, source, sex, age range and any manipulations. State what population the sample is meant to represent when applicable. For studies involving existing datasets, describe the data and its source.* |
| Sampling strategy | *Note the sampling procedure. Describe the statistical methods that were used to predetermine sample size OR if no sample-size calculation was performed, describe how sample sizes were chosen and provide a rationale for why these sample sizes are sufficient.* |
| Data collection | *Describe the data collection procedure, including who recorded the data and how.* |
| Timing and spatial scale | *Indicate the start and stop dates of data collection, noting the frequency and periodicity of sampling and providing a rationale for these choices. If there is a gap between collection periods, state the dates for each sample cohort. Specify the spatial scale from which the data are taken* |
| Data exclusions | *If no data were excluded from the analyses, state so OR if data were excluded, describe the exclusions and the rationale behind them, indicating whether exclusion criteria were pre-established.* |
| Reproducibility | *Describe the measures taken to verify the reproducibility of experimental findings. For each experiment, note whether any attempts to repeat the experiment failed OR state that all attempts to repeat the experiment were successful.* |
| Randomization | *Describe how samples/organisms/participants were allocated into groups. If allocation was not random, describe how covariates were controlled. If this is not relevant to your study, explain why.* |
| Blinding | *Describe the extent of blinding used during data acquisition and analysis. If blinding was not possible, describe why OR explain why blinding was not relevant to your study.* |

Did the study involve field work? ☐ Yes ☒ No

## Field work, collection and transport

| | |
|---|---|
| Field conditions | *Describe the study conditions for field work, providing relevant parameters (e.g. temperature, rainfall).* |
| Location | *State the location of the sampling or experiment, providing relevant parameters (e.g. latitude and longitude, elevation, water depth).* |
| Access & import/export | *Describe the efforts you have made to access habitats and to collect and import/export your samples in a responsible manner and in compliance with local, national and international laws, noting any permits that were obtained (give the name of the issuing authority, the date of issue, and any identifying information).* |
| Disturbance | *Describe any disturbance caused by the study and how it was minimized.* |

# Reporting for specific materials, systems and methods

We require information from authors about some types of materials, experimental systems and methods used in many studies. Here, indicate whether each material, system or method listed is relevant to your study. If you are not sure if a list item applies to your research, read the appropriate section before selecting a response.

### Materials & experimental systems

| n/a | Involved in the study |
|---|---|
| ☐ | ☒ Antibodies |
| ☐ | ☒ Eukaryotic cell lines |
| ☒ | ☐ Palaeontology and archaeology |
| ☒ | ☐ Animals and other organisms |
| ☒ | ☐ Clinical data |
| ☒ | ☐ Dual use research of concern |
| ☒ | ☐ Plants |

### Methods

| n/a | Involved in the study |
|---|---|
| ☒ | ☐ ChIP-seq |
| ☒ | ☐ Flow cytometry |
| ☒ | ☐ MRI-based neuroimaging |

## Antibodies

| | |
|---|---|
| Antibodies used | anti-β-actin antibody (Abcam, ab8226), anti-CCT4 antibody (Merck, HPA029349), anti-DYKDDDDK Tag antibody (Cell Signal, 2368S), anti-GAPDH antibody (Merck, MAB374), anti-Halo-tag antibody (Promega, G9211), anti-prefoldin 3 antibody (Santa Cruz, sc-390524), anti-prefoldin 4 antibody (Thermo Fisher, 16045-1-AP), anti-RPL10A antibody (Abcam, ab174318), anti-RPL29 antibody (Thermo Fisher, PA5-27545), anti-GCN4 antibody (Addgene, 218104-rAb) and anti-SNAP-tag antibody (NEB, P9310S), conjugated goat anti-mouse immunoglobulin G (IgG)-horseradish peroxidase (HRP) (Merck, A4416), conjugated goat anti-rabbit immunoglobulin G (IgG)-horseradish peroxidase (HRP) (Merck, A9169). |
| Validation | All antibodies are validated by commercial suppliers:<br>anti-β-actin antibody (Abcam, ab8226): validated by Abcam and referenced in 3365 articles.<br>anti-CCT4 antibody (Merck, HPA029349): validated by Merck using orthogonal RNAseq.<br>anti-DYKDDDDK Tag antibody (Cell Signaling Technologies, 2368S): validated by Cell Signaling Technologies and referenced in 899 articles.<br>anti-GAPDH antibody (Merck, MAB374): validated by Merck and referenced in 3155 articles.<br>anti-Halo-tag antibody (Promega, G9211): validated by Promega.<br>anti-prefoldin 3 antibody (Santa Cruz, sc-390524): validated by Santa Cruz Biotechnology.<br>anti-prefoldin 4 antibody (Proteintech, 16045-1-AP): validated by Proteintech and referenced in one article.<br>anti-RPL10A antibody (Abcam, ab174318): validated by Abcam and referenced in 10 articles.<br>anti-RPL29 antibody (Thermo Fisher, PA5-27545): Knockdown validation by Thermo Fisher and referenced in one article.<br>anti-GCN4 antibody (Addgene, 218104-rAb): validated by Zahnd et al., 2004<br>anti-SNAP-tag antibody (NEB, P9310S): validated by NEB and referenced in 513 articles. |

## Eukaryotic cell lines

Policy information about <u>cell lines and Sex and Gender in Research</u>

| | |
|---|---|
| Cell line source(s) | U2OS cell line was purchased from American Type Culture Collection (ATCC) (#HTB-96). |
| Authentication | No further authentication was performed. |
| Mycoplasma contamination | All stable U2OS cell lines as well as the wild type U2OS cell line were negative for mycoplasma. |

| Commonly misidentified lines<br>(See ICLAC register) | None. |
|---|---|

## Palaeontology and Archaeology

| Specimen provenance | Provide provenance information for specimens and describe permits that were obtained for the work (including the name of the issuing authority, the date of issue, and any identifying information). Permits should encompass collection and, where applicable, export. |
|---|---|
| Specimen deposition | Indicate where the specimens have been deposited to permit free access by other researchers. |
| Dating methods | If new dates are provided, describe how they were obtained (e.g. collection, storage, sample pretreatment and measurement), where they were obtained (i.e. lab name), the calibration program and the protocol for quality assurance OR state that no new dates are provided. |

☐ Tick this box to confirm that the raw and calibrated dates are available in the paper or in Supplementary Information.

| Ethics oversight | Identify the organization(s) that approved or provided guidance on the study protocol, OR state that no ethical approval or guidance was required and explain why not. |
|---|---|

Note that full information on the approval of the study protocol must also be provided in the manuscript.

## Animals and other research organisms

Policy information about studies involving animals; ARRIVE guidelines recommended for reporting animal research, and Sex and Gender in Research

| Laboratory animals | For laboratory animals, report species, strain and age OR state that the study did not involve laboratory animals. |
|---|---|
| Wild animals | Provide details on animals observed in or captured in the field; report species and age where possible. Describe how animals were caught and transported and what happened to captive animals after the study (if killed, explain why and describe method; if released, say where and when) OR state that the study did not involve wild animals. |
| Reporting on sex | Indicate if findings apply to only one sex; describe whether sex was considered in study design, methods used for assigning sex. Provide data disaggregated for sex where this information has been collected in the source data as appropriate; provide overall numbers in this Reporting Summary. Please state if this information has not been collected. Report sex-based analyses where performed, justify reasons for lack of sex-based analysis. |
| Field-collected samples | For laboratory work with field-collected samples, describe all relevant parameters such as housing, maintenance, temperature, photoperiod and end-of-experiment protocol OR state that the study did not involve samples collected from the field. |
| Ethics oversight | Identify the organization(s) that approved or provided guidance on the study protocol, OR state that no ethical approval or guidance was required and explain why not. |

Note that full information on the approval of the study protocol must also be provided in the manuscript.

## Clinical data

Policy information about clinical studies
All manuscripts should comply with the ICMJE guidelines for publication of clinical research and a completed CONSORT checklist must be included with all submissions.

| Clinical trial registration | Provide the trial registration number from ClinicalTrials.gov or an equivalent agency. |
|---|---|
| Study protocol | Note where the full trial protocol can be accessed OR if not available, explain why. |
| Data collection | Describe the settings and locales of data collection, noting the time periods of recruitment and data collection. |
| Outcomes | Describe how you pre-defined primary and secondary outcome measures and how you assessed these measures. |

## Dual use research of concern

Policy information about dual use research of concern

### Hazards

Could the accidental, deliberate or reckless misuse of agents or technologies generated in the work, or the application of information presented in the manuscript, pose a threat to:

| No | Yes | |
|----|-----|---|
| ☐ | ☐ | Public health |
| ☐ | ☐ | National security |
| ☐ | ☐ | Crops and/or livestock |
| ☐ | ☐ | Ecosystems |
| ☐ | ☐ | Any other significant area |

## Experiments of concern

Does the work involve any of these experiments of concern:

| No | Yes | |
|----|-----|---|
| ☐ | ☐ | Demonstrate how to render a vaccine ineffective |
| ☐ | ☐ | Confer resistance to therapeutically useful antibiotics or antiviral agents |
| ☐ | ☐ | Enhance the virulence of a pathogen or render a nonpathogen virulent |
| ☐ | ☐ | Increase transmissibility of a pathogen |
| ☐ | ☐ | Alter the host range of a pathogen |
| ☐ | ☐ | Enable evasion of diagnostic/detection modalities |
| ☐ | ☐ | Enable the weaponization of a biological agent or toxin |
| ☐ | ☐ | Any other potentially harmful combination of experiments and agents |

# Plants

| | |
|---|---|
| Seed stocks | Not applicable |
| Novel plant genotypes | Not applicable |
| Authentication | Not applicable |

# ChIP-seq

## Data deposition

☐ Confirm that both raw and final processed data have been deposited in a public database such as GEO.

☐ Confirm that you have deposited or provided access to graph files (e.g. BED files) for the called peaks.

| | |
|---|---|
| Data access links<br>*May remain private before publication.* | *For "Initial submission" or "Revised version" documents, provide reviewer access links. For your "Final submission" document, provide a link to the deposited data.* |
| Files in database submission | *Provide a list of all files available in the database submission.* |
| Genome browser session<br>(e.g. UCSC) | *Provide a link to an anonymized genome browser session for "Initial submission" and "Revised version" documents only, to enable peer review. Write "no longer applicable" for "Final submission" documents.* |

## Methodology

| | |
|---|---|
| Replicates | *Describe the experimental replicates, specifying number, type and replicate agreement.* |
| Sequencing depth | *Describe the sequencing depth for each experiment, providing the total number of reads, uniquely mapped reads, length of reads and whether they were paired- or single-end.* |
| Antibodies | *Describe the antibodies used for the ChIP-seq experiments; as applicable, provide supplier name, catalog number, clone name, and lot number* |
| Peak calling parameters | *Specify the command line program and parameters used for read mapping and peak calling, including the ChIP, control and index files used.* |

| | |
|---|---|
| Data quality | *Describe the methods used to ensure data quality in full detail, including how many peaks are at FDR 5% and above 5-fold enrichment.* |
| Software | *Describe the software used to collect and analyze the ChIP-seq data. For custom code that has been deposited into a community repository, provide accession details.* |

## Flow Cytometry

### Plots

Confirm that:

☐ The axis labels state the marker and fluorochrome used (e.g. CD4-FITC).

☐ The axis scales are clearly visible. Include numbers along axes only for bottom left plot of group (a 'group' is an analysis of identical markers).

☐ All plots are contour plots with outliers or pseudocolor plots.

☐ A numerical value for number of cells or percentage (with statistics) is provided.

### Methodology

| | |
|---|---|
| Sample preparation | *Describe the sample preparation, detailing the biological source of the cells and any tissue processing steps used.* |
| Instrument | *Identify the instrument used for data collection, specifying make and model number.* |
| Software | *Describe the software used to collect and analyze the flow cytometry data. For custom code that has been deposited into a community repository, provide accession details.* |
| Cell population abundance | *Describe the abundance of the relevant cell populations within post-sort fractions, providing details on the purity of the samples and how it was determined.* |
| Gating strategy | *Describe the gating strategy used for all relevant experiments, specifying the preliminary FSC/SSC gates of the starting cell population, indicating where boundaries between "positive" and "negative" staining cell populations are defined.* |

☐ Tick this box to confirm that a figure exemplifying the gating strategy is provided in the Supplementary Information.

## Magnetic resonance imaging

### Experimental design

| | |
|---|---|
| Design type | *Indicate task or resting state; event-related or block design.* |
| Design specifications | *Specify the number of blocks, trials or experimental units per session and/or subject, and specify the length of each trial or block (if trials are blocked) and interval between trials.* |
| Behavioral performance measures | *State number and/or type of variables recorded (e.g. correct button press, response time) and what statistics were used to establish that the subjects were performing the task as expected (e.g. mean, range, and/or standard deviation across subjects).* |

### Acquisition

| | |
|---|---|
| Imaging type(s) | *Specify: functional, structural, diffusion, perfusion.* |
| Field strength | *Specify in Tesla* |
| Sequence & imaging parameters | *Specify the pulse sequence type (gradient echo, spin echo, etc.), imaging type (EPI, spiral, etc.), field of view, matrix size, slice thickness, orientation and TE/TR/flip angle.* |
| Area of acquisition | *State whether a whole brain scan was used OR define the area of acquisition, describing how the region was determined.* |

Diffusion MRI ☐ Used ☐ Not used

### Preprocessing

| | |
|---|---|
| Preprocessing software | *Provide detail on software version and revision number and on specific parameters (model/functions, brain extraction, segmentation, smoothing kernel size, etc.).* |
| Normalization | *If data were normalized/standardized, describe the approach(es): specify linear or non-linear and define image types used for transformation OR indicate that data were not normalized and explain rationale for lack of normalization.* |

| Normalization template | *Describe the template used for normalization/transformation, specifying subject space or group standardized space (e.g. original Talairach, MNI305, ICBM152) OR indicate that the data were not normalized.* |
| Noise and artifact removal | *Describe your procedure(s) for artifact and structured noise removal, specifying motion parameters, tissue signals and physiological signals (heart rate, respiration).* |
| Volume censoring | *Define your software and/or method and criteria for volume censoring, and state the extent of such censoring.* |

## Statistical modeling & inference

| Model type and settings | *Specify type (mass univariate, multivariate, RSA, predictive, etc.) and describe essential details of the model at the first and second levels (e.g. fixed, random or mixed effects; drift or auto-correlation).* |
| Effect(s) tested | *Define precise effect in terms of the task or stimulus conditions instead of psychological concepts and indicate whether ANOVA or factorial designs were used.* |

Specify type of analysis: ☐ Whole brain ☐ ROI-based ☐ Both

| Statistic type for inference<br><br>(See Eklund et al. 2016) | *Specify voxel-wise or cluster-wise and report all relevant parameters for cluster-wise methods.* |
| Correction | *Describe the type of correction and how it is obtained for multiple comparisons (e.g. FWE, FDR, permutation or Monte Carlo).* |

## Models & analysis

| n/a | Involved in the study |
| --- | --- |
| ☐ | ☐ Functional and/or effective connectivity |
| ☐ | ☐ Graph analysis |
| ☐ | ☐ Multivariate modeling or predictive analysis |

| Functional and/or effective connectivity | *Report the measures of dependence used and the model details (e.g. Pearson correlation, partial correlation, mutual information).* |
| Graph analysis | *Report the dependent variable and connectivity measure, specifying weighted graph or binarized graph, subject- or group-level, and the global and/or node summaries used (e.g. clustering coefficient, efficiency, etc.).* |
| Multivariate modeling and predictive analysis | *Specify independent variables, features extraction and dimension reduction, model, training and evaluation metrics.* |

