## [Peer Review File · Nature]

Single molecule dynamics of the TRiC chaperonin system in vivo

Corresponding Author: Professor F. Ulrich Hartl

Version 0:

Reviewer comments:

Referee #1

(Remarks to the Author)

Review of Li et al. manuscript:

The manuscript submitted by Li et al. is a tour de force in terms of innovative and powerful methods to follow chaperone actions in cellulose. The experimental design empowered the authors to follow how the chaperonin TRiC interacts with nascent polypeptides as they emerge from the ribosome but are still being translated (co-translational situation) and to explore the extent to which TRiC works in concert with its co-chaperone prefoldin (PFD). The results are compelling, with careful attention to controls and lovely exploitation of single particle tracking methods. New insights are provided about the length of time the chaperonin and its partner spend associated with nascent chains and with each other.

The authors deploy ribosome profiling to augment this portion of the study and identify the subproteome that associates with TRiC and/or PFD co-translationally. While these results are informative, no major conclusions emerge from this portion of the work.

The other major part of the study is the examination of post-translational interactions between TRiC and an obligate substrate of this chaperonin, actin. Again, this study required innovative approaches which are stunning and well-controlled. Tethering actin to the ribosome but with a linker that caused the actin to behave as though it had been fully translated and released was a very powerful approach. One can't help but wonder whether the folding of tethered actin and its binding to TRiC/PFD are the same as if the actin were free, but the controls support this conclusion. The exploration of a folding defective actin and its prolonged interactions with TRiC is also a very nice addition to the story.

So the real question for the authors is can they point out more clearly the implications of the new data they are able to obtain through their technological masterpiece? The way the paper is currently written places most emphasis on the novelty and potential usefulness of the creative methods. For sure, it constitutes a landmark in proteostasis mechanistic work in that up until this group's efforts, there has been heavy reliance on in vitro systems of study. So for sure this is super impactful simply based on the demonstration that one can do this! But apart from the "protective zone" conclusion, the rest of the findings are largely presented without discussion of their implications or mechanistic analysis. My fear is that many readers will not work through the very detailed and complex experimental design and will seek to know what was learned, in a clearer and more accessible fashion. For this reason, I recommend that the authors do a minor revision to bring out the main points they want a reader to take away. If these are focused on the innovation in methods, fine, but I believe there is more that the new data offer in the way of insights into the nature of the PFD/TRiC system. What was known before? How do these new in cellulose data compare to the in vitro work done previously on the PFD/TRiC system?

Minor points:

1. The abstract and title emphasize 'dynamics', which is a word with many meanings...

I think it would help for the authors to explain what they mean by that.

2. P. 3, line 69: it would be helpful to readers to know what the past work found about PFD impact on actin folding without asking them to go read the relevant paper(s).

3. P. 3, line 85: was the Halo tag also added by CRISPR/Cas?

4. The discussion about PFD and TRiC action on nascent chains is confusing. What does the interaction of PFD on its own do to nascent chains? How often is TRiC working with PFD?

5. P. 6: why is it notable that PFD binding to actin 305 was 4X more frequent than to actin 105? Explain

6. P. 8: the comment about PFD and TRiC action on co- or post- translational clients being the same is a significant

conclusion... bring it out more?

7. P. 10, lines 292-294: This is a reasonable speculation but not supported by this study... please make this clear. For example start the sentence with: "We hypothesize..." or similar wording.

8. Several abbreviations were not defined: RPL10A (can guess, but please define); HBSS, ... check for others please.

9. PFD3 KD is sometimes referred as PFD KD and that is confusing.

Referee #2

(Remarks to the Author)

I co-reviewed this manuscript with one of the reviewers who provided the listed reports.

Referee #3

(Remarks to the Author)

Review of "In vivo dynamics of the TRiC/CCT chaperonin system visualized by single particle tracking " by Li et al.

In this manuscript, Li et al. employ single-molecule methodologies to visualize binding of the TRiC/CCT and its co-chaperone PFD to nascent proteins in living cells. TRiC and PDF molecules are labeled using the Halo-tag and their recruitment dynamics to SNAP-tag labeled ribosomes, as well as MS2 and SunTag labeled actin mRNAs is investigated. The authors show that TRiC and PDF undergo many transient binding events to cellular nascent chains, and that TRiC is dependent on PFD for efficient recruitment. The co and post-translational recruitment of the chaperonin system to actin is next studied in more detail, and the authors find that TRiC and PDF interaction lifetimes are substantially increased in the very late stages of actin translation. In addition, TRiC interaction lifetimes and the number of PDF binding events are substantially increased when a folding-deficient actin mutant is interrogated. Finally, data is presented that PDF and TRiC can undergo multiple successive binding events to the same client protein without diffusing into the bulk cytosol, suggesting that the cellular chaperonin system may form functional compartments to facilitate repeated folding attempts of proteins. This is a technologically highly impressive study that provides the first insights into the dynamics of TRiC and PFD. Overall, the study is well-executed and quantified and the conclusions are largely supported by the findings. I should say that the manuscript was not easy to read and I sometimes found it hard to follow the logic. Before publication, a number of important issues need to be resolved:

1. The authors need to include statistical analysis of all their data, many currently lack a test of significance. In addition, the Y-axis should start at 0 in all graphs. Starting the axis at other values to make differences between samples look artificially large is not appropriate.

2. The authors conclude that PFD binds slightly more frequently to NCs than TRiC (Fig. 1e and lines 106-108). However, this observation depends heavily on the assumption that an equal number of PFD and TRiC molecules have been labeled across the two experimental conditions that are compared. The authors claim that they label 10% of TRiC and 30% of PFD and refer to extended data figure 1j. I assume they use the concentration indicated by the arrowhead (although this is not explained in the figure legends). I'm also not sure I understand how they conclude that these dye concentrations represent 10% and 30%. I guess they conclude that at these concentrations the total fluorescence concentrations is 10% and 30% of the max, but at the same time, the intensity values plateau between 10E2 and 10E3, suggesting that the fluorescence observed here is autofluorescence or background, rather than JF specific signal (in which case intensities should change upon a 10 fold reduction in dye concentration). It's important to determine the precise fraction of molecules that is labeled.

3. How long do the authors treat cells with puromycin before imaging? This treatment should be very brief (a few minutes, certainly not more than 30 min) to ensure one is looking at direct effects of nascent chain release, rather than stress-induced secondary effects of puromycin.

4. If PFD acts as a recruitment factor for TRiC, one would expect that TRiC interactions to occur mostly on nascent chains that also interact with PFD. Moreover, the interactions of TRiC should occur at the same site or downstream (more 3' or C-terminal) to the site of PFD-nascent chain interactions. Was this ever observed?

5. "Similar numbers of ribosome-engaging mRNA molecules were detected in the cytosol for each construct by SPT (Extended Data Fig. 5a)." There is a very small change in diffusion values for these constructs, and I'm not sure how confident one can be that one can conclude the statement above from these data.

6. I don't understand this sentence: "Post-translational folding of actin, which has a half-time of ~50 s, is thought to involve multiple cycles of protein encapsulation in the TRiC cavity. Indeed, the near full-length actin368 was not recognized by DNase-I (Extended Data Fig. 6a), which binds specifically and with high-affinity (Kd ~2 nM) to a loop region in SD2 of folded actin" How does lack of binding to DNase-I prove that post-translational folding of actin involves multiple cycles of protein encapsulation?

7. Fig. 1j shows that the number of PFD binding events to actin mRNA increases by ~4-fold as the coding sequence length is changed from 101 to 305 codons. Does the increased binding frequency not merely reflect the longer dwell time of exposed nascent chains on the mRNA, as ribosomes take more time translating a CDS of 305 codons than a CDS of 101 codons?

8. When assessing PFD-TRiC interactions, the authors should use a negative control, ideally of mutant proteins that don't interact, or, alternatively of similar sized proteins (to have similar diffusion kinetics) to determine the frequency of co-incident co-localization. This is needed to determine the actual interaction frequency.

9. In Figure 2k, the authors show that upon expression of the actin-Sun reporter, cells display a ~3-fold increase in cellular PFD-TRiC association events. How do the authors explain this striking result? As the actin-sun mRNA presumably constitutes only a small fraction of the cellular mRNA pool, how can its expression change PFD-TRiC association events to such a big extent?

10. When examining post-translation interactions of PFD and TRiC with actin using the Xbp1 sequence, did the authors determine whether either binds to the SunTag sequence of Xbp1 sequence and normalize their data for such binding?

11. Figure 4a-f are just examples. One can always find an example molecule that shows some sort of behavior. I would like to see quantification and statistics to prove that local retention of TRiC/PFD is indeed occurring. Similarly, I would like to see quantification of the so-called stationary molecules that are observed.

12. Was the local retention of TRiC and PFD observed in specific regions of the cell or does this phenomenon occur equally throughout the cell? I don't understand the proposed model of a 'protective folding compartment'. A typical human cell has ~5 million ribosomes, are there 5 million local folding compartments, one around each ribosome? And ribosomes diffuse rapidly, are the folding compartments moving along?

13. I did not understand how the experiments in Fig 4h-l explain the local retention of TRiC molecules near clients. The TRiC molecule moves up to 1.5 micron away from the client, I assume such a distance cannot be bridged by a multimer of TRiC. This section is difficult to understand the logic.

Minor comments:

- The descriptions for figure panels m and n in Extended Data Fig. 1 are missing
- Line 140: PFD KD reduced TRiC binding to actin NCs by ~70% (Fig. 1g and Extended Data Fig. 4h). How was this number calculated? The fold-change presented in the Fig. 1g does not appear to be 70%.
- Line 171: ... in agreement with ribosome profiling data (Extended data Fig. 5g). There is no Extended data Fig. 5g. Was 5e meant instead?
- Sun-tag is typically written as SunTag in the literature
- In extended data figure 8e, the authors measure the distance between nascent chains and the 3'UTR labeled by MS2. However, in this setup, individual mRNAs are likely bound by multiple ribosomes, each with a nascent chain. I assume they cannot resolve individual nascent chains and are thus measuring the distance between the average nascent chain the MS2 signal. It is possible that some nascent chains, especially those at the 5' end of the mRNA are further away from the MS2 signal.

Referee #4

(Remarks to the Author)

I co-reviewed this manuscript with one of the reviewers who provided the listed reports.

Referee #5

(Remarks to the Author)

In this manuscript, the authors investigated the interaction dynamics of the chaperonin TRiC and its co-chaperone, prefoldin (PFD), with substrates in living cells. They employed cutting-edge real-time single particle tracking (SPT) technology to do so. Using SPT, they monitored the co-translational interactions of TRiC and PFD with nascent polypeptides and full-length substrates. They demonstrated that both chaperones interact co-translationally with a variety of nascent chains (NCs) in multiple binding events, which typically last for approximately 1 second. PFD was found to function by recruiting TRiC. Using actin as an obligate substrate, they found that the co-translational interactions of PFD and TRiC increased in frequency and duration during nascent chain elongation. Close to translation termination, PFD remained bound for several seconds, facilitating TRiC recruitment for post-translational folding. This process involved multiple reaction cycles of approximately 2.5 seconds each.

Using a folding-defective actin mutant, the authors showed that the lifetime of TRiC interactions was up to four times longer, suggesting that the substrate's conformational state modulates TRiC function. Interestingly, the authors discovered that TRiC remained confined near its substrates between binding cycles. This indicates that the chaperonin system operates within a localised 'protective zone', suggesting supra-molecular organisation of the chaperonin system.

A plethora of important research has been conducted on the TRiC and PFD systems in the past; however, this has predominantly been in vitro. In this manuscript, the authors provide the first in vivo analysis of the PFD/TRiC chaperonin system and its clients, mainly by determining single-molecule dynamics. To this end, they employed highly sophisticated

microscopy techniques and intelligent labelling methods for target molecules. This study provides new insights into the lifetime of chaperonin-client interactions *in vivo*, the timing and frequency of TRiC binding to nascent chains during translation and how chaperonin dynamics are affected by folding-defective substrates. All of the experiments described here are technically sound and well controlled. Of course, the effects of artificial labels on proteins and mRNA could never be fully excluded. It would be interesting to study the potential supramolecular organisation of the chaperonin machinery and gain some initial insight into how this might be achieved, for example, by the mRNA encoding TRiC substrates. However, it is clear that such an analysis is beyond the scope of this study. Overall, this is an excellent piece of work and I support publication in the Nature journal.

Minor points:

- a) What about the 20% of substrates that are only bound to PDF co-translationally? Are there any specific determinants for nascent chains that only interact with PDF?
- b) The authors expressed the G150P mutant of actin that occupies TRiC for longer and thus might inhibit chaperonin activity. Did the authors observe any phenotypic changes in cells expressing the mutant form, e.g. a heat shock response or cell death? More generally, how do these multiple labels affect cell viability and growth?

Version 1:

Reviewer comments:

Referee #1

(Remarks to the Author)

In their revised manuscript, Li et al. have adequately addressed the concerns we raised about the earlier version, with one exception, described below. Moreover, we were gratified to read the extremely careful and technically expert review offered by Reviewer 3. It seems that this led the authors to carry out further analyses and some experiments that strengthen the results of their study. Compliments to this reviewer and to the authors!

The one point that we remain concerned about is the use of the term “dynamics”. In order to persuade the authors that this term may confuse the reader and also does not correctly convey the results they present in this project, we did some “homework” using an AI search to check whether our understanding of this term is generally acknowledged. We attribute a meaning to “dynamics” that relates to molecular motions (from side chains to domains) and conformational changes among different conformational states. By contrast, the term that describes the timing of binding events, including on/off rates and residence times, is “kinetics”. We feel strongly that the work in this paper is “kinetics”, not “dynamics”. We cite below the AI retrieved definitions and references that support them. To make their title more correct, the authors could state “The timing of TRiC chaperone action *in vivo* by single molecule particle tracking” or a variation on this.

From an AI search:

1. Kinetics = Rates of binding and unbinding

Kinetics describes how fast binding and dissociation happen.

• It's about rate constants:

o k_{on} (association rate) → how quickly two molecules find and bind each other.

o k_{off} (dissociation rate) → how quickly the complex falls apart.

• The equilibrium dissociation constant, $KD = k_{off}/k_{on}$, gives the overall binding affinity.

• Kinetics focuses on measurable rates and time scales — what you'd see in a SPR (surface plasmon resonance) or stopped-flow experiment.

Example:

If Protein A binds Protein B with a high k_{on} and low k_{off} , it binds quickly and stays bound — strong and stable interaction.

2. Dynamics = Motions and conformational changes

Dynamics describes the structural motions and flexibility that occur before, during, and after binding.

• It's about how proteins move and fluctuate over time (e.g., side-chain rearrangements, loop flexibility, domain motions).

• Protein dynamics influence how accessible binding sites are and how the binding interface adapts.

• Studied with techniques like NMR relaxation, molecular dynamics simulations, or single-molecule FRET.

Example:

A protein may need to “breathe” (open a loop) to expose its binding site — this conformational flexibility affects how and when binding occurs, but not directly the rate constants themselves.

In short

Concept Focus Key Quantities Typical Methods Biological Meaning

Kinetics How fast binding/unbinding occurs k_{on} , k_{off} , K_D SPR, stopped-flow, ITC Reaction rates, affinity

Dynamics How proteins move and fluctuate Conformational ensembles, time correlations NMR, MD simulations, smFRET

Structural flexibility, induced fit, allostery

Some expert references on this distinction:

1. Protein Binding Kinetics

These works describe the mathematical and experimental basis of association (k_{on}) and dissociation (k_{off}) rates:

- Copeland, R. A. (2016). The drug–target residence time model: a 10-year retrospective. *Nature Reviews Drug Discovery*, 15, 87–95.

Defines kinetic parameters (k_{on} , k_{off} , K_D) and explains how binding kinetics govern biological efficacy.

• Swinney, D. C. (2004). Biochemical mechanisms of drug action: what does it take for success? *Nature Reviews Drug Discovery*, 3, 801–808.

Clear description of kinetic parameters in ligand–protein interactions.

• Schreiber, G., & Keating, A. E. (2011). Protein binding specificity versus promiscuity. *Current Opinion in Structural Biology*, 21, 50–61.

Reviews how kinetic rate constants underlie specificity and affinity in protein–protein interactions.

🔄 2. Protein Dynamics and Conformational Flexibility

These sources describe motions, conformational ensembles, and structural transitions that underlie binding:

• Henzler-Wildman, K. A., & Kern, D. (2007). Dynamic personalities of proteins. *Nature*, 450, 964–972.

Classic review on protein dynamics across timescales and how these motions enable function, including ligand binding.

• Boehr, D. D., Nussinov, R., & Wright, P. E. (2009). The role of dynamic conformational ensembles in biomolecular recognition. *Nature Chemical Biology*, 5, 789–796.

Explains how protein dynamics affect recognition and binding, introducing “conformational selection” and “induced fit” models.

• Nussinov, R., Tsai, C.-J., & Ma, B. (2013). The key role of protein dynamics in allosteric effects and drug discovery. *Nature Chemical Biology*, 9, 693–694.

Highlights how conformational flexibility governs binding and allostery.

🔗 3. Linking Kinetics and Dynamics

These discuss how dynamics modulate kinetics, bridging both concepts:

• Hammes, G. G., Chang, Y.-C., & Oas, T. G. (2009). Conformational selection or induced fit: a flux description of reaction mechanism. *PNAS*, 106, 13737–13741.

Quantitative model linking conformational dynamics to observed binding kinetics.

• Changeux, J.-P., & Edelstein, S. J. (2011). Conformational selection or induced fit? 50 years of debate resolved. *FASEB Journal*, 25, 1169–1175.

Integrates the two views, showing how intrinsic dynamics influence kinetic rate constants.

✓ Summary from Sources

Together, these establish that:

• Kinetics quantifies the rates of binding/unbinding via measurable constants (Copeland 2016; Schreiber & Keating 2011).

• Dynamics describes structural motions and conformational ensembles that make binding possible (Henzler-Wildman & Kern 2007; Boehr et al. 2009).

• The two connect via conformational selection or induced fit models (Hammes et al. 2009; Changeux & Edelstein 2011).

Referee #2

(Remarks to the Author)

I co-reviewed this manuscript with one of the reviewers who provided the listed reports.

Referee #3

(Remarks to the Author)

Point 4: The fact that TRiC frequently (33%) binds before PFD is not consistent with a simple ‘recruitment’ model. Perhaps PFD can act as a recruitment factor in some cases, but additionally it could stabilize transient binding events of TRiC with the RNA (which I wouldn’t call ‘recruitment’). It would be nice if the authors can specify this in the statement on lines 130–131.

Point 7: From the data shown in Fig. 1j, it is evident that increasing the length of the actin coding sequence results in a corresponding increase in the dwell time of PFD. However, the observed increase in the number of interaction events (Fig. 1j, upper panel) can be interpreted differently: The translation of actin 305, for example, would take approximately three times longer than that of actin 101, thereby providing more time for PFD to bind to the actin nascent actin and co-localize with the labeled mRNA as the coding sequence lengthens. In addition, the ribosome exit tunnel accommodates roughly 30–40 amino acids, which prevents nascent actin from being exposed during the early stages of translation. Taken together, these considerations suggest that the 4-fold increase in PFD interaction events between actin 101 and actin 305 can be simply explained by the prolonged availability of nascent actin chains on the mRNA, rather than by a size-dependent increase in the PFD on-rate for individual nascent chains. The same logic also applies to TRiC’s interaction with the growing nascent chain (Fig. 1k).

Point 11: “To exclude this effect, we only considered TRiC trajectories in which TRiC remained in focus for at least 500 ms (10 frames) after detaching from substrate (actin nascent chain or completely translated actin with SunTag-Xbp1u+), thus allowing re-binding to be observed.”

I feel that these analysis criteria will overestimate the number of events called as ‘local retention’, as most events of local retention will be included in this analysis (as they remain local, they are more likely to remain in the z-plane of imaging), but the majority of non-local retention events will be discarded (as in the latter case, the molecule is far more likely to leave the z-plane of imaging). Thus, this analysis likely (greatly) overestimates the number of events of local retention. It would be good if the authors can show the same analysis, but without the inclusion criteria of spots remaining with the imaging plane for a

set number of time-points.

Point 12: The average polysome contains perhaps 3-7 ribosomes, so that would still leave close to 1 million polysomes per cell. To get to the value of several thousand folding hubs, on average dozens to hundreds of polysomes would need to cluster together in each of these folding hubs. Such clusters would be very large, and show very slow diffusion. Rather, what is observed in this study, as well as many previous studies is that diffusion of typical mRNAs is quite fast (relative to the expected diffusion of cluster of many polysomes that form a hub), for example see PMID: 26760529. If association of mRNAs with folding hubs would be translation dependent, then puromycin treatment should dramatically increase diffusion rates, by several orders of magnitude (as mRNAs would go from being associated with hubs of dozens of mRNAs, ribosomes and co-factors, to single mRNAs). In reality the diffusion speed difference is much smaller.

Perhaps such hubs exist for a (subset of) polysomes, but based on the current data, I don't think it is likely that most, or even many polysomes are part of such hubs. I would suggest the authors put the existence of folding hubs into this context. Additionally, the authors actually do not address my primary question, whether the 'hubs' are observed in a specific region of the cell, for example perinuclear, a region where more ribosomes are likely engaged with the ER.

Referee #4

(Remarks to the Author)

I co-reviewed this manuscript with one of the reviewers who provided the listed reports.

Referee #5

(Remarks to the Author)

The authors have carefully addressed all concerns raised. This is an excellent piece of work, and I strongly recommend it for publication in Nature.

Version 2:

Reviewer comments:

Referee #1

(Remarks to the Author)

The authors' responses are satisfactory and the revised manuscript is exciting and should be published.

Referee #2

(Remarks to the Author)

I co-reviewed this manuscript with one of the reviewers who provided the listed reports.

Referee #3

(Remarks to the Author)

The following sentence could benefit from some restructuring and additional explanation regarding how the restricted TIRF illumination influenced the analysis.

"While this On-Off-On binding behaviour was observed in only ~12-18% of all recorded co- and post-interaction trajectories, due to the limited depth of TIRF illumination, it was observed for ~30-40% of TRiC complexes that could be tracked long enough in the Off-state to detect rebinding (Extended Data Fig. 8g)"

Otherwise, the authors have fully addressed all of my points and I support publication of this manuscript in Nature.

Referee #4

(Remarks to the Author)

I co-reviewed this manuscript with one of the reviewers who provided the listed reports.

Response to reviewers

We thank the reviewers for their very encouraging comments on our manuscript and for providing helpful advice for improvements. In revising the manuscript, we have added new data and clarified various aspects in the text. However, we ask the reviewers to take into account that we also had to shorten the text, which prevented a more extensive discussion in some places.

Reviewer #1

Review of Li and Dalheimer et al. manuscript:

The manuscript submitted by Li et al. is a tour de force in terms of innovative and powerful methods to follow chaperone actions in cellulose. The experimental design empowered the authors to follow how the chaperonin TRiC interacts with nascent polypeptides as they emerge from the ribosome but are still being translated (co-translational situation) and to explore the extent to which TRiC works in concert with its co-chaperone prefoldin (PFD). The results are compelling, with careful attention to controls and lovely exploitation of single particle tracking methods. New insights are provided about the length of time the chaperonin and its partner spend associated with nascent chains and with each other. The authors deploy ribosome profiling to augment this portion of the study and identify the subproteome that associates with TRiC and/or PFD co-translationally. While these results are informative, no major conclusions emerge from this portion of the work.

The other major part of the study is the examination of post-translational interactions between TRiC and an obligate substrate of this chaperonin, actin. Again, this study required innovative approaches which are stunning and well-controlled. Tethering actin to the ribosome but with a linker that caused the actin to behave as though it had been fully translated and released was a very powerful approach. One can't help but wonder whether the folding of tethered actin and its binding to TRiC/PFD are the same as if the actin were free, but the controls support this conclusion. The exploration of a folding defective actin and its prolonged interactions with TRiC is also a very nice addition to the story.

So the real question for the authors is can they point out more clearly the implications of the new data they are able to obtain through their technological masterpiece? The way the paper is currently written places most emphasis on the novelty and potential usefulness of the creative methods. For sure, it constitutes a landmark in proteostasis mechanistic work in that up until this group's efforts, there has been heavy reliance on in vitro systems of study. So for sure this is super impactful simply based on the demonstration that one can do this! But apart from the "protective zone" conclusion, the rest of the findings are largely presented without discussion of their implications or mechanistic analysis. My fear is that many readers will not work through the very detailed and complex experimental design and

will seek to know what was learned, in a clearer and more accessible fashion. For this reason, I recommend that the authors do a minor revision to bring out the main points they want a reader to take away. If these are focused on the innovation in methods, fine, but I believe there is more that the new data offer in the way of insights into the nature of the PFD/TRiC system. What was known before? How do these new in cellulo data compare to the in vitro work done previously on the PFD/TRiC system?

Following the reviewer's suggestion, we have revised the text in several places in order to more clearly state the mechanistic implications of our findings, and also to emphasize the adaptability of the methods we developed to the analysis of different chaperone systems and client proteins. See lines 47-49, 302-303.

Minor points:

1. The abstract and title emphasize 'dynamics', which is a word with many meanings... I think it would help for the authors to explain what they mean by that.

Thank you for this insightful comment. In our manuscript, the term "dynamics" specifically refers to the residence time (or dwell time) of TRiC/PFD in complex with substrate proteins, both co- and post-translationally, as well as to the frequency of these interactions observed by single particle tracking. The distribution of dwell times allows the estimation of off-rates and the frequency of interactions approximately correlates with the on-rate of binding. Thus, the two parameters provide insight into the affinity of the interactions. For example, if both dwell time and frequency of interactions increase substantially, as is the case for PFD and TRiC binding to nascent actin chains of increasing length (Fig. 1j,k), this would reflect a strong increase in overall affinity. Moreover, in the context of single particle tracking, "dynamics" also refers to the diffusion properties and trajectories of TRiC/PFD and their clients in live cells, thereby elucidating their spatiotemporal behavior.

We slightly changed the title of the manuscript, but prefer to keep the term 'dynamics'. To clarify its intended meaning, we added an explanatory statement in the introduction (line 46-49).

2. P. 3, line 69: it would be helpful to readers to know what the past work found about PFD impact on actin folding without asking them to go read the relevant paper(s).

We thank the reviewer for this comment. We have expanded on the effect of PFD on actin folding in line 64-67, which now reads: "PFD, a hetero-hexameric complex of coiled-coil subunits¹ (Extended Data Fig. 1b), is thought to deliver client proteins into the TRiC chamber² and is required for efficient actin folding, enhancing both the rate and yield of TRiC-mediated actin folding³".

3. P. 3, line 85: was the Halo tag also added by CRISPR/Cas?

The Halo-Tag was added to PFD4 using CRISPR/Cas. We have revised the text to clarify this (line 82-84).

4. The discussion about PFD and TRiC action on nascent chains is confusing. What does the interaction of PFD on its own do to nascent chains? How often is TRiC working with PFD?

Regarding the reviewer's first question: *What does the interaction of PFD on its own do to nascent chains?* As previously described², PFD recognizes exposed hydrophobic regions in nascent chains and likely acts as a holding chaperone that recruits TRiC for either co- or post-translational folding. In the case of actin, which has a large post-translational folding component, both the duration of PFD binding to the nascent protein and the frequency of interactions are substantially enhanced towards translation termination, as observed by single particle tracking in Fig.1j and supported by ribosome profiling data (Extended Fig. 5f). These binding parameters were previously unknown. The enhanced binding towards translation termination presumably serves to stabilize the near full-length actin and recruits TRiC for efficient post-translational completion of folding. We have made changes in the Discussion to make this clearer.

Regarding the reviewer's second question: *How often is TRiC working with PFD?* Depletion of PFD strongly diminished (by 66%) the interactions of TRiC with nascent chains and by 34% the post-translational interactions with actin (Fig.1g and Fig.2d). Our experiments adding puromycin to stop translation and release nascent chains from ribosomes show that PFD and TRiC interact frequently, particularly after translation, as only the co-translational interactions (occurring on the nascent chain) are abolished by puromycin, but not the post-translational interactions. Note that the frequency of post-translational interactions in Fig. 2k was overestimated due to a formatting error, which we corrected. The decrease in PFD-TRiC interaction frequency upon Puro treatment is only ~25%, not 50%. Furthermore, ribosome profiling data (Extended Data Fig. 3f) show that the majority of TRiC substrates (78%) are also PFD substrates, with the first binding position of PFD on the nascent chain preferentially preceding TRiC binding (see new Extended Data Fig. 3h). Note that ribosome profiling data do not indicate whether a chaperone interaction is productive or not. Thus, we conclude that PFD and TRiC cooperate broadly in the folding of a wide range of substrates. However, based on the ribosome profiling data (Extended Data Fig. 3f), PFD also interacts with nascent chains for which we detected no significant binding to TRiC, suggesting a role in proteostasis beyond its function as a co-chaperone of TRiC. We have clarified these points in the revised manuscript.

5. P. 6: why is it notable that PFD binding to actin 305 was 4X more frequent than to actin 105? Explain

We thank the reviewer for pointing this out. We believe this observation is notable and important because it highlights the length-dependence of PFD–nascent chain interactions. Specifically, the more frequent binding of PFD to actin 305 suggests that PFD has stronger affinity for actin chains that expose additional subdomains (SD2 and SD4) compared to the shorter actin 101. This suggests that PFD may not only function to deliver nascent chains to TRiC, but also plays a critical role in maintaining substrate solubility and preventing premature aggregation, thereby facilitating subsequent TRiC-mediated folding. We have clarified this in the text (line 306-313).

6. P. 8: the comment about PFD and TRiC action on co- or post- translational clients being the same is a significant conclusion... bring it out more?

We thank the reviewer for this comment. Indeed, previous studies have mostly focused on the role of PFD to recruit TRiC to substrate proteins for post-translational folding. Our data show that this function initiates co-translationally. As demonstrated specifically with actin, PFD recruits TRiC to nascent actin, especially to longer chains. Interestingly, the co- and post-translational interaction lifetimes for the interactions of PFD with TRiC are similar. This is mentioned in the text (line 220-221).

7. P. 10, lines 292-294: This is a reasonable speculation but not supported by this study... please make this clear. For example start the sentence with: “We hypothesize...” or similar wording.

Thank you. We modified the statement accordingly (line 288-289).

8. Several abbreviations were not defined: RPL10A (can guess, but please define); HBSS, ... check for others please.

We thank the reviewer for pointing this out. We have changed the text accordingly and ensured that all abbreviations are defined when first mentioned, including RPL10A and HBSS (line 79, line 547).

9. PFD3 KD is sometimes referred as PFD KD and that is confusing.

To disrupt the function of the entire prefoldin complex, we targeted its central α -subunit, PFD3, for knockdown. PFD3 is critical for assembly of the PFD complex. We have revised the text to clarify that knockdown of PFD3 leads to downregulation of the entire prefoldin complex, as demonstrated by total proteome analysis (Extended Data Fig. 2c) (line 100-101). Accordingly, we prefer to refer to this condition as PFD KD rather than PFD3 KD throughout the manuscript (except in Extended Data Fig. 2). This is now explicitly stated (line line100-101).

Referee #2:

In this manuscript, the authors investigated the interaction dynamics of the chaperonin TRiC and its co-chaperone, prefoldin (PFD), with substrates in living cells. They employed cutting-edge real-time single particle tracking (SPT) technology to do so. Using SPT, they monitored the co-translational interactions of TRiC and PFD with nascent polypeptides and full-length substrates. They demonstrated that both chaperones interact co-translationally with a variety of nascent chains (NCs) in multiple binding events, which typically last for approximately 1 second. PFD was found to function by recruiting TRiC.

Using actin as an obligate substrate, they found that the co-translational interactions of PFD and TRiC increased in frequency and duration during nascent chain elongation. Close to translation termination, PFD remained bound for several seconds, facilitating TRiC recruitment for post-translational folding. This process involved multiple reaction cycles of approximately 2.5 seconds each.

Using a folding-defective actin mutant, the authors showed that the lifetime of TRiC interactions was up to four times longer, suggesting that the substrate's conformational state modulates TRiC function. Interestingly, the authors discovered that TRiC remained confined near its substrates between binding cycles. This indicates that the chaperonin system operates within a localised 'protective zone', suggesting supra-molecular organisation of the chaperonin system.

A plethora of important research has been conducted on the TRiC and PFD systems in the past; however, this has predominantly been in vitro. In this manuscript, the authors provide the first in vivo analysis of the PDF/TRiC chaperonin system and its clients, mainly by determining single-molecule dynamics. To this end, they employed highly sophisticated microscopy techniques and intelligent labelling methods for target molecules. This study provides new insights into the lifetime of chaperonin-client interactions in vivo, the timing and frequency of TRiC binding to nascent chains during translation and how chaperonin dynamics are affected by folding-defective substrates. All of the experiments described here are technically sound and well controlled. Of course, the effects of artificial labels on proteins and mRNA could never be fully excluded. It would be interesting to study the potential supramolecular organisation of the chaperonin machinery and gain some initial insight into how this might be achieved, for example, by the mRNA encoding TRiC substrates. However, it is clear that such an analysis is beyond the scope of this study. Overall, this is an excellent piece of work and I support publication in the Nature journal.

Minor points:

a) What about the 20% of substrates that are only bound to PDF co-translationally? Are there any specific determinants for nascent chains that only interact with PDF?

Domain and fold annotation (InterPro, Pfam) of PFD-only substrates did not reveal a clear enrichment pattern of specific domains or protein families. Among the 20 most differentially represented families, the only apparent change was a relative depletion of the histone fold (PF00125). Amino acid composition analysis further showed that TRiC-only substrates are relatively enriched in leucine and serine residues compared to PFD-only substrates, albeit the observed effects are modest. Unfortunately, the format of this manuscript does not allow an in-depth discussion of these findings. We use ribosome profiling mainly as an orthogonal method to validate some of the conclusions from single particle tracking.

b) The authors expressed the G150P mutant of actin that occupies TRiC for longer and thus might inhibit chaperonin activity. Did the authors observe any phenotypic changes in cells expressing the mutant form, e.g. a heat shock response or cell death? More generally, how do these multiple labels affect cell viability and growth?

The reviewer raises an interesting point. In our experiments the actin G150P mutant was transiently expressed for 24 h and we focused mainly on its interactions with PFD/TRiC. Expression of the mutant actin, as performed here, had no apparent effect on cell viability. We do not rule out that toxicity may arise after longer expression. More generally, we carefully controlled that replacement of PFD/TRiC with Halo-tagged versions did not affect cell viability, indicating that the tagged chaperones are functional (see Extended Data Fig. 1d,f).

Referee #3:

Review of "In vivo dynamics of the TRiC/CCT chaperonin system visualized by single particle tracking" by Li et al.

In this manuscript, Li et al. employ single-molecule methodologies to visualize binding of the TRiC/CCT and its co-chaperone PFD to nascent proteins in living cells. TRiC and PFD molecules are labeled using the Halo-tag and their recruitment dynamics to SNAP-tag labeled ribosomes, as well as MS2 and SunTag labeled actin mRNAs is investigated. The authors show that TRiC and PFD undergo many transient binding events to cellular nascent chains, and that TRiC is dependent on PFD for efficient recruitment. The co and post-translational recruitment of the chaperonin system to actin is next studied in more detail, and the authors find that TRiC and PFD interaction lifetimes are substantially increased in the very late stages of actin translation. In addition, TRiC interaction lifetimes and the number of PFD binding events are substantially increased when a folding-deficient actin mutant is interrogated. Finally, data is presented that PFD and TRiC can undergo multiple successive binding events to the same client protein without diffusing into the bulk cytosol,

suggesting that the cellular chaperonin system may form functional compartments to facilitate repeated folding attempts of proteins.

This is a technologically highly impressive study that provides the first insights into the dynamics of TRiC and PFD. Overall, the study is well-executed and quantified and the conclusions are largely supported by the findings. I should say that the manuscript was not easy to read and I sometimes found it hard to follow the logic. Before publication, a number of important issues need to be resolved:

1. The authors need to include statistical analysis of all their data, many currently lack a test of significance. In addition, the Y-axis should start at 0 in all graphs. Starting the axis at other values to make differences between samples look artificially large is not appropriate. We thank the reviewer for these comments. We were initially hesitant to use significance tests that were not robust against data from interaction lifetime measurements. The lifetime data are generally not normally distributed. Moreover, after puromycin treatment or PFD knockdown in particular, the number of detected interaction events is markedly reduced for the same number of cells, highlighting the strong phenotypes we observed for different conditions.

Nevertheless, we agree that including statistical analysis is important. We have now used the Welch's ANOVA test, which remains reliable even when sample sizes are unequal and not normally distributed, as long as sample sizes are greater than 15 per group^{4,5} (the smallest data set in our sample is exactly 15). The statistical analyses (Fig. 1d,g,j,k; Fig. 2d,j; Fig. 3b,e; Fig. 4h) support our original conclusions. We also note that statistical analysis of interaction frequency data was already included in the original manuscript.

Regarding the concern that Y-axis values do not start at 0 in graphs showing interaction lifetimes, thus possibly exaggerating effect size. Our intention was the opposite: We displayed the Y-axis on a **log₂ scale** to emphasize the high density of short lifetimes (0.5–4 s), while the plotted values themselves represent the **original, non-transformed data**. This approach compresses the outliers and makes differences among the majority of events more apparent. Since logarithmic functions are only defined for positive values, setting the Y-axis to start at 0 is not feasible. We have clarified this point in the figure legends to avoid any confusion.

In addition, in all panels showing interaction lifetimes, the Y-axis is adjusted to match the lower limit of the analyzed range, as all plotted data correspond to events ≥ 500 ms, a predefined cutoff described in the Methods section to exclude non-specific interactions. Including the range below 500 ms would only create an empty region that visually compresses the plotted distribution and obscures meaningful variation within the

observed data range. We therefore consider starting the axis near the lowest observed value to be the most appropriate representation.

2. The authors conclude that PFD binds slightly more frequently to NCs than TRiC (Fig. 1e and lines 106-108). However, this observation depends heavily on the assumption that an equal number of PFD and TRiC molecules have been labeled across the two experimental conditions that are compared. The authors claim that they label 10% of TRiC and 30% of PFD and refer to extended data figure 1j. I assume they use the concentration indicated by the arrowhead (although this is not explained in the figure legends). I'm also not sure I understand how they conclude that these dye concentrations represent 10% and 30%. I guess they conclude that at these concentrations the total fluorescence concentrations is 10% and 30% of the max, but at the same time, the intensity values plateau between $10E2$ and $10E3$, suggesting that the fluorescence observed here is autofluorescence or background, rather than JF specific signal (in which case intensities should change upon a 10 fold reduction in dye concentration). It's important to determine the precise fraction of molecules that is labeled.

We thank the reviewer for raising this point, which led us to improve the quantification of labeling efficiencies. As the reviewer pointed out, the fluorescence observed from the low dye concentrations may not be well distinguished from the autofluorescence or background due to the fact that low laser intensity has been applied to avoid saturation at the high concentrations. To quantify the fraction of labeled molecules, we now replaced the original binding curve and instead performed a fluorescence in-gel assay with the respective concentrations used for single particle tracking of TRiC-Halo and PFD-Halo, as well as concentrations that allow saturation of binding (see below and new Extended Data Fig. 1j,k). Consistent with our earlier conclusion, we indeed labeled roughly 3-fold more PFD (relative to total) than TRiC. As PFD is about 3-times less abundant than TRiC⁶, this ensures comparable numbers of labeled TRiC and PFD complexes in our experiments (see below). Notably, the fraction of labeled molecules is $\sim 3.4 \pm 0.4\%$ for TRiC-Halo and $\sim 8.5 \pm 2.5\%$ for PFD-Halo, based on the fluorescence in-gel assay, indicating that we indeed overestimated the fraction labeled based on the fluorescence saturation curve. Thank you for helping us to make this correction.

To directly determine the numbers of labeled molecules in the relevant experimental setting, we counted the number of labeled TRiC and PFD molecules detected in the first frame of the video used for data analysis. As shown below and in the new Extended Data Fig. 1l, the numbers of labeled TRiC and PFD molecules are very similar, confirming the biochemical analysis. Formally, this analysis of particle numbers alone is sufficient to provide the basis for our conclusions regarding relative frequency of TRiC and PFD binding, independent of the biochemical analysis of labeling efficiency.

Nevertheless, given a certain degree of unavoidable variation in labeling between experiments, we try to be careful with conclusions that are directly based on labeling efficiencies.

Fractions of labeled TRiC-Halo and PFD-Halo. *a*, Normalized fluorescence intensity of cells endogenously expressing TRiC-Halo or PFD-Halo when incubated with different fluorescent ligand JF646. “no significance (ns)” indicates the dye concentration at which the fluorescence signal reaches a plateau. ($n=8$ cells for each condition) *b*, In-gel fluorescence assay used to quantify labeling efficiency. To adjust the detected fluorescence intensity, 5-fold less protein was loaded for samples with dye concentration corresponding to the plateau than for those at the imaging concentration. Fluorescence from the gel (top) and total protein staining with Coomassie (bottom). $n=3$ independent experiments. Labeling efficiencies determined by in-gel fluorescence were $\sim 3.4 \pm 0.4\%$ for TRiC-Halo and $\sim 8.5 \pm 2.5\%$ for PFD-Halo. *c*, Numbers of TRiC-Halo and PFD-Halo single molecules detected in the first frame of each video. Cell numbers [n]: TRiC-Halo, $n=10$, PFD-Halo, $n=10$.

3. How long do the authors treat cells with puromycin before imaging? This treatment should be very brief (a few minutes, certainly not more than 30 min) to ensure one is looking at direct effects of nascent chain release, rather than stress-induced secondary effects of puromycin.

We thank the reviewer for this comment. Puromycin treatment was performed for exactly 30 min. This protocol was established based on preliminary experiments in which we compared the diffusion behavior of actin full-length mRNA in untreated cells and cells treated with 150 $\mu\text{g}/\text{mL}$ puromycin for different times (11, 18, 21, 31, 37, and 44 minutes) (see below). We observed that when cells were treated for 30 min or longer (dark red), the diffusion of actin mRNA was substantially accelerated, as expected upon release from ribosomes. Based on these results, we selected 30 min as the treatment duration for all subsequent experiments. We conclude that in life cells under our experimental conditions, puromycin needs ~ 30 min to exert its full effect.

Distribution of diffusion coefficients of actin full-length mRNA. Diffusion coefficients were measured under normal condition (untreated, gray) and after puromycin (Puro) treatment for different times (light red to dark red). The diffusion of mRNA molecules showed a clear difference between 21 min and 31 min of Puro treatment. Numbers of actin mRNA trajectories [n]: Untreated, n=196; 11 min, n=279; 18 min, n=197; 21 min, n=151; 31 min, n=289; 37 min, n= 240; 44 min, n=305.

4. If PFD acts as a recruitment factor for TRiC, one would expect that TRiC interactions to occur mostly on nascent chains that also interact with PFD. Moreover, the interactions of TRiC should occur at the same site or downstream (more 3' or C-terminal) to the site of PFD-nascent chain interactions. Was this ever observed?

We agree with the reviewer. As shown by selective ribosome profiling in Extend Data Fig. 3f, most TRiC substrates (~80%) also interact with PFD, supporting the model that PFD recruits TRiC to nascent chain substrates. Moreover, we found that the interactions of TRiC and PFD generally do not occur at the same sites (Extend Data Fig. 3g). To further address the reviewer's question, we analyzed the first binding events for TRiC and PFD on the shared substrates. This analysis indicates that PFD binding tends to precede TRiC engagement in 64% of mRNAs (see below and new Extended data Fig. 3h). However, we point out that selective ribosome profiling data do not provide information on whether a chaperone interaction is productive. Thus, it is possible that TRiC binding to NCs may be enhanced even if PFD binds later, consistent with the recruitment function of PFD.

First binding position of TRiC relative to PFD on shared nascent chain substrates based on selective ribosome profiling.

5. “Similar numbers of ribosome-engaging mRNA molecules were detected in the cytosol for each construct by SPT (Extended Data Fig. 5a).” There is a very small change in diffusion values for these constructs, and I’m not sure how confident one can be that one can conclude the statement above from these data.

To address this comment, we now present the diffusion behavior of the different actin mRNA constructs using an alternative representation by plotting the mean square displacement (MSD) over time (see below and new Extended Data Fig. 5a). To better present the change of the diffusion value, we also included actin full-length (FL) as a reference, shown in a separate figure in the manuscript (new Extended Data Fig. 4b). This approach better highlights the differences observed after puromycin treatment and confirms our previous conclusion. Of note, the differences in diffusion are more pronounced for actin truncations with longer coding sequences, suggesting loading of more ribosomes on translating mRNA.

To validate our conclusion that similar numbers of ribosome-engaged mRNA molecules were detected in the cytosol, we analyzed 10 cells expressing either truncated actin variants (101, 168, 305, 368) or full-length (FL) actin, and quantified the number of mRNA molecules (see below and new Extended Data Fig. 5b). On average, we observed 200 molecules for actin101, 192 for actin168, 204 for actin305, 191 for actin368, and 205 for actin FL. Statistical testing (ANOVA) revealed no significant differences between any of these conditions.

Together, these data support the conclusion that similar numbers of ribosome-engaged mRNAs were present for the different constructs.

Quantification of truncated actin mRNA molecules. a, Mean square displacement (MSD) of different actin truncations under normal condition (blue) and after puromycin (Puro) treatment (magenta). Number of cells [n]: Control: actin 101 n = 8, actin 168 n = 10, actin 305 n = 10, actin 368 n = 10, full-length (FL) actin n=10; Puro: actin 101 n = 16, actin 168 n = 15, actin 305 n = 10, actin 368 n = 9, FL actin n=8. b, Number of truncated actin mRNA molecules detected in the first frame of each video. Cell numbers [n]: 101, n=10; 168, n=10; 305, n=10; 368, n=10; FL, n=10.

6. I don't understand this sentence: "Post-translational folding of actin, which has a half-time of ~50 s, is thought to involve multiple cycles of protein encapsulation in the TRiC cavity. Indeed, the near full-length actin368 was not recognized by DNase-I (Extended Data Fig. 6a), which binds specifically and with high-affinity ($K_d \sim 2$ nM) to a loop region in SD2 of folded actin" How does lack of binding to DNase-I prove that post-translational folding of actin involves multiple cycles of protein encapsulation?

We apologize for the confusing statement. The lack of DNase-I binding to the near full-length actin 368 chains indicates that the complete actin sequence (FL actin) and its release from the ribosome are necessary for folding. This finding supports the notion that actin folding is largely a post-translational process. The reviewer is correct that it does not allow the conclusion that actin folding needs multiple TRiC cycles. We have modified the sentence to clarify this (line 181-186).

7. Fig. 1j shows that the number of PFD binding events to actin mRNA increases by ~4-fold as the coding sequence length is changed from 101 to 305 codons. Does the increased

binding frequency not merely reflect the longer dwell time of exposed nascent chains on the mRNA, as ribosomes take more time translating a CDS of 305 codons than a CDS of 101 codons?

Thank you for allowing us to clarify this important point.

By binding frequency we mean the number of binding events during an observation time of 25 s per cell, corresponding to 500 frames of 50 ms each. While the dwell time of PFD on nascent actin also increases with chain length, this increase (at most 2-fold) does not explain the increase in binding frequency. If only the dwell time increased, the binding frequency could remain the same or even decline. In this case, the values of both parameters increase. We interpret the increase in binding frequency with nascent chain length in terms of longer chains exposing more (or more extensive) binding regions for the multiple binding sites on PFD (6 in total). We interpret binding frequency as approximately correlating with on-rate for binding, and dwell time with off-rate. So overall, the result reflects a substantial increase in binding affinity of PFD as chain length increases.

8. When assessing PFD-TRiC interactions, the authors should use a negative control, ideally of mutant proteins that don't interact, or, alternatively of similar sized proteins (to have similar diffusion kinetics) to determine the frequency of co-incidental co-localization. This is needed to determine the actual interaction frequency.

We thank the reviewer for suggesting this control. We have now conducted dual-color imaging by transiently expressing TRiC-SNAP and VCP-Halo. VCP (aka P97) is a AAA-ATPase complex of ~500 kDa with similar diffusion properties to PFD (see below and new Extended Data Fig, 6l). Based on the literature and our own proteomic analysis, TRiC and VCP are not expected to functionally interact. However, given the broad chaperone function of TRiC, some interactions with VCP may nevertheless occur. TRiC-SNAP was constructed with the SNAP-tag inserted in CCT4 as for TRiC-Halo. As shown below, TRiC-SNAP and TRiC-Halo have indistinguishable diffusion properties (see below). VCP was labelled with Halo-tag at the C-terminus. Co-incidental co-localizations between VCP-Halo and TRiC-SNAP do occur, but are rare and short (see below and new Extended Data Fig. 6m-n). Importantly, the frequency of co-incidental interactions is significantly different compared to the TRiC-PFD interaction frequency (Extended Data Fig. 6n). We have modified the text to take the level of co-incidental interactions into consideration.

*Identification of co-incidental interactions. a. Distributions of diffusion coefficients of TRiC-Halo (blue) and TRiC-SNAP (magenta) are similar. b. Distribution of diffusion coefficients of PFD-SNAP (blue) and VCP-Halo (magenta) are similar. c. Interaction lifetime of TRiC-PFD and TRiC-VCP (collision control). Number of cells [n]: TRiC-PFD, n = 24; collision control (TRiC-VCP), n=25. Horizontal lines indicate mean \pm s.e.m.. * $P < 0.05$ by Welch's ANOVA. d. Co-movement events per cell (gray) and normalized co-movement events (blue) of TRiC-PFD and TRiC-VCP (collision control) under the conditions in (c). The horizontal line within the box indicates the median, boxes indicate upper and lower quartile and whisker caps 10th - 90th percentile, respectively. ** $P < 0.05$ by Mann Whitney test.*

9. In Figure 2k, the authors show that upon expression of the actin-Sun reporter, cells display a ~3-fold increase in cellular PFD-TRiC association events. How do the authors explain this striking result? As the actin-sun mRNA presumably constitutes only a small fraction of the cellular mRNA pool, how can its expression change PFD-TRiC association events to such a big extent?

Thank you for pointing this out.

Related to point 8 above, the purpose of this experiment was to obtain binding data that could be clearly assigned to the interaction of PFD with TRiC during post-translational actin folding. Actin-Sun was therefore expressed from the strong CMV promoter. Actin is an obligate substrate of TRiC. The observed increase in PFD-TRiC association events suggests that TRiC may prioritize actin over other substrates. Thus, when a folding-defective mutant actin is expressed, a substantial fraction of TRiC capacity may be occupied, displacing other substrates. We are now making this point more clearly in the text (line 223-228).

10. When examining post-translation interactions of PFD and TRiC with actin using the Xbp1 sequence, did the authors determine whether either binds to the SunTag sequence of Xbp1 sequence and normalize their data for such binding?

To directly address this point, we have conducted experiments using the same post-translational reporter construct but lacking actin, so that only the SunTag and Xbp1u+ are translated on the ribosome. Analysis of both interaction lifetimes and number of interaction events of this construct with TRiC and PFD showed that binding to the SunTag-Xbp1u+ is minor (low frequency and shorter lifetime), with ANOVA testing confirming the significant difference between constructs with and without actin (i.e. SunTag-Xbp1u+ alone) (see below and revised Fig. 2d,e). Notably, the interactions between TRiC/PFD-Halo and the SunTag construct without actin closely resemble those between Ribosome-SNAP and TRiC-PFD-Halo following puromycin treatment (Fig. 1d,e). This suggests that the interactions observed with SunTag-Xbp1u+ alone are not with the SunTag itself, but rather with the ribosome (or its associated factors) to which the SunTag is tethered via the Xbp1u+ sequence. However, analyzing interactions between TRiC/PFD-Halo and a SunTag construct without the Xbp1u+ arrest sequence is not feasible, as the free SunTag diffuses too rapidly to allow meaningful measurements.

In further support of the absence of significant interactions with the SunTag-Xbp1u+ sequence, we note that Actin-Sun is released from TRiC with kinetics correlating with actin folding, as demonstrated by bulk measurements using TRiC-Halo pulldown, followed by anti-SunTag immunoblotting in Extended Fig.6g,h. Notably, this release occurs with the same kinetics as the release of actin without tag⁷. This would not be expected if TRiC were to retain significant affinity for the SunTag-Xbp1 sequence. Moreover, a single point mutation in actin (actin G150P-Sun) markedly changes the interaction properties, consistent with binding parameters being dominated by the folding properties of the actin moiety.

Post-translational interaction between TRiC-actin and PFD-actin. a. Distribution of interaction lifetimes of

*co-translational and post-translational TRiC-actin and PFD-actin interactions. As a control (Ctrl), interactions of PFD and TRiC with SunTag-Xbp1u+ without actin were analyzed. Number of cells [n]: TRiC-actin-Co, n = 38; TRiC-actin-Post, n = 24; TRiC-actin-Post/PFD-KD, n = 24; TRiC-SunTag control, n=29; PFD-actin-Co, n = 38; PFD-actin-Post, n = 38; PFD-SunTag control, n=31). Horizontal lines indicate mean \pm s.e.m.. * $P < 0.05$, ** $P < 0.01$, *** $P < 0.001$ and **** $P < 0.0001$ by Welch's ANOVA. b, Co-movement events per cell (gray) and normalized co-movement events (blue) analyzed in (a). The horizontal line within the box indicates the median, boxes indicate upper and lower quartile and whisker caps 10th - 90th percentile, respectively. * $P < 0.05$, ** $P < 0.01$, *** $P < 0.001$ and **** $P < 0.0001$ by ANOVA.*

11. Figure 4a-f are just examples. One can always find an example molecule that shows some sort of behavior. I would like to see quantification and statistics to prove that local retention of TRiC/PFD is indeed occurring. Similarly, I would like to see quantification of the so-called stationary molecules that are observed.

We thank the reviewer for this comment, which led us to analyze the data more quantitatively. We performed a quantitative analysis of all the interactions between TRiC and actin in the co- and post-translational mode. This analysis is rather time consuming due to the necessity for manual curation. We therefore focused only on TRiC. It is important to note that we used TIRF illumination for data acquisition, which only excites a thin section of the cell. In addition, free TRiC is dynamic and often moves in and out of focus. Such molecules will automatically be counted as different, potentially leading to an underestimation of On-Off-On events. To exclude this effect, we only considered TRiC trajectories in which TRiC remained in focus for at least 500 ms (10 frames) after detaching from substrate (actin nascent chain or completely translated actin with SunTag-Xbp1u+), thus allowing re-binding to be observed. Based on this criterion, we observed that 41% of such trajectories showed On-Off-On behavior during translation (Co) and 33% of them showed On-Off-On binding after completion of actin translation (post) (see below and new Extended Data Fig. 8g). This analysis is now explained in the text (line 282-283) and also legend for new Extended Data Fig. 8g. Note that the lower On-Off-On frequency during post-translational folding is consistent with TRiC not re-binding when actin is successfully folded in the preceding interaction cycle.

To address the reviewer's second question regarding the quantification of the stationary molecules, we further analyzed the diffusion behaviour of TRiC molecules in the Off-state prior to re-binding. For the co-translational events, we find that 80% of these TRiC molecules show stationary behavior and 20% are sub-diffusive, while for the post-translational events, 67% are stationary and 33% are sub-diffusive (see below and new Extended Data Fig. 8h). The distance from substrate that TRiC molecules explore in the Off-state is up to 1.5 μm (Fig.4d), with an average maximal distance of 0.79 μm and 0.70 μm for Co and Post, respectively (see below and new Extended Data Fig. 8f). These results are now described in the text (line 279-280).

Quantification of On-Off-On binding events. a, Frequency of co- and post-translational On-Off-On binding events between TRiC-Halo and actin. Number of trajectories analyzed: Co, 59 trajectories with Off-state TRiC remaining in focus for 500 ms out of 133 total trajectories. 24 trajectories with On-Off-On behavior (41%); Post, 42 trajectories with Off-state TRiC remaining in focus for 500 ms out of 118 total trajectories. 14 trajectories with On-Off-On behavior (33%). b, Diffusion behavior of TRiC-Halo during the Off state before rebinding in the co- and post-translational mode. c, Average maximal distance between TRiC-Halo and actin in the co- and post-translational Off state. Number of cells: TRiC-Co, n=38; TRiC-Post, n=24.

12. Was the local retention of TRiC and PFD observed in specific regions of the cell or does this phenomenon occur equally throughout the cell? I don't understand the proposed model of a 'protective folding compartment'. A typical human cell has ~5 million ribosomes, are there 5 million local folding compartments, one around each ribosome? And ribosomes diffuse rapidly, are the folding compartments moving along?

The reviewer's assumption does not fully reflect the spatial organization of ribosomes in cells. While a human cell indeed contains millions of ribosomes, they are not evenly dispersed as individual units but are typically organized into polysomes. Moreover, polysomes themselves often cluster in specific subcellular regions, for instance near the endoplasmic reticulum or within localized translation hotspots⁸. This implies that 'folding compartments' would not correspond to each individual ribosome, but rather to groups of ribosomes and their associated translation machinery. Consequently, instead of millions of isolated folding compartments, one would expect several thousand localized folding hubs, which is a more plausible and physiologically consistent scenario. We envision that in the context of polysomes, weak interactions between translation factors

and chaperones add up to exert a significant retention force, resulting in a local concentration of chaperones such as PFD and TRiC.

Interestingly, a recent report by Gade et al.⁹ shows that the diffusion of genetically encoded multimers (GEMs) with a diameter of 20 nm used as rheological probes (TRiC has a diameter of ~16 nm¹⁰) is slowed in cells containing polysomes compared to puromycin treated cells where ribosomes are not assembled into polysomes. We now include this reference in the discussion.

13. I did not understand how the experiments in Fig 4h-l explain the local retention of TRiC molecules near clients. The TRiC molecule moves up to 1.5 micron away from the client, I assume such a distance cannot be bridged by a multimer of TRiC. This section is difficult to understand the logic.

The reviewer is correct that a TRiC cluster (up to 7 molecules were observed to form a cluster in the paper by Xing et al.¹⁰), would not be sufficient to bridge this distance. Such a cluster could have a dimension of at most 100 nm, while the maximum distance of TRiC from substrate in the Off-state is ~0.7 μm on average (see above). What we suggest is that the retention of TRiC in proximity to substrate is due to a combination of interactions, with a contribution of TRiC-TRiC interactions. We have rewritten this paragraph to express this more clearly.

Minor comments:

- The descriptions for figure panels m and n in Extended Data Fig. 1 are missing
We apologize and thank the reviewer for detecting this error. We have updated the figure legends accordingly.

- Line 140: PFD KD reduced TRiC binding to actin NCs by ~70% (Fig. 1g and Extended Data Fig. 4h). How was this number calculated? The fold-change presented in the Fig. 1g does not appear to be 70%.

We thank the reviewer for this comment. After PFD knockdown (KD), the interaction lifetimes between TRiC and its substrates became shorter, causing most data points to cluster near the bottom of the plot and making individual events not clearly distinguishable. To calculate binding frequency, we used the average number of TRiC-actin interaction events per cell under normal conditions as the reference (set to 100%). Under normal conditions, we observed 133 interaction events from 38 cells (an average 3.5 events per cell). After PFD KD, we observed 53 interaction events from 45 cells (an average 1.2 events per cell). This corresponds to ~34% of the normal binding frequency, i.e., a ~66% reduction (see Fig. 1g and Extend Data Fig. 4g).

- Line 171: ... in agreement with ribosome profiling data (Extended data Fig. 5g).

There is no Extended data Fig. 5g. Was 5e meant instead?

We apologize. The correct reference should be Extended Data Fig. 5e, not 5g in the original manuscript. In the revised manuscript, it is Extended Data Fig. 5f. We have corrected this in the revised manuscript (168-169).

- Sun-tag is typically written as SunTag in the literature

We have changed the writing as suggested.

- In extended data figure 8e, the authors measure the distance between nascent chains and the 3'UTR labeled by MS2. However, in this setup, individual mRNAs are likely bound by multiple ribosomes, each with a nascent chain. I assume they cannot resolve individual nascent chains and are thus measuring the distance between the average nascent chain the MS2 signal. It is possible that some nascent chains, especially those at the 5' end of the mRNA are further away from the MS2 signal.

Yes, the reviewer is correct that what we measure is a distance distribution between NCs in a polysome and their respective mRNA. However, the largest distance we detect is 385 nm, with the frequency of distances longer than 300 nm being 0.13% (3 out of 2211) (see Extended Data Fig. 8e). Moreover, it should be considered that the 3' polyA sequence of the mRNA is thought to contact the 5' region¹¹, which would help to explain the narrow distance distribution observed.

Reference

- 1 Siegert, R., Leroux, M. R., Scheufler, C., Hartl, F. U. & Moarefi, I. Structure of the molecular chaperone prefoldin: unique interaction of multiple coiled coil tentacles with unfolded proteins. *Cell* **103**, 621–632 (2000). [https://doi.org/10.1016/s0092-8674\(00\)00165-3](https://doi.org/10.1016/s0092-8674(00)00165-3)
- 2 Gestaut, D. *et al.* The chaperonin TRiC/CCT associates with prefoldin through a conserved electrostatic interface essential for cellular proteostasis. *Cell* **177**, 751–765 e715 (2019). <https://doi.org/10.1016/j.cell.2019.03.012>
- 3 Siegers, K. *et al.* Compartmentation of protein folding in vivo: sequestration of non-native polypeptide by the chaperonin-GimC system. *EMBO J* **18**, 75–84 (1999). <https://doi.org/10.1093/emboj/18.1.75>
- 4 Blanca, M. J., Alarcon, R., Arnau, J., Bono, R. & Bendayan, R. Non-normal data: is ANOVA still a valid option? *Psicothema* **29**, 552–557 (2017). <https://doi.org/10.7334/psicothema2016.383>
- 5 Lix, L. M., Keselman, J. C. & Keselman, H. J. Consequences of assumption violations revisited: A quantitative review of alternatives to the one-way analysis of variance F test. *Rev Educ Res* **66**, 579–619 (1996). <https://doi.org/Doi 10.3102/00346543066004579>
- 6 Wisniewski, J. R., Hein, M. Y., Cox, J. & Mann, M. A "proteomic ruler" for protein copy number and concentration estimation without spike-in standards. *Mol Cell Proteomics* **13**, 3497–3506 (2014). <https://doi.org/10.1074/mcp.M113.037309>

- 7 Thulasiraman, V., Yang, C. F. & Frydman, J. newly translated polypeptides are sequestered in a
protected folding environment. *Embo Journal* **18**, 85–95 (1999). <https://doi.org/DOI>
10.1093/emboj/18.1.85
- 8 Zhang, Z. J. *et al.* A subcellular map of translational machinery composition and regulation at the
single-molecule level. *Science* **387** (2025). <https://doi.org/ARTN>
eadn262310.1126/science.adn2623
- 9 Gade, V. R. *et al.* Polysomes and mRNA control the biophysical properties of the eukaryotic
cytoplasm. *Cell Rep* **44**, 116204 (2025). <https://doi.org/10.1016/j.celrep.2025.116204>
- 10 Xing, H. *et al.* In situ analysis reveals the TRiC duty cycle and PDCD5 as an open-state cofactor.
Nature **637**, 983–990 (2025). <https://doi.org/10.1038/s41586-024-08321-z>
- 11 Fakim, H. & Fabian, M. R. Communication Is Key: 5'-3' Interactions that Regulate mRNA
Translation and Turnover. *Adv Exp Med Biol* **1203**, 149–164 (2019). [https://doi.org/10.1007/978-
3-030-31434-7_6](https://doi.org/10.1007/978-3-030-31434-7_6)

Response to reviewers

Referee Q1:

In their revised manuscript, Li et al. have adequately addressed the concerns we raised about the earlier version, with one exception, described below. Moreover, we were gratified to read the extremely careful and technically expert review offered by Reviewer 3. It seems that this led the authors to carry out further analyses and some experiments that strengthen the results of their study. Compliments to this reviewer and to the authors!

The one point that we remain concerned about is the use of the term “dynamics”. In order to persuade the authors that this term may confuse the reader and also does not correctly convey the results they present in this project, we did some “homework” using an AI search to check whether our understanding of this term is generally acknowledged. We attribute a meaning to “dynamics” that relates to molecular motions (from side chains to domains) and conformational changes among different conformational states. By contrast, the term that describes the timing of binding events, including on/off rates and residence times, is “kinetics”. We feel strongly that the work in this paper is “kinetics”, not “dynamics”. We cite below the AI retrieved definitions and references that support them. To make their title more correct, the authors could state “The timing of TRiC chaperone action in vivo by single molecule particle tracking” or a variation on this.

From an AI search:

1. Kinetics = Rates of binding and unbinding

Kinetics describes how fast binding and dissociation happen.

- It's about rate constants:
 - o k_{on} (association rate) → how quickly two molecules find and bind each other.
 - o k_{off} (dissociation rate) → how quickly the complex falls apart.
- The equilibrium dissociation constant, $KD=k_{off}/k_{on}$, gives the overall binding affinity.
- Kinetics focuses on measurable rates and time scales — what you'd see in a SPR (surface plasmon resonance) or stopped-flow experiment.

Example:

If Protein A binds Protein B with a high k_{on} and low k_{off} , it binds quickly and stays bound — strong and stable interaction.

2. Dynamics = Motions and conformational changes

Dynamics describes the structural motions and flexibility that occur before, during, and after binding.

- It's about how proteins move and fluctuate over time (e.g., side-chain rearrangements, loop flexibility, domain motions).

- Protein dynamics influence how accessible binding sites are and how the binding interface adapts.
- Studied with techniques like NMR relaxation, molecular dynamics simulations, or single-molecule FRET.

Example:

A protein may need to “breathe” (open a loop) to expose its binding site — this conformational flexibility affects how and when binding occurs, but not directly the rate constants themselves.

In short

Concept Focus Key Quantities Typical Methods Biological Meaning

Kinetics How fast binding/unbinding occurs k_{on} , k_{off} , K_D SPR, stopped-flow, ITC

Reaction rates, affinity

Dynamics How proteins move and fluctuate Conformational ensembles, time

correlations NMR, MD simulations, smFRET Structural flexibility, induced fit, allostery

Some expert references on this distinction:

1. Protein Binding Kinetics

These works describe the mathematical and experimental basis of association (k_{on}) and dissociation (k_{off}) rates:

- Copeland, R. A. (2016). The drug–target residence time model: a 10-year retrospective. *Nature Reviews Drug Discovery*, 15, 87–95.

Defines kinetic parameters (k_{on} , k_{off} , K_D) and explains how binding kinetics govern biological efficacy.

- Swinney, D. C. (2004). Biochemical mechanisms of drug action: what does it take for success? *Nature Reviews Drug Discovery*, 3, 801–808.

Clear description of kinetic parameters in ligand–protein interactions.

- Schreiber, G., & Keating, A. E. (2011). Protein binding specificity versus promiscuity. *Current Opinion in Structural Biology*, 21, 50–61.

Reviews how kinetic rate constants underlie specificity and affinity in protein–protein interactions.

2. Protein Dynamics and Conformational Flexibility

These sources describe motions, conformational ensembles, and structural transitions that underlie binding:

- Henzler-Wildman, K. A., & Kern, D. (2007). Dynamic personalities of proteins. *Nature*, 450, 964–972.

Classic review on protein dynamics across timescales and how these motions enable function, including ligand binding.

- Boehr, D. D., Nussinov, R., & Wright, P. E. (2009). The role of dynamic conformational

ensembles in biomolecular recognition. *Nature Chemical Biology*, 5, 789–796.
Explains how protein dynamics affect recognition and binding, introducing “conformational selection” and “induced fit” models.

- Nussinov, R., Tsai, C.-J., & Ma, B. (2013). The key role of protein dynamics in allosteric effects and drug discovery. *Nature Chemical Biology*, 9, 693–694.
Highlights how conformational flexibility governs binding and allostery.

3. Linking Kinetics and Dynamics

These discuss how dynamics modulate kinetics, bridging both concepts:

- Hammes, G. G., Chang, Y.-C., & Oas, T. G. (2009). Conformational selection or induced fit: a flux description of reaction mechanism. *PNAS*, 106, 13737–13741.
Quantitative model linking conformational dynamics to observed binding kinetics.
- Changeux, J.-P., & Edelstein, S. J. (2011). Conformational selection or induced fit? 50 years of debate resolved. *FASEB Journal*, 25, 1169–1175.
Integrates the two views, showing how intrinsic dynamics influence kinetic rate constants.

Summary from Sources

Together, these establish that:

- Kinetics quantifies the rates of binding/unbinding via measurable constants (Copeland 2016; Schreiber & Keating 2011).
- Dynamics describes structural motions and conformational ensembles that make binding possible (Henzler-Wildman & Kern 2007; Boehr et al. 2009).
- The two connect via conformational selection or induced fit models (Hammes et al. 2009; Changeux & Edelstein 2011).

We thank the reviewer for the very favorable comments and sincerely appreciate the thoughtful discussion regarding the most appropriate term to describe the interaction between the chaperonin and its substrates. As the reviewer correctly noted, *dynamics* refers to molecular motions (from side chains to domains) and conformational changes among different conformational states, but it can also be used to describe **time-dependent changes in binding or association**. In contrast, *kinetics* specifically describes **measurable rates of binding and dissociation**, typically expressed as the association (k_{on}) and dissociation (k_{off}) rate constants.

In our experiments using single particle tracking, we describe the interaction lifetime between chaperone and substrates under different conditions (e.g., puromycin treatment, PFD knockdown, domain-specific actin truncations, actin mutations, or co-versus post-translational interactions). Rather than determining k_{on} and k_{off} values, we focus on the temporal behavior and variability of these interactions, as well as the

trajectories and diffusion behavior of the chaperones. Therefore, we believe that the term “dynamics,” being broader and more descriptive, more accurately reflects the nature of our experimental observations.

Nevertheless, we fully agree with the reviewer that *dynamics* can also imply molecular motions and conformational transitions. To avoid ambiguity, we have clarified in the revised manuscript that in our context, *dynamics* specifically refers to **changes in binding or association behavior over time**. These changes were already made in response to the first round of comments. Please see lines 46-49 at the beginning of Introduction where we write: **“A key unresolved aspect of this fundamental process is the dynamic behavior of chaperone systems in vivo; how long, how often and when during protein biogenesis do chaperones interact with their client proteins?”**

In addition, we also reviewed relevant literature describing time-dependent interactions, where the term “*dynamics*” is commonly used rather than “*kinetics*,” for example:

1. Yan X, Hoek T A, Vale R D, et al. Dynamics of translation of single mRNA molecules in vivo[J]. *Cell*, 2016, 165(4): 976-989.
2. Chen, J., Zhang, Z., Li, L., Chen, B. C., Revyakin, A., Hajj, B., ... & Liu, Z. (2014). Single-molecule dynamics of enhanceosome assembly in embryonic stem cells. *Cell*, 156(6), 1274-1285.
3. Morisaki, T., Lyon, K., DeLuca, K. F., DeLuca, J. G., English, B. P., Zhang, Z., ... & Stasevich, T. J. (2016). Real-time quantification of single RNA translation dynamics in living cells. *Science*, 352(6292), 1425-1429.
4. Wu, B., Eliscovich, C., Yoon, Y. J., & Singer, R. H. (2016). Translation dynamics of single mRNAs in live cells and neurons. *Science*, 352(6292), 1430-1435.
5. Wheat, J. C., Sella, Y., Willcockson, M., Skoultchi, A. I., Bergman, A., Singer, R. H., & Steidl, U. (2020). Single-molecule imaging of transcription dynamics in somatic stem cells. *Nature*, 583(7816), 431-436.
6. Morisaki, T., Wiggan, O. N., & Stasevich, T. J. (2024). Translation dynamics of single mRNAs in live cells. *Annual review of biophysics*, 53(2024), 65-85.
7. Boersma, S., Rabouw, H. H., Bruurs, L. J., Pavlovič, T., van Vliet, A. L., Beumer, J., ... & Tanenbaum, M. E. (2020). Translation and replication dynamics of single RNA viruses. *Cell*, 183(7), 1930-1945.
8. Horvathova, I., Voigt, F., Kotrys, A. V., Zhan, Y., Artus-Revel, C. G., Eglinger, J., ... & Chao, J. A. (2017). The dynamics of mRNA turnover revealed by single-molecule imaging in single cells. *Molecular cell*, 68(3), 615-625.
9. Liu, Z., & Tjian, R. (2018). Visualizing transcription factor dynamics in living cells. *Journal of Cell Biology*, 217(4), 1181-1191.

10. Esbin, M. N., Cookis, T., Anantakrishnan, S., Abidi, A. A., Karr, J., Cattoglio, C., ... & Tjian, R. (2025). Assembly and Dynamics of Transcription Initiation Complexes. *Annual Review of Biochemistry*, 94.

Finally, we additionally include below various explanations obtained from an AI search that summarize the conceptual distinction between *kinetics* and *dynamics* in the context of molecular association.

We begin with how AI defines Single particle tracking and what it can do:

“Single-particle tracking (SPT) provides insight into **molecule dynamics** by following individual fluorescently-labeled molecules to create trajectories that **reveal their movement, interactions, and function over time**. This allows researchers to analyze individual behaviors like diffusion, speed, and distance, which can reveal different dynamic populations, the duration of interactions, and the mechanics of molecular processes in a way that averaging from bulk measurements cannot.”

1. Kinetics = quantitative rates of association and dissociation

Kinetics describes the **measurable rates** of molecular events — how fast something happens.

It is **quantitative**, often expressed with **rate constants**.

Example (protein A–B binding):

Parameter	Meaning
k_{on} (association rate constant)	How fast A binds B
k_{off} (dissociation rate constant)	How fast A–B complex falls apart
$K_d = k_{off} / k_{on}$	Binding affinity (equilibrium constant)

□ **Kinetics** is what you measure in a binding experiment (SPR, BLI, stopped-flow, etc.).

It answers “**how fast**” and “**how strong**”.

□ 2. Dynamics = structural or temporal behavior of the system

Dynamics is a broader, more descriptive term.

It refers to **how the system changes over time** — not necessarily quantified by rate constants.

Dynamics can include:

- Structural fluctuations of A and B during binding
- Conformational rearrangements after complex formation
- Transient vs. stable interactions
- Regulation over time or under different cellular conditions

□ So, “interaction dynamics” often describes the **nature and variability** of the association rather than the rate.

□ Think of it this way:

Term	Focus	Usually measured by	Typical wording
Kinetics	Speed and affinity of binding	SPR, stopped-flow, BLI	“We measured the kinetics of A–B association (k_{on} , k_{off}).”
Dynamics	Time-dependent structural or regulatory changes	MD simulations, FRET, live imaging	“We analyzed the dynamics of A–B interaction during stress response.”

□ 3. Example to illustrate

- **Kinetics:**
“The A–B complex forms with a k_{on} of $1 \times 10^5 \text{ M}^{-1}\text{s}^{-1}$ and a k_{off} of $1 \times 10^{-3} \text{ s}^{-1}$.”
→ Quantitative, rate-based.
- **Dynamics:**
“The A–B interaction is highly dynamic, with transient contacts and conformational fluctuations.”
→ Qualitative, describing flexibility or time-dependent association.

Referee Q3:

Point 4: The fact that TRiC frequently (33%) binds before PFD is not consistent with a simple 'recruitment' model. Perhaps PFD can act as a recruitment factor in some cases, but additionally it could stabilize transient binding events of TRiC with the RNA (which I wouldn't call 'recruitment'). It would be nice if the authors can specify this in the statement on lines 130-131.

We thank the reviewer for this insightful comment. Indeed, we agree that since TRiC binds prior to PFD for approximately one-third of its substrates, it is not accurate to describe this as a general recruitment model. This observation suggests that TRiC interacts with a subset of substrates in a manner that is not PFD dependent, at least initially. We already stated in lines 139-143 of the revised manuscript: **“Notably, the TRiC interactions remaining after PFD KD were shifted to shorter lifetimes (Fig. 1g) with only 11% of binding events lasting for >1 s (Extended Data Fig. 4h). This suggests that PFD, beyond recruiting TRiC, prolongs TRiC-NC associations, either through direct interaction or by acting as a holding chaperone in stabilizing actin in a conformation competent for TRiC binding”**. We have now inserted an additional statement in lines 115-117: **“Together, these findings indicate that both TRiC and PFD monitor a broad range of NCs primarily through brief, dynamic interactions, with PFD serving as recruitment factor for TRiC for most substrates.”**

Point 7: From the data shown in Fig. 1j, it is evident that increasing the length of the actin coding sequence results in a corresponding increase in the dwell time of PFD. However, the observed increase in the number of interaction events (Fig. 1j, upper panel) can be interpreted differently: The translation of actin 305, for example, would take approximately three times longer than that of actin 101, thereby providing more time for PFD to bind to the actin nascent actin and co-localize with the labeled mRNA as the coding sequence lengthens. In addition, the ribosome exit tunnel accommodates roughly 30–40 amino acids, which prevents nascent actin from being exposed during the early stages of translation. Taken together, these considerations suggest that the 4-fold increase in PFD interaction events between actin 101 and actin 305 can be simply explained by the prolonged availability of nascent actin chains on the mRNA, rather than by a size-dependent increase in the PFD on-rate for individual nascent chains. The same logic also applies to TRiC's interaction with the growing nascent chain (Fig. 1k).

We agree that it is reasonable to assume that the longer exposure of actin 305 chains on the ribosome contributes to the observed increase in binding frequency. To clarify this, we have changed the description of this result to read as follows (lines 154-159):

“The PFD binding events with actin 305, exposing up to the complete SD2 and SD4, were ~4-times more frequent than binding to actin 101, exposing only the N-terminal segment of SD1 (Fig. 1i,j and Extended Data Fig. 5c), **consistent with the longer availability of actin 305 NCs on the ribosome during translation**. In addition, a gradual upshift in interaction lifetimes during translation was observed from actin 305 to actin 368 to FL actin (Fig. 1j), **suggesting that PFD binding strength was also enhanced by emergent conformational properties of the growing NCs.**”

Point 11: “To exclude this effect, we only considered TRiC trajectories in which TRiC remained in focus for at least 500 ms (10 frames) after detaching from substrate (actin nascent chain or completely translated actin with SunTag-Xbp1u+), thus allowing re-binding to be observed.”

I feel that these analysis criteria will overestimate the number of events called as ‘local retention’, as most events of local retention will be included in this analysis (as they remain local, they are more likely to remain in the z-plane of imaging), but the majority of non-local retention events will be discarded (as in the latter case, the molecule is far more likely to leave the z-plane of imaging). Thus, this analysis likely (greatly) overestimates the number of events of local retention. It would be good if the authors can show the same analysis, but without the inclusion criteria of spots remaining with the imaging plane for a set number of time-points.

We understand the reviewer’s concern and have re-analyzed the frequency of local retention events as suggested. The On–Off–On binding behavior between TRiC and actin was observed in 18% and 12% of all detected co- and post-translational interactions, respectively. It is worth noting that our imaging was performed under TIRF illumination, with a penetration depth of less than 500 nm. However, during the “Off” state, the chaperone can diffuse more than 1 μm away from the substrate (see Extended Data Fig. 8f). Therefore, analyzing the frequency of local retention events using all the trajectories will strongly underestimate the frequency of these events. Given that only ~3% of TRiC complexes are labeled, the frequency of On-Off-On behavior is nevertheless substantially higher than expected if the TRiC complexes interacting with a specific substrate were in free exchange with all TRiC particles. We conclude that On-Off-On behavior is a robust phenomenon, providing a mechanism that likely contributes to the efficiency of chaperone-assisted folding. The generally lower frequency of this behavior in the post-translational mode is consistent with re-binding no longer occurring when actin has reached native state in the preceding interaction cycle.

We have added the data from all trajectories (see dashed lines in Extended Fig. 8g) and modified the text in lines 285-288 to read as follows: “While this On-Off-On binding behaviour was observed in only ~12-18% of all recorded co- and post-interaction trajectories, due to the limited depth of TIRF illumination, it was observed for ~30-40% of TRiC complexes that could be tracked long enough in the Off-state to detect re-binding (Extended Data Fig. 8g).”

Point 12: The average polysome contains perhaps 3-7 ribosomes, so that would still leave close to 1 million polysomes per cell. To get to the value of several thousand folding hubs, on average dozens to hundreds of polysomes would need to cluster together in each of these folding hubs. Such clusters would be very large, and show very slow diffusion. Rather, what is observed in this study, as well as many previous studies is that diffusion of typical mRNAs is quite fast (relative to the expected diffusion of cluster of many polysomes that form a hub), for example see PMID: 26760529. If association of mRNAs with folding hubs would be translation dependent, then puromycin treatment should dramatically increase diffusion rates, by several orders of magnitude (as mRNAs would go from being associated with hubs of dozens of mRNAs, ribosomes and co-factors, to single mRNAs). In reality the diffusion speed difference is much smaller.

Perhaps such hubs exist for a (subset of) polysomes, but based on the current data, I don't think it is likely that most, or even many polysomes are part of such hubs. I would suggest the authors put the existence of folding hubs into this context.

Additionally, the authors actually do not address my primary question, whether the 'hubs' are observed in a specific region of the cell, for example perinuclear, a region where more ribosomes are likely engaged with the ER.

We thank the reviewer for this interesting discussion, but note that most of this detailed exchange had been limited to our response to reviewer comments and cannot be treated in depth in the actual manuscript due to space limitations. However, to take the relevant comment of the reviewer into account, we now refer to the Katz et al. reference in this context and have changed the respective statement in the discussion to read in line 353-356: “This finding suggests a supra-molecular organization of the chaperonin system, perhaps most prevalent at local translation hot-spots²⁷, that generates a ‘virtual folding compartment’ through a network of low affinity interactions with translation machinery and/or other chaperone factors, effective under conditions of (local) macromolecular crowding^{25,52,53}.”.

We apologize for not having answered the reviewer's question as to the possible localization of the 'hubs'. This aspect needs to be analyzed in the future. We note that TIRF illumination microscopy only allows us to observe a thin layer of the cell.

Combining these analyses with labeling of specific subcellular structures is outside the scope of the current study.

Referee Q4: Co-reviewer

Referee Q5: Co-reviewer.

Response to reviewers

Referee #3

The following sentence could benefit from some restructuring and additional explanation regarding how the restricted TIRF illumination influenced the analysis.

"While this On-Off-On binding behaviour was observed in only ~12-18% of all recorded co- and post-interaction trajectories, due to the limited depth of TIRF illumination, it was observed for ~30-40% of TRiC complexes that could be tracked long enough in the Off-state to detect rebinding (Extended Data Fig. 8g)"

We thank the reviewer for this comment.

We have restructured the sentence in line 287-292 as follows: "On-Off-On binding behaviour was observed in ~12-18% of all recorded co and post interaction trajectories. However, due to the limited depth of TIRF illumination (< 500 nm), TRiC may leave the focal plane during the "Off" state and thus re-binding cannot be tracked, likely underestimating re-binding frequency. Indeed, On-Off-On cycles were observed for ~30-40% of TRiC complexes that could be tracked long enough in the Off-state to detect re-binding (Extended Data Fig. 8g)."